# Mixing state of refractory black carbon aerosol in the South Asian outflow over the northern Indian Ocean during winter

Sobhan Kumar Kompalli[1], Surendran Nair Suresh Babu[1], Krishnaswamy Krishnamoorthy[2], Sreedharan Krishnakumari Satheesh[2,3], Mukunda M Gogoi[1], Vijayakumar S Nair[1], Venugopalan Nair Jayachandran[1],

5    Dantong Liu[4], Michael Flynn[4], Hugh Coe[4]

[1]Space Physics Laboratory, Vikram Sarabhai Space Centre, Thiruvananthapuram, India.

[2]Centre for Atmospheric & Oceanic Sciences, Indian Institute of Science, Bengaluru, India.

[3]Divecha Centre for Climate Change, Indian Institute of Science, Bengaluru, India

10   [4]Centre for Atmospheric Science, School of Earth and Environmental Sciences, University of Manchester, Manchester, U.K.

*Correspondence to*: S. Suresh Babu (sureshsplvssc@gmail.com), Sobhan Kumar Kompalli (sobhanspl@gmail.com)

**Abstract.** Regional climatic implications of aerosol black carbon (BC) are well recognized over South Asia, which has a wide variety of anthropogenic sources in a large abundance. Significant uncertainties remain in its quantification due to lack of sufficient information on the microphysical properties (its concentration, size, and mixing state with other aerosol components), which determine the absorption potential of BC. Especially the information on the mixing state of BC is extremely sparse over this region. In this study, first-ever observations of the size distribution and mixing state of individual refractory black carbon (rBC) particles in the south Asian outflow to Southeastern Arabian Sea, northern and equatorial Indian Ocean regions are presented based on measurements using a single particle soot photometer (SP2) aboard the ship cruise of the Integrated Campaign for Aerosols, gases, and Radiation Budget (ICARB-2018) during winter-2018 (16 January to 13 February). The results revealed significant spatial heterogeneity of BC characteristics. The highest rBC mass concentrations ($\sim$938 $\pm$ 293 ng m$^{-3}$) with the highest relative coating thickness (RCT; the ratio of BC core to its coating diameters) of $\sim$2.16 $\pm$ 0.19 are found over the Southeast Arabian Sea (SEAS) region, which is in the proximity of the continental outflow. As we move to farther oceanic regions, though the mass concentrations decreased by nearly half ($\sim$546 $\pm$ 80 ng m$^{-3}$), BC still remained thickly coated (RCT $\sim$ 2.05 $\pm$ 0.07). The air over the remote equatorial Indian Ocean, which received considerable marine air masses compared to the other regions, showed the lowest rBC mass concentrations ($\sim$206 $\pm$ 114 ng m$^{-3}$), with a moderately thick coating (RCT $\sim$1.73 $\pm$ 0.16). Even over oceanic regions far from the landmass, regions which received the outflow from the more industrialized east coast/the Bay of Bengal had thicker coating ($\sim$104 nm) compared to regions that received outflow from the west coast/peninsular India ($\sim$86 nm). Although different regions of the ocean depicted contrasting concentrations and mixing state parameters due to varying extent and nature of the continental outflow as well as the atmospheric lifetime of air masses, the modal parameters of rBC mass-size distributions (mean mass median diameters $\sim$ 0.19-0.20 $\mu$m) were similar over all the regions. The mean fraction of BC containing particles ($F_{BC}$) varied in the range 0.08-0.12 (suggesting significant amounts of non-BC particles), whereas the bulk mixing ratio of coating mass to rBC mass was highest (8.31 $\pm$ 2.40) over the outflow regions compared to the remote ocean (4.24 $\pm$ 1.45) highlighting the role of outflow in providing condensable material for coating on rBC. These parameters, along with the information on the size-resolved mixing state of BC cores, throw light on the role of sources and secondary processing of their complex mixtures for coating on BC under highly polluted conditions. Examination of the non-refractory sub-micrometre aerosol chemical composition obtained using the aerosol chemical speciation monitor (ACSM) suggested that the overall aerosol system was sulfate-dominated over the far-oceanic regions. In contrast, organics were equally prominent adjacent to the coastal landmass. Association between the BC mixing state and aerosol chemical composition suggested that sulfate was the probable dominant coating material on rBC cores.

## 1. Introduction

Black carbon (BC) is the dominant light-absorbing atmospheric aerosol species that perturbs regional and global radiation balance through the positive radiative forcing arising out of its strong absorption of solar radiation and its ability to reduce cloud albedo (Menon et al., 2002; Ramanathan and Carmichael, 2008; IPCC 2013; Bond et al., 2013; Huang et al., 2016). Produced by the incomplete combustion of hydrocarbon fuels, BC has a global direct radiative forcing of + 0.71 W m$^{-2}$ (+0.08 to +1.26 W m$^{-2}$), of which fossil and biofuel emissions contribute +0.51 W m$^{-2}$, and the rest is from biomass burning (Bond et al., 2013). Such large forcing due to BC is reported to be capable of causing significant perturbations to atmospheric circulation, cloud dynamics, rainfall pattern, static stability, and convective activity over regional scales, especially over the Indian region (Menon et al., 2002; Ramanathan et al., 2005; Meehl et al., 2008; Bollasina et al., 2008; Lawrence and Lelieveld 2010; Babu et al., 2011; D'Errico et al., 2015; Boos and Storelvmo, 2016). While fresh BC is fractal-like, hydrophobic, and externally mixed, atmospheric ageing (temporally and/or chemically) results in internally mixed BC with hydrophilic compounds (e.g., organic acids and ammonium sulfate) and altered mixing state, size, and morphology. Also, the ageing process leads to enhanced absorption potential of BC (Schnaiter et al., 2005; Shiraiwa et al., 2010; Cappa et al., 2012, 2019; Zhang et al., 2015; Peng et al., 2016; Ueda et al., 2016). The mixing state of BC is a vital parameter that determines its optical and radiative properties (Moffet and Prather 2009; Liu et al., 2017) and is a critical input for the models used to estimate BC direct radiative forcing (Bond et al., 2013). The information on the nature of the coating material along with the state of mixing of BC particles gives insight into the magnitude of the mixing-induced absorption enhancement for BC (Cappa et al., 2012, 2018; Peng et al., 2016; Liu et al., 2017). Further, the coating of other soluble species on BC modifies its hygroscopicity and cloud condensation nuclei (CCN) activity (McMeeking et al., 2011; Liu et al., 2013; Laborde et al., 2013), and therefore, the mixing state alters BC-induced cloud changes and indirect radiative effects. Thus, the characterization of BC size and its mixing state is critical to reducing the uncertainties in its direct and indirect radiative effects (Jacobson 2001; Bond et al., 2013).

The BC sources are highly varying, both seasonally and spatially, over the Indian region (e.g., Kompalli et al., 2014a; Prasad et al., 2018 and references therein). Aerosol BC has an average atmospheric lifetime of about a week (Lund et al., 2018; Bond et al., 2013). It is prone to regional as well as long-range transport during its short atmospheric lifetime and found even over remote regions, such as the Polar Regions, albeit in lower concentrations (Raatikainen et al., 2015; Liu et al., 2015; Sharma et al., 2017; Zanatta et al., 2018). The alteration to BC mixing state depends on various factors, which include the BC size distribution, nature of sources, the concentration of condensable materials that BC encounters during its atmospheric lifetime, and processes such as photochemical ageing (Liu et al., 2013; Ueda et al., 2016; Miyakawa et al., 2017; Wang et al., 2018). Consequently, the nature and extent of coating on BC vary in space and time, and as such, BC in a polluted environment chemically-ages faster than in a relatively clean environment (e.g., Peng et al., 2016; Liu et al., 2010, 2019; Cappa et al.,2019). This calls for region-specific characterization of the spatio-temporal variability of the BC mixing state. This is particularly important over the South Asian region (with rapidly increasing anthropogenic activities and enhanced emissions from a variety of sources) and its outflow into the adjoining oceans (Lawrence and Lelieveld 2010; Babu et al., 2013; IPCC 2013). Aerosol BC over this region has a wide variety of sources (industrial and vehicular emissions, biomass, crop residue, and residential fuel burning) and is co-emitted with a broad spectrum of gaseous compounds which form secondary aerosol species such as sulfates, nitrates, phosphates, and secondary organic aerosols (SOA) (Gustafsson et al., 2009; Pandey et al., 2014) leading to complex mixing states of BC during its atmospheric chemical-ageing. The absorption potential of the resultant mixed-phase particles would be quite different from those of nascent BC (Lawrence and Lelieveld 2010; Srivastava and Ramachandran, 2013; Srinivas and Sarin 2014; Moorthy et al., 2016). When air masses from such complex source regions are transported to remote regions devoid of any sources of BC, the mixing state of BC may change. This is due to (a) restructuring of the BC aggregates during the transport due to different processes (Kutz and Schmidt-Ott, 1992; Weingartner et al., 1995; Slowik et al., 2007b; Pagels et al., 2009), and (b) varying nature and amounts of coating material arising due to the different atmospheric

lifetimes and microphysical processes involving different species (McFiggans et al., 2015). Therefore, the characterization of aerosol and trace species properties gained much attention over the years. Lawrence and Lelieveld (2010) have highlighted many field experiments that attempted to assess the impact of continental outflow of anthropogenic emissions from South Asia to the surrounding oceanic regions and its climate implications. The past field campaigns such as the Indian Ocean Experiment (INDOEX) during 1998-1999 (Ramanathan et al., 2001), the Integrated Campaign for Aerosols, gases, and Radiation Budget (ICARB) during March-May 2006 (phase-1); December-January 2008-2009 (phase-2) (Moorthy et al., 2008; Babu et al., 2012; Kompalli et al., 2013) have characterized regional aerosols over the northern Indian Ocean during different seasons.

However, the information on BC microphysical properties (especially its size distribution, mixing state, and extent of coating) over the northern Indian Ocean remained elusive primarily due to lack of instruments for near-real-time measurements to estimate BC size and coating on it (Kompalli et al., 2020a). A combination of analytical instruments [such as the single particle soot photometer (SP2) based on laser-induced incandescence technique for the measurements of microphysical properties of refractory BC (rBC) at a single particle level (Moteki and Kondo, 2007; Schwarz et al., 2008, 2013; Laborde et al., 2012; Liu et al., 2014), and mass spectroscopy-based aerosol chemical composition measurements (Liu et al., 2014; Gong et al., 2016) such as the aerosol mass spectrometer (AMS) (Jayne et al., 2000; Jimenez et al., 2003; Allan et al., 2003), or the aerosol chemical speciation monitor (ACSM) (Ng et al., 2011) that provide near real-time information on the possible coating substances] provides a way to address this issue (Kompalli et al., 2020a and references therein).

In this study, we report measurements of BC microphysical properties over the Southeastern Arabian Sea, the northern and equatorial Indian Ocean regions, made for the first time. The observations were carried out as a part of the third phase of the Integrated Campaign for Aerosols, gases, and Radiation Budget (ICARB) campaign during the winter season when the above mentioned oceanic regions are strongly impacted by the South Asian outflow aided by the favourable synoptic winds (Lawrence and Lelieveld 2010; Nair et al., 2020). The weak winds and absence of strong precipitation during this season are conducive for longer atmospheric lifetimes and support inter-hemispheric transport of the pollutants. The main aims of our measurements included: (i) characterization of the spatio-temporal variation of BC size distributions over the northern Indian Ocean, (ii) examination of the extent of BC transport from distinct source regions and changes to its mixing state during the transport to the ocean and (iii) quantification of the degree of coating on BC and identification of the nature of potential coating species by using concurrent chemical composition measurements during the South Asian outflow. The results of the campaign are presented, and implications are discussed here.

## 2. Experimental measurements

### 2.1 Campaign details and meteorology

Phase-3 of the 'Integrated Campaign for Aerosols, gases, and Radiation Budget' ship cruise-based experiment (referred to as the ICARB -2018 hereafter) was carried out during the winter period (16 January - 13 February 2018) along the track shown by the solid black line in Fig. 1, covering different parts of the Southeastern Arabian Sea (SEAS), the northern Indian Ocean (NIO), and the equatorial Indian Ocean (EIO), as highlighted by the different boxes about the track. More details about the experiment and sampling conditions are available in earlier publications (Gogoi et al.,2019; Nair et al., 2020; Kompalli et al., 2020b). Briefly, the measurements were made from the specially configured aerosol laboratory on the top deck of the ship, ~15 meters above the sea level, and the instruments sampled air from a community aerosol inlet set up with an upper size cut-off at 10 µm at a flow rate of 16.67 litres per minute (LPM). Membrane-based dryers were installed in the sampling lines to remove the excess moisture (to limit the sampling RH to < 40%). Proper care was taken to avoid the contamination from ship emission by aligning the bow of the ship against the wind direction, and any spurious data were removed during post-processing (as has been done in earlier such campaigns, Moorthy et al., 2008).

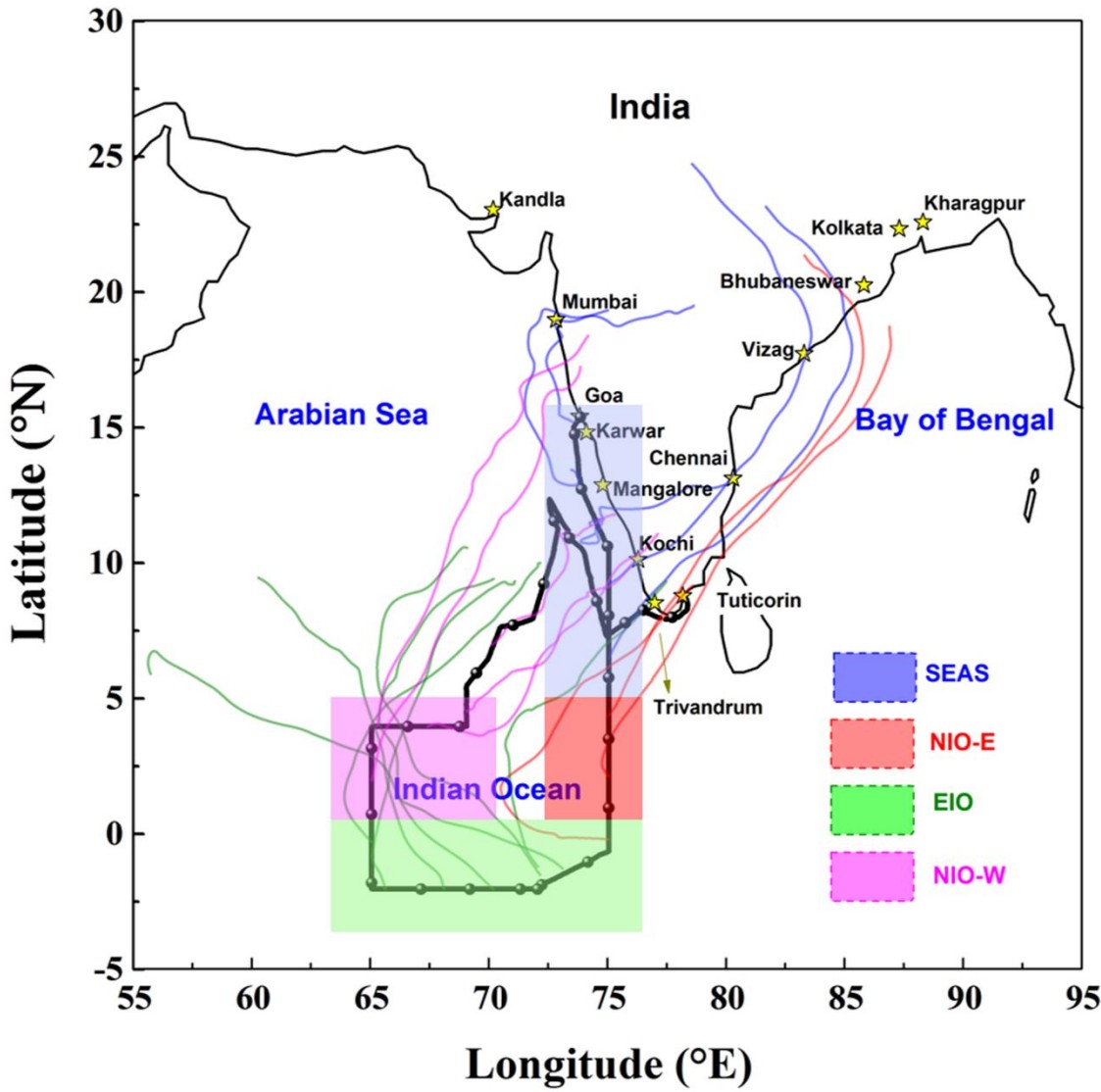

**Figure 1. Cruise track of the ICARB-2018 over the northern Indian Ocean from 16 January 2018 to 13 February 2018; the different classified sub-regions are highlighted with shaded portions, and major industrial cities and ports along with coastline are marked with a star symbol; HYSPLIT five-day isentropic airmass back trajectories arriving at 100 m amsl (dashed lines) above the ship location at 5:30 UTC on each day for different sub-regions are shown in different colours. The filled circles on the track indicate the daily mean position of the ship. SEAS: Southeast Arabian Sea; NIO-E: Northern Indian Ocean-East; EIO: Equatorial Indian Ocean; NIO-W: Northern Indian Ocean-West; Southern Arabian Sea: SAS, this region was affected by rain and weather system. The data from the SAS is not included in the overall analysis.**

South Asian region is known for its seasonally contrasting synoptic meteorology associated with the Asian monsoon and north-south excursion of the inter-tropical convergence zone (ITCZ) and monsoonal circulations (Das 1986; Asnani 1993). During the winter (December to February), calm north-easterly winds prevail over the Indian landmass, which facilitates extensive transport of continental air mass to the surrounding ocean. The synoptic conditions during the campaign period were quite similar to the climatological pattern, as revealed by Fig. S1 (panel a) in the Supplement, which shows the prevailing synoptic mean wind vectors at 925 hPa derived from ERA-Interim wind data from ECMWF (European Center for Medium range Weather Forecasting; https://apps.ecmwf.int/datasets/data/interim-full-daily/levtype=sfc/) data. The spatial distribution of fire counts over the continental landmass lying upwind of the campaign area, as derived from the Moderate Resolution Imaging Spectroradiometer (MODIS) fire radiative power (MODIS Thermal Anomalies / Fire locations, Collection 6 product

obtained from https://earthdata.nasa.gov/firms) for the period 10 January to 14 February 2018, is shown Fig. S1b. It reveals a significant number of fire events in the upwind regions. Monthly mean tropospheric $NO_2$ column abundances obtained from TROPOspheric Monitoring Instrument (TROPOMI) (http://www.temis.nl/airpollution/no2col/no2regio_tropomi.php) data, shown in the bottom panels of the same figure (for January 2018 (Fig. S1c) and February 2018 (Fig. S1d)), show significant emissions over the continental areas upwind, a part of which would be transported to the oceanic regions during the ICARB-2018. Earlier, based on the observations using the optical attenuation technique (aethalometer) over the upwind locations (Kharagpur, Bhubaneswar, Vizag, Trivandrum), Kompalli et al. (2013, 2014b) have reported that the highest equivalent black carbon (EBC; Optically measured BC) mass concentrations throughout the year are seen during the winter period (mean values ranging from~$5389 \pm 1245$ ng m$^{-3}$ over Trivandrum to $11691 \pm 4457$ ng m$^{-3}$ over Kharagpur) which highlighted the source strength during this season. Also, the east coast of India is more industrialized compared to the west coast/peninsular India (Fig.S1 c&d; Moorthy et al., 2005; Kompalli et al., 2013).

Air mass back trajectories derived using the Hybrid Single-Particle Lagrangian Integrated Trajectory (HYSPLIT) (https://www.arl.noaa.gov/hysplit/ready/) and shown in Fig. 1, highlight the potential long-range transport of the continental emissions to different oceanic regions covered during the campaign. Accordingly, the cruise region is divided into five distinct sub-regions: (i) Southeastern Arabian Sea (SEAS), which encountered direct outflow from the strong source regions in the western coastal and peninsular India region (shown with blue colour box and air mass trajectories); (ii) Northern Indian Ocean-East (NIO-E) (eastern leg of the cruise covering the NIO region), that experienced air mass from the east coast of India and the Bay of Bengal regions (red colour); (iii) Equatorial Indian Ocean (EIO), where mostly marine air masses originated/confined within the north-eastern Arabian Sea, and without any direct influence of continental outflow (green colour); (iv) Northern Indian Ocean-West (NIO-W), which experiences outflow mainly from western coastal regions of Peninsular India after considerable transit over the Sea (magenta colour); and (v) Southern Arabian Sea (SAS), the unshaded region of the track, where widespread rainfall associated with the passage of a large scale meteorological system was encountered. We have not included the data collected over the SAS region in the overall analysis of the present study, and a brief discussion about it is provided in the Supplement. During the rest of the cruise period, calm winds ($< 5$ m s$^{-1}$) and clear sky conditions prevailed with no significant variation in air temperature (mean $\sim 28 \pm 0.8$ °C) and relative humidity (mean ~$73 \pm 5\%$) conditions. Table-1 gives the details of the regional mean (Avg.), maximum (Max.) and minimum (Min.) values of meteorological variables (air temperature (AT), relative humidity (RH), wind speed (WS), wind direction (WD), total accumulated rainfall (RF) amount) for different regions covered during the ICARB-2018.

**Table-1.** Regional values of meteorological parameters observed during the cruise period.

| Region | AT (ºC) | | | RH (%) | | | WS (m s$^{-1}$) | | | WD | RF |
|---|---|---|---|---|---|---|---|---|---|---|---|
| | Avg. | Max. | Min. | Avg. | Max. | Min. | Avg. | Max. | Min. | (º) | (mm) |
| SEAS | 27.6 | 29.3 | 26.8 | 76.2 | 86.3 | 65.9 | 2.7 | 5.5 | 0.3 | NE | 0 |
| NIO-E | 28.0 | 28.7 | 27.0 | 69.0 | 76.5 | 60.1 | 3.1 | 4.8 | 2.0 | NE | 0 |
| EIO | 28.0 | 29.1 | 26.9 | 72.2 | 79.4 | 65.5 | 4.7 | 8.8 | 2.3 | NW | 0 |
| NIO-W | 28.4 | 28.9 | 27.7 | 72.7 | 78.5 | 66.1 | 4.2 | 6.2 | 1.7 | NW | 7.1 |
| SAS | 28.2 | 30.0 | 27.4 | 73.9 | 81.9 | 61.1 | 2.8 | 5.7 | 0.1 | N | 50.5 |

## 2.2 Measurements

Of the several measurements made aboard, the measurements of the BC single particle microphysical properties were carried out using a single-particle soot photometer (SP2) (Model: SP2-D; Droplet Measurement Technologies, Boulder, USA), which was operated at a flow rate of 0.08 litres min$^{-1}$. The SP2 employs a 1064 nm Nd:YAG intracavity laser, and by using a laser-

induced incandescence technique, it characterizes the physical properties of refractory BC (rBC) at the individual particle level (Moteki and Kondo, 2007; Schwarz et al., 2008, 2013; Laborde et al., 2012; Liu et al., 2014; Shiraiwa 2007; Kompalli et al., 2020a). It provides information about mass and number concentrations and size distributions of rBC. The amplitude of the incandescence signal is proportional to the rBC mass present in the BC containing particles, and the mass equivalent diameter (the diameter of a sphere containing the same mass of rBC as measured), or BC core diameter ($D_c$), is obtained from the measured rBC mass by assuming a density, $\rho \sim 1.8$ g cm$^{-3}$ for atmospheric BC (Bond and Bergstrom, 2006; Moteki and Kondo, 2007, 2010; McMeeking et al., 2011). Further, the amplitude of the scattering signal provides information about the scattering cross-section of the particles, which is used to determine the optical sizing of the particles. In the case of BC-containing particles, the scattering signal gets distorted as it passes through the laser beam because of the intense thermal heating of the particle and evaporation of the coating. Thus, the scattering signal of the BC particle is reconstructed using a leading-edge only (LEO) fitting technique, as described in earlier publications (Gao et al., 2007; Liu et al., 2010, 2014, 2017), and this scattering cross-section is matched with the modelled values in a Mie-lookup table to derive the optical diameter of a BC particle or the coated BC size ($D_p$). Here, the total particle is treated as an ideal two-component sphere with a concentric core-shell morphology, with a core (rBC) refractive index value of $2.26 - 1.26i$ (Moteki et al., 2010; Liu et al., 2014; Taylor et al., 2015) and a coating refractive index of $1.5+0i$ (which is an optimum value and in the range of refractive indices of inorganic salts (($NH_4$)$_2$$SO_4$ = 1.51; NaCl = 1.53) and secondary organic aerosol (~1.44-1.5) at $\lambda$ = 1064 nm (Schnaiter et al., 2005; Metcalf et al., 2012; Lambe et al., 2013; Laborde et al., 2013; Taylor et al., 2015). These two diameters ($D_p$ and $D_c$) are used to infer the coating thickness. Before the experiment, the SP2 was calibrated by using Aquadag® black carbon particle standards (Aqueous Deflocculated Acheson Graphite, manufactured by Acheson Inc., USA), and a correction factor of 0.75 is applied to address the difference between Aquadag® standards and ambient BC (e.g., Moteki and Kondo 2010; Laborde et al., 2012). A detailed description of the instrument, data interpretation procedures, uncertainties, and caveats involved can be found elsewhere (Liu et al., 2010,2014; McMeeking et al., 2010; Sedlacek III et al., 2012, 2018; Kompalli et al., 2020a). It is recognized that the SP2 cannot provide the details of rBC aggregate morphology or the relative position of the BC within the particle, which can be determined better through microscopy-based studies (e.g., Adachi et al., 2010; Ueda et al., 2018). However, the intensity of the incandescence signal detected by the SP2 is proportional to the refractory black carbon mass present in the particle and is independent of particle morphology and mixing state (Slowik et al., 2007a; Moteki and Kondo, 2007; Schwarz et al., 2008). Again, though the SP2 has limited detection sensitivity towards pure scatterers because of the limited size range it covers, the light scattering information at 1064 has been widely used to accurately derive the size of the coated particle (Gao et al., 2007; Moteki et al., 2010; Shiraiwa et al., 2008; 2010; Laborde et al. 2013; Taylor et al., 2015; Liu et al., 2017).

Supplementing the above, we have used the information on the mass concentration of non-refractory PM1.0 aerosols (organics, sulfate, ammonium, nitrate, and chloride) from a collocated aerosol chemical speciation monitor (ACSM; Model: 140; Aerodyne, USA) (Ng et al., 2011). The objective here is to identify the possible coating material on rBC particles. The ACSM consists of a particle sampling inlet, three vacuum chambers (differentially pumped by turbopumps, backed by the main diaphragm pump), a residual gas analyzer (RGA) mass spectrometer (Pfeiffer Vacuum GmbH). The particles are drawn to an aerodynamic lens assembly having $D_{50}$ limits (50% transmission range) of 75-650 nm and 30 to 40 % transmission efficiency at 1 μm (Liu et al., 2007)  through a 100 μm critical orifice. These particles are focused into a narrow beam and transmitted to a vacuum environment where they are flash-vaporized by the thermal capture vaporizer (Xu et al., 2017; Hu et al., 2017a, 2017b) operating at 525 °C. Subsequently, these vapors are ionized via 70 eV electron impact ionization and detected with a quadrupole mass spectrometer. The data is processed as per the prescribed methodology (Ng et al., 2011; Middlebrook et al., 2012; Kompalli et al., 2020a). We have used software provided by the manufacturer (Aerodyne Research, ACSM Local, version 1.6.0.3, within IGOR Pro version 7.0.4.1) for processing and analysis of data. Using the default fragmentation table (Allan et al., 2004), the measured fractions of unit mass resolution spectra signals were apportioned to individual aerosol species. The required corrections for the instrument performance for the varying inlet pressures and $N_2$ signal were performed

(Ng et al., 2011; Sun et al., 2012). Mass-dependent ion transmission efficiency correction of the residual gas analyzer was carried out using the signals from the internal diffuse naphthalene source (m/z 128). The calibrations of ionization efficiency (IE) and relative IE (RIE) calibrations were performed prior to the experiment by using monodisperse (300 nm) particles of $NH_4NO_3$ and $(NH_4)_2SO_4$ (Jayne et al., 2000; Allan et al., 2003; Jimenez et al., 2003; Canagaratna et al., 2007). The present ACSM consists of a capture vaporizer with an inner cavity to reduce the particle bounce (Xu et al., 2017), resulting in a higher collection efficiency (about unity) (Hu et al., 2017a; 2017b). Therefore, the composition-dependent collection efficiency correction prescribed by Middlebrook et al. (2012), applicable to standard vaporizer instruments, was not applied to our data. More than 1200 quality checked individual observations with a time resolution of ~30 minutes formed the database for this study.

Continuous measurements of the particle number size distributions in size range 10 to 414 nm have also been carried out aboard, at the 5-minute interval, using a scanning mobility particle sizer spectrometer (SMPS; TSI Inc., USA) during the campaign (Kompalli et al., 2020b). The SMPS consists of an electrostatic classifier (TSI 3080), a long differential mobility analyzer to size segregate the particles based on their electrical motilities (Wiedensohler 1988), which are subsequently counted by using a water-based condensation particle counter (TSI 3786). Concurrent measurements of the particle number size distributions in the aerodynamic diameters range 542 to 19800 nm (which can be converted to stokes diameters using an effective particle density) have also been made using the aerodynamic particle sizer (Make: TSI, Model: 3321) that works based on 'time-of-flight' technique (Leith and Peters 2003). Though the contribution from the particles in the sizes measured by the APS to the overall aerosol number concentrations is found to be < 2 %, combining both these measurements gives the total particle number concentrations covering a wide size range (10 -10000 nm). We have used the particle number size distributions (and total particle number concentrations) from 10-1000 nm from the SMPS+APS measurements, along with the number concentration of rBC from the SP2 to estimate the fraction of rBC-containing particles.

### 2.3 Analysis

The extent of coating on rBC particles is quantified in terms of the bulk relative coating thickness (RCT) defined as $D_p/D_c$, where $D_p$ and $D_c$ are coated and core BC particle diameters, respectively. It is estimated by dividing the total volume of coated BC with that of rBC cores in a given time window (5 minutes in our case) following Liu et al. (2014, 2019):

$$\frac{D_p}{D_c} = \sqrt[3]{\frac{\Sigma_i D_{p,i}^3}{\Sigma_i D_{c,i}^3}} \tag{1}$$

where $D_{p,i}$ and $D_{c,i}$ are the diameters of coated and core rBC respectively for each single particle i. In addition to RCT, we have used bulk volume-weighted absolute coating thickness (ACT, in nm), defined as $(D_p–D_c)/2$ (both $D_p$ and $D_c$ used here are volume averaged diameters) based on the assumption of a concentric core-shell morphology, as another diagnostic of the coating on the population of rBC particles (Gong et al., 2016; Cheng et al., 2018; Brooks et al., 2019). More details about the parameters bulk RCT and ACT, the methodology used here, and uncertainties associated were described elsewhere (Liu et al., 2017, 2019; Sedlacek III et al., 2018; Brooks et al., 2019; Kompalli et al., 2020a).

The means of the mass median diameter (MMD) and number median diameters (NMD) were determined from the size distributions of BC cores for each time window by least-squares fitting to an analytical monomodal log-normal distribution (Liu et al., 2010, 2014; Kompalli et al., 2020a) of the following form:

$$\frac{dX}{d\ln D_q} = \sum \frac{X_0}{\sqrt{2\pi}\ln\sigma_m} \exp\left[ -\frac{\left(\ln D_q - \ln D_m\right)^2}{2\ln\sigma_m} \right] \tag{2}$$

Here $X_0$ corresponds to mass/number concentration of the mode, $D_m$ is the mass/number median diameter, $D_q$ is particle diameter, dX is mass/number concentrations in an infinitesimal diameter interval $dlnD_q$, and $\sigma_m$ is the geometric standard deviation (of the median diameter).

Using the bulk RCT and MMD of the BC cores, the volume-weighted coated BC size ($D_{p,v}$) is calculated as below to indicate the mean coated BC size:

$$D_{p,v} = \frac{D_p}{D_c} \times MMD \qquad (3)$$

Since the ratio of the mass of non-absorbing coating material to the rBC core is an important parameter in determining the degree of absorption enhancement of BC, we quantified their mixing in terms of the bulk mixing ratio of coating mass over rBC mass ($M_{R,bulk}$) derived by assuming densities for the bulk coating ($\rho_{coating}$) and rBC core ($\rho_{rBCcore}$ ~1.8 g cm$^{-3}$) (Liu et al., 2019) as below:

$$M_{R,bulk} = \left( (\frac{D_p}{D_c})^3 - 1 \right) \times \frac{\rho_{coating}}{\rho_{rBCcore}} \qquad (4)$$

Here we have used the effective dry density ($\rho_{coating}$) of ambient NR-PM1.0 based on the measured near-real-time chemical composition (Budisulistiorini et al., 2016), by assuming densities of organics and inorganics as 1.4 (Hallquist et al., 2009) and 1.77 g cm$^{-3}$ (Park et al., 2004), respectively.

To explore the distribution of BC core-coatings, a parameter of scattering enhancement ($E_{sca}$) for each single particle is determined using the expression (Liu et al., 2014, 2019):

$$E_{sca} = \frac{S_{measured,coatedBC}}{S_{calculated,uncoatedBC}} \qquad (5)$$

where the term in the numerator is the scattered light intensity of the coated rBC particle measured by the scattering detector of the SP2 and reconstructed using the LEO technique, while the denominator is the scattering intensity of the uncoated BC calculated using Mie single particle scattering solutions assuming sphericity (Liu et al., 2014, 2019; Taylor et al., 2014; Brooks et al., 2019). For this purpose, the measured rBC mass and a refractive index of BC ~2.26 ± 1.26i (Moteki et al., 2010) at the SP2 laser wavelength, ~1064 nm, were used. For an uncoated rBC particle, $E_{sca}$ is equal to 1, and $E_{sca}$ increases with an increased coating at a given core size. Combined with rBC core diameters and coating parameters, $E_{sca}$ is helpful in identifying the nature of sources, though under the assumption that no material loss via oxidative and/or photochemistry occurs, which can either alter overall particle size and/or the refractive index of the coating.

## 3. Results and discussion

### 3.1 Spatial distribution of rBC mass/number concentrations and size distributions

The spatial variations of the number and mass concentrations of refractory BC core particles during the cruise, along with the statistics over different regions depicted using the box-and-whisker plots, are shown in Fig.2.

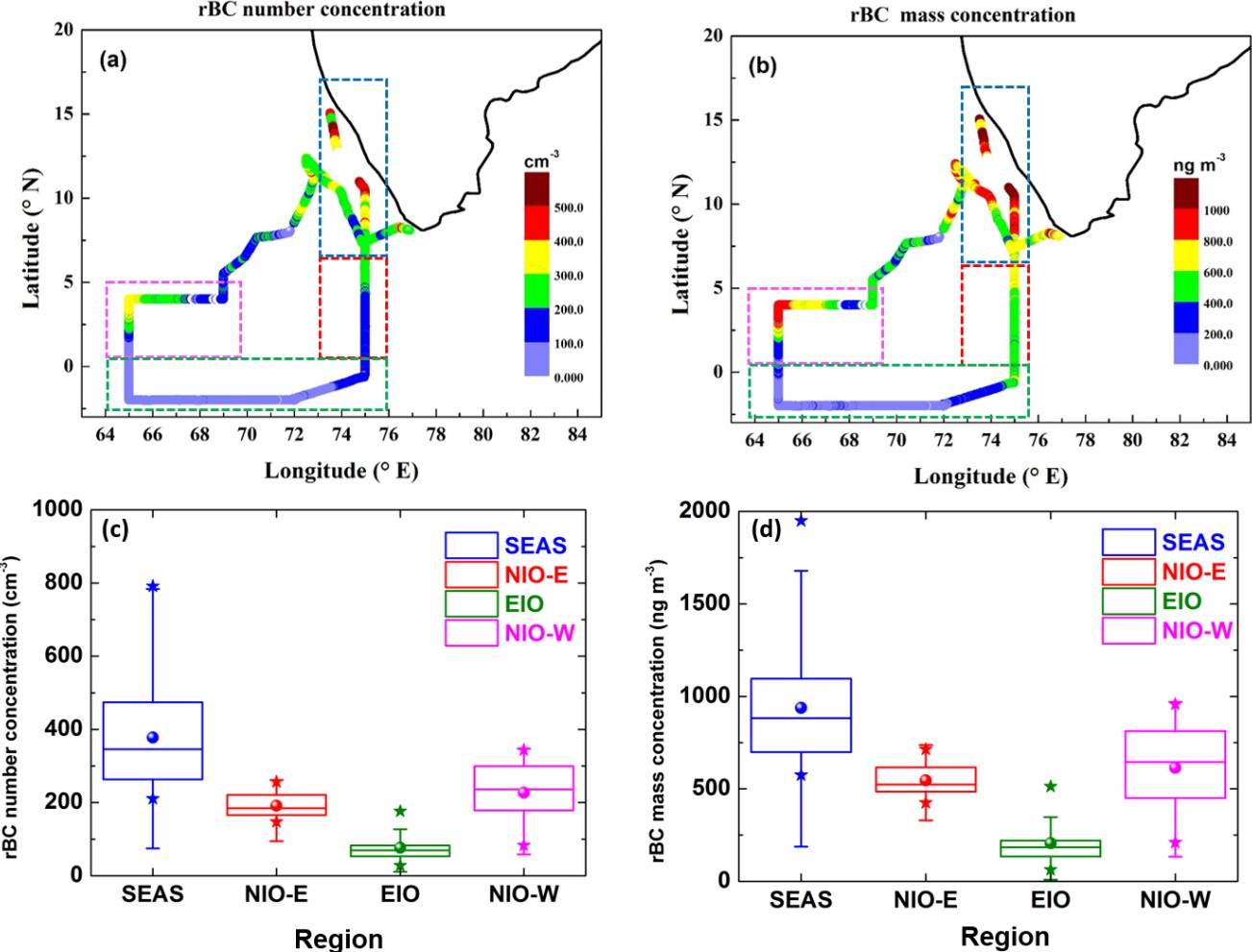

**Figure 2**. **Spatial distribution and the box-and-whisker plots of refractory BC (rBC) number (a & c) and mass concentrations (b& d). The colour scale in the spatial map (upper panels) indicates the magnitude of the property. Rectangles with dashed borders highlight different sub-regions. The box-and-whisker plots (bottom panels) illustrate the mean (sphere), median (the horizontal bar in the box), 25th and 75th percentile (the lower and upper lines of the box), 5th and 95th percentile (end of error bars) and maximum and minimum values for the regions (as solid stars).**

The highest values and variabilities (standard deviation) in the rBC mass (mean ~938 ± 293 ng m$^{-3}$) and number (~378 ± 137 cm$^{-3}$) concentrations are noticed over the SEAS region, which is in the proximity of the source regions/outflow from the western coast and peninsular India, where industrialized cities and major ports are located (Fig.1 & Fig.S1). The concentrations decreased gradually by half as the ship headed towards the NIO-E (eastern leg of the NIO), where it received outflow mostly from the east coast/ the Bay of Bengal regions. The lowest concentrations (which were 4-5 fold lower than the values seen over the SEAS) are observed over the remote EIO region. The lower concentrations (<200 ng m$^{-3}$) highlighted, the cleaner nature of this region, which encountered mostly oceanic air masses. The concentrations increased again (almost by 3-folds compared to the values seen over the EIO region) as the ship traversed to the NIO-W (western leg of the NIO) region, which experienced continental outflow air masses from the west coast/peninsular India; similar to the SEAS, but farther from the coast and SEAS region. Thus, the concentrations in the NIO-W were lower than those seen in the SEAS and comparable to those seen in its eastern counterpart (NIO-E). This also indicated varying amounts of BC in the outflow from different parts of the peninsula, apparently due to different source strengths and transit times involved. A similar spatial variability pattern was also reported for other aerosol parameters during the present campaign (Gogoi et al., 2019; Nair et al., 2020; Kompalli et al., 2020b).

The size distributions of rBC are influenced by the source, sinks, and transformation processes taking place during advection and are known to be important in assessing the light absorption characteristics (Reddington et al., 2013). The MMD values of the rBC size distributions are strongly influenced by the source of BC emissions (e.g., Ko et al., 2020; Cheng et al., 2018). Recently, Ko et al. (2020) have compared the MMD and NMD values of rBC size distributions from different dominant sources. Previous studies report that the MMD (and NMD) values over the regions dominated by fresh fossil fuel emissions are smaller (MMD ~100-178 nm and NMD ~ 30 to 80 nm) compared to the areas with dominant solid-fuel sources (biomass, biofuel, coal-burning) (MMD ~130-210 nm and NMD ~100 to 140 nm), whereas, well-aged and background BC particles in outflow regions have MMD values in between (MMD ~ 180-225 nm and NMD ~ 90-120 nm) (McMeeking et al., 2010; Liu et al., 2010, 2014; Kondo et al., 2011; Cappa et al., 2012; Sahu et al., 2012; Metcalf et al., 2012; Laborde et al., 2013; Reddington et al., 2013; Gong et al., 2016; Raatikainen et al., 2017; Krasowsky et al., 2018; Brooks et al., 2019; Kompalli et al., 2020a; Ko et al., 2020). The spatial distribution of mass median diameter and number median diameters during the ICARB-2018 shown in Fig. 3 was interpreted based on this backdrop. The observed mean NMD (0.10-0.11 µm) and MMD (0.19-0.20 µm) values over the entire study region (Fig. 3c and 3d) are within the range of values reported in earlier studies for chemically-aged continental outflow and a combination of sources. Chemical-ageing of BC is another important factor affecting the rBC core sizes owing to transformation processes (such as collapsing of the BC cores and/or due to coagulation) taking place during the long-range transport (Shiraiwa et al., 2008; Bond et al., 2013; Ko et al., 2020). Freshly produced BC particles comprise fractal-like aggregates of spherical graphitic monomers with diameters of 10-50 nanometers (Köylü et al., 1995; Bond and Bergstrom, 2006; Bond et al., 2013; Petzold et al., 2013). However, as they evolve in the atmosphere, restructuring of these aggregates occurs due to the above processes and/or condensation of vapors. Compaction can be induced by capillary forces while vapor condensation fills the voids of the aggregates (capillary condensation) (Weingartner et al., 1995; Pagels et al., 2009; Khalizov et al., 2009; Chen et al., 2018, 2016; Invanova et al., 2020 and references therein), and/or restructuring driven by surface tension forces at the solid-liquid interfaces during condensation of coating material (Kutz and Schmidt-Ott, 1992; Slowik et al., 2007b; Zhang et al., 2008; Zhang et al., 2016; Schnitzler et al., 2017). Recently, Invanova et al. (2020) have presented a detailed account of the above processes. As such, increased ageing (temporal and/or chemical) is more likely to result in compact cores (Liu et al., 2019; Laborde et al., 2013); however, the effectiveness of a condensable vapor to cause restructuring depends on its chemical composition also (Xue et al., 2009; Chen et al., 2016). The observed MMD values in this study reflected such transformation processes.

Notably, the NIO-E region depicted a slightly larger mean MMD (~0.20 µm) due to frequent larger values (35 % of the measurements showed MMD> 0.20 µm) compared to all the other regions (Fig. 3d). This is a result of the following possibilities: (i) Self-coagulation of rBC cores due to enhanced atmospheric ageing during their transport from the source regions in the east-coast to the adjacent marine regions (at the same time, sedimentation of larger particles resulting in a large reduction in number concentration and mass concentration). It may be noted that coagulation, though increases the rBC core diameters and reduces number concentrations, is a slow process. The coagulation rate depends on the square of the particle number concentrations and is the least between particles of the same size. Thus, the coagulation rates would be higher near source regions of the nascent aerosols and dropping off gradually at farther distances; (ii) the second and most important possibility is associated with the cloud processing of rBC particles. The less-soluble BC particles remain within a non-precipitating cloud as interstitial particles. A cloud undergoes multiple evaporation-condensation cycles before it transforms into a precipitating system. During such cycles, interstitial BC in cloud droplets can grow larger (especially following the evaporation of cloud droplets containing multiple rBC particles) due to agglomeration with other interstitial rBC aerosols; (iii) The third possibility is the varying nature of dominant sources. A sizeable increase in the contribution from solid fuel sources (biomass/crop residue/coal burning) in the upwind regions (the eastern coast of India) through the transported air masses can lead to larger BC cores (Brooks et al., 2019; Kompalli et al., 2020a). Interestingly, the EIO

region showed the largest variability with a non-negligible contribution (~8 %) from smaller BC cores (MMD < 0.18 µm). Over NIO-W, the MMD values remained between 0.18-0.20 µm suggesting advection of BC originating from mixed sources over peninsular India/west coast. It may be noted that the exact sources cannot be identified from the MMD value of rBC size distributions alone. More details on source apportionment are provided in section 3.3.

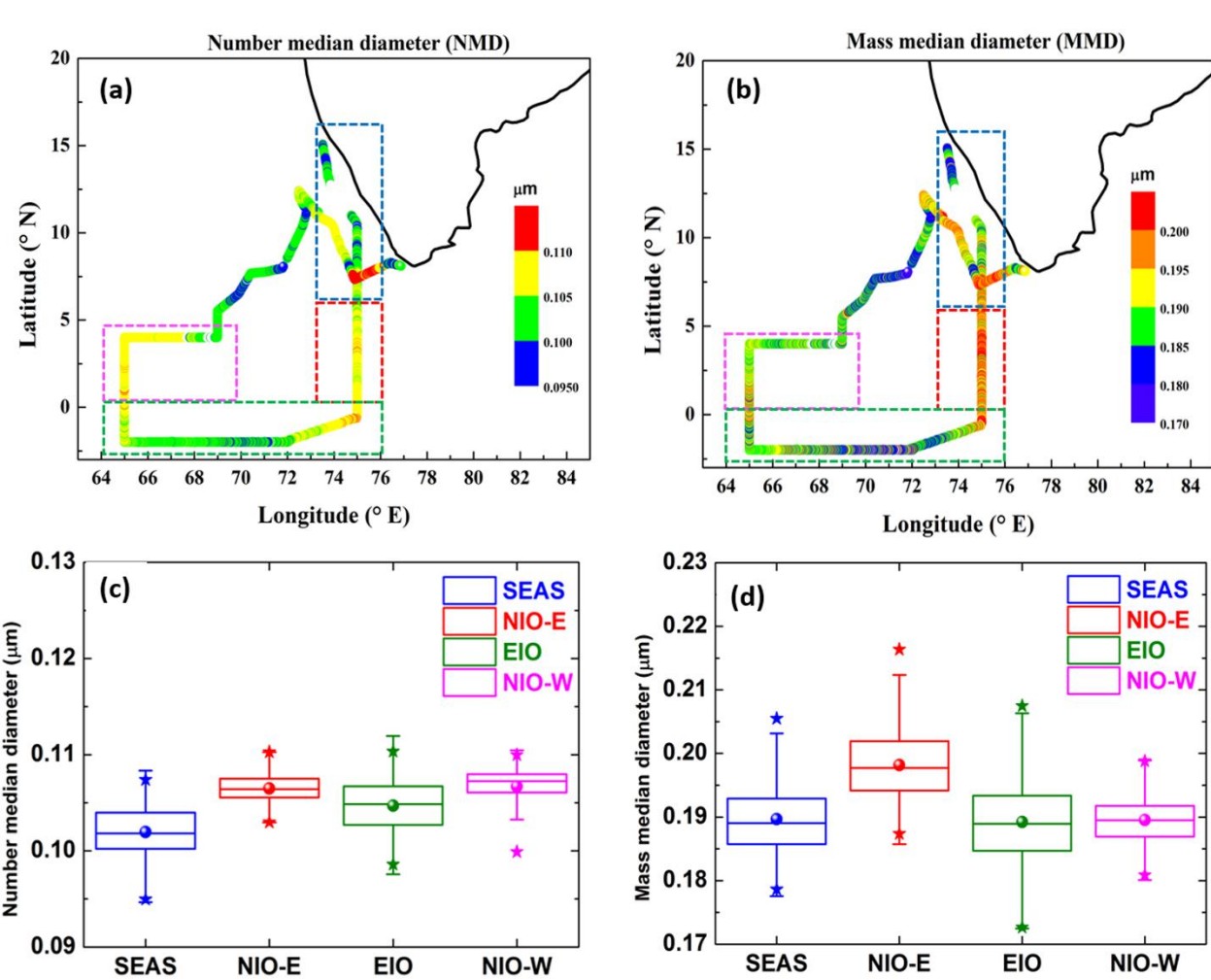

**Figure 3. Spatial distribution and the box-and-whisker plots of number median diameter (NMD) (a & c) and mass median diameter (MMD) (b & d) of rBC core size distributions during the ICARB-2018. The colour scale in the spatial map (upper panels) indicates the magnitude of the parameter. Rectangles with dashed borders highlight different sub-regions. The box-and-whisker plots (bottom panels) illustrate the mean (sphere), median (the horizontal bar in the box), 25th and 75th percentile (the lower and upper lines of the box), 5th and 95th percentile (end of error bars) and maximum and minimum values for the regions (as solid stars).**

The spatial distribution of NMD also showed a similar picture to that of MMD over all the regions. We have compared the MMD values of rBC observed in our campaign with the values reported from selected locations with distinct dominant sources in different environments in Table-2.

**Table-2**: A comparison of rBC average mass median diameters/mode of mass size distributions (MSD) reported from selected locations with distinct sources in different environments.

| S.No. | Location | Type of location/ sources | MSD mode/ MMD (µm) | Reference |
|---|---|---|---|---|
| **Aged air masses in remote/outflow regions** | | | | |
| 1. | Southeastern Arabian Sea | Continental outflow/mixed sources | 0.18-0.20 (mean~0.19± 0.01) | Present study |
| 2. | Northern Indian Ocean | Continental outflow/mixed sources | 0.19-0.21 | Present study |
| 3. | Equatorial Indian Ocean | Outflow impacted remote marine/mixed sources | 0.18 -0.21 (mean~0.19 ± 0.01) | Present study |
| 4. | Fukue Island, Japan | Asian outflow | 0.20-0.22 | Shiraiwa et al. (2008) |
| 5. | Suzu, Japan | Urban/east Asian outflow site | 0.200 | Ueda et al. (2016) |
| 6. | Mukteshwar, the Himalayas, India | High-altitude/biofuel, crop residue outflow | 0.21 ± 0.02 | Raatikainen et al. (2017) |
| 7. | Jungfraujoch, Switzerland | High-altitude remote background/biomass burning, aged BC | 0.22-0.24 | Liu et al. (2010) |
| 8. | Finnish Arctic | Remote background/aged air mass | 0.15-0.20 | Raatikainen et al. (2015) |
| 9. | Alert, Nunavut, Canada (within the Arctic Circle) | Remote background/aged air mass | 0.16-0.18 | Sharma et al. (2017) |
| 10. | Catalina Island (~ 70 km southwest of Los Angeles) | Aged air masses | 0.153-0.170 | Ko et al. (2020) |
| 11. | Atlantic Ocean | European continental outflow | 0.199 | McMeeking et al. (2010) |
| **12.** | Zeppelin, European Arctic | Remote background/aged air mass | 0.24 | Zanatta et al. (2018) |
| **Urban locations** | | | | |
| 13. | Regional average over Europe | Near source to high-altitudes | 0.17-0.21 | McMeeking et al. (2010) |
| | | (a) European continental | (a) 0.18-0.20 | |
| | | (b) Urban outflow | (b) 0.17 ± 0.01 | |
| 14. | Bhubaneswar, India | Urban/fresh urban emissions | 0.17 ± 0.01 | Kompalli et al. (2020a) |
| | | Urban/continental outflow, aged BC | 0.18-0.19 | |
| | | Urban/with high solid fuel emissions | 0.22 ± 0.01 | |
| 15. | Indo-Gangetic Plain (aircraft experiment) | Urban polluted/mixed sources | 0.18-0.21 | Brooks et al. (2019) |
| 16. | Gual Pahari, India | Urban polluted/ fresh biofuel, crop residue | 0.22 ± 0.01 | Raatikainen et al. (2017) |
| 17. | Shanghai, China | Urban/pollution episode with high biomass burning | 0.23 | Gong et al. (2016) |
| 18. | London, England | Urban/traffic emissions Wood burning | 0.119-0.124 0.170 | Liu et al. (2014) |
| 19. | Canadian oil sand mining, Canada | Urban/fresh urban emissions | 0.135-0.145 | Cheng et al. (2018) |

| 20. | Catalina Island (~ 70 km southwest of Los Angeles) | Biomass burning Fossil fuel emissions | 0.149-0.171 0.112-0.129 | Ko et al. (2020) |

As evident from Table-2, the MMD values during ICARB-2018 mostly fall in the category of BC from the continental outflow and originated from mixed sources (McMeeking et al., 2010, 2011; Ueda et al., 2016; Cheng et al., 2018). Recently, Kompalli et al. (2020a) had also reported mean MMD values of 0.18-0.19 μm over Bhubaneswar (located on the east coast of India) during the winter when urban continental outflow with mixed sources from the Indo-Gangetic Plain prevailed. Similarly, Liu et al. (2019) have reported MMD ~0.19-0.21 μm in the urban environment of Beijing with mixed sources. The mean MMD values (Table-2) and mass size distributions over different regions covered in this study (Fig. S2 in the Supplement) revealed that though the peak amplitudes varied in proportion to the magnitude of the BC loading, which decreased with increasing distance from the peninsula, the modal diameters (0.19-0.20 μm) showed little variability, which is also underlined by similar geometric standard deviation values ~1.55-1.59. This is also consistent with the widespread nature of the continental outflow to the northern Indian Ocean (from west to east) and mixed sources for rBC particles in the outflow (McMeeking et al., 2010).

### 3.2 Spatial variation of the BC aerosol mixing state

### 3.2.1 The bulk coating parameters: RCT and ACT

The variation of bulk relative coating thickness (RCT) estimated using eq.1 and absolute coating thickness (ACT) describes the physicochemical changes in the characteristics of rBC taking place during atmospheric chemical-ageing from the outflow to the oceanic regions. The spatial variation of these parameters during the cruise is shown in the top panels of Fig. 4, while the bottom panels show the frequency of occurrence of these parameters over the different oceanic regions. Corresponding median values are also written in the figures.

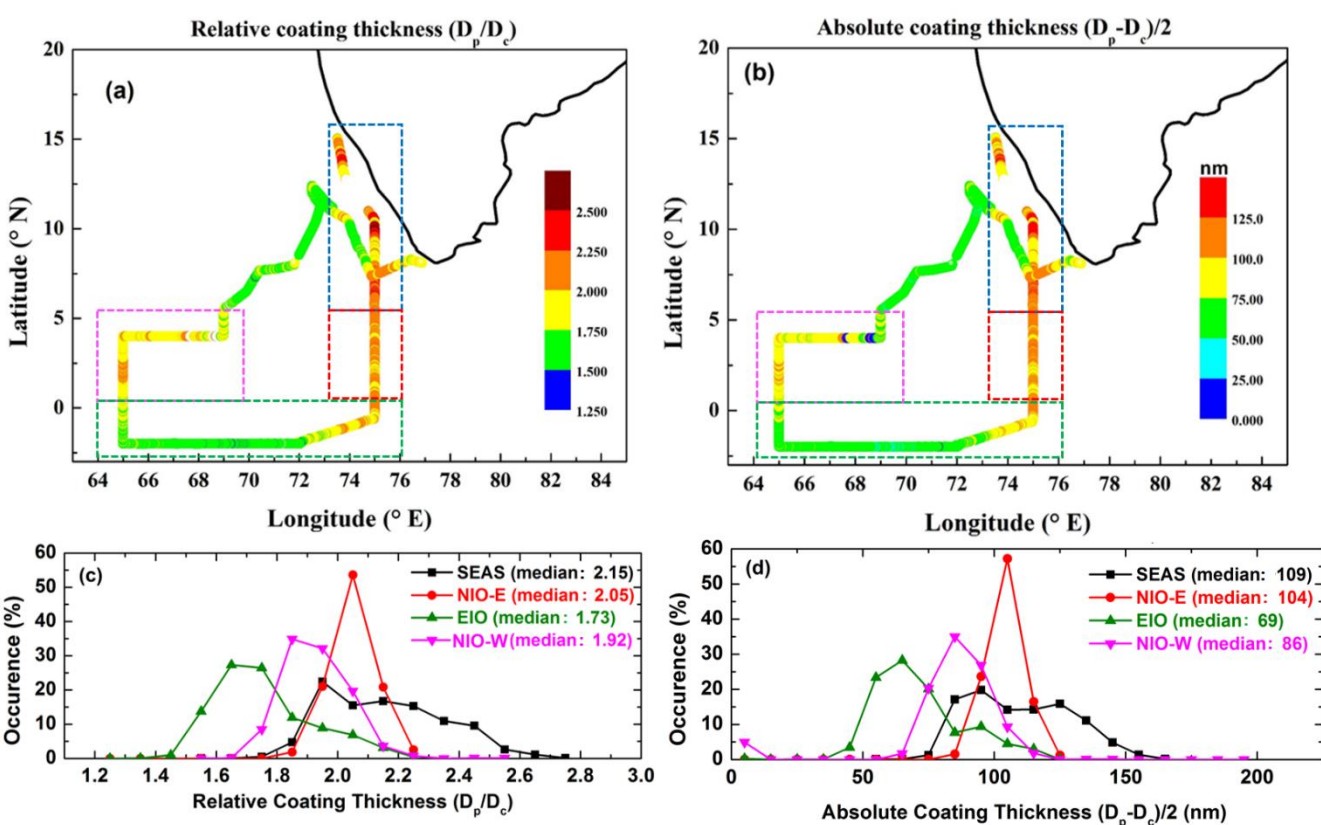

Figure 4. Spatial variation of the bulk (a) relative coating thickness ($D_P/D_c$), (b) absolute coating thickness (($D_P$-$D_c$)/2), (the colour scale indicates the magnitude) and (c-d) frequency of occurrence of the bulk RCT and ACT in different

**oceanic sub-regions (shaded following the criteria in Figure 1). Sub-regional median values are written in the bottom panels.**

The median values show a clear spatial variation of the coating thicknesses (both RCT and ACT), being highest over the SEAS (closest to the coast) and lowest over the EIO (farthest from the landmass). This is attributed to the steadily decreasing concentrations of the coating material in the outflow due to possible dispersion and reactions. The SEAS region, which is in proximity to the coast and immediately impacted by the outflow, displayed a wide range of coating values (Fig.4c & 4d), with the highest overall median values (RCT~ 2.15 and ACT~109 nm). Notably, two peaks of comparable magnitudes (RCT~1.95 and 2.3; ACT ~ 93 and 126 nm) are visible in the frequency distribution over this region (Fig. 4d), highlighting the large variability due to varying amounts of condensable species and rBC chemical-ageing. Nearly 95% of the observational points over the SEAS region indicated that rBC particles have an additional coating over their cores to the extent of > 90% of their size. Such high levels of the coating indicate the availability of high concentrations of condensable materials in the outflow, as have also been reported by other investigators elsewhere (e.g., Gong et al., 2016; Liu et al., 2019; Brooks et al., 2019). Over the NIO-E region, only thickly coated BC particles are observed, where the frequency distributions show a narrow/ sharp peak for bulk RCT (median ~2.05) and ACT (median ~104 nm). It highlighted the contrasting nature of the condensable coating material in the Bay of Bengal/east coast outflow channel compared to the west coast/ peninsular India outflow channel. Earlier, Moorthy et al. (2005) showed that the east coast /coastal Bay of Bengal has stronger hotspots of surface aerosols and gases, as well as a higher abundance of submicron aerosols. Such variability in the species concentrations in the outflow channels is responsible for the marked contrast in the coating parameters examined here.

As we move farther to the EIO region, RCT and ACT decreased conspicuously, with median values of 1.73 and 69 nm, respectively, with frequency distributions skewed towards lower values. It may be recollected that the lowest BC loading was also noticed over this region (Fig. 2), which experienced air masses that have spent considerable time in the marine atmosphere. The lower coating thickness here is attributed to dilution of the outflow and preferential scavenging processes during the advection restricting the concentrations of both the BC particles and condensable material. With atmospheric/chemical-ageing, BC particles become increasing internally mixed with condensable soluble material, which enhances their removal probability by dry deposition and in-cloud scavenging processes in the atmosphere, including both nucleation scavenging and scavenging by the pre-existing cloud droplets (Miyakawa et al., 2017; Ueda et al., 2018; Zhang et al., 2008). While the larger BC particles are scavenged rather quickly, the smaller and relatively less-coated BC particles (occasionally, even bare soot particles) can persist in the outflow and be transported to the remote marine regions (Ueda et al., 2018). As the particles spend more time in the atmosphere, they tend to gain coating material on them. Simultaneously, the loss of coating material on the particles cannot be ruled out due to photolysis or heterogeneous oxidation that can bring about fragmentation, leading to thinner coatings. Thus, preferential scavenging of larger particles leaving behind smaller and more thinly coated particles and atmospheric processes leading to loss of condensable material, explains the broad range of MMD (Fig. 3d) and lower RCT values observed over the EIO. . Furthermore, in cleaner maritime regions like the equatorial Indian Ocean, the chemical-ageing of BC occurs slowly due to reduced availability of coating material.  It possibly resulted in the observed smaller coatings on rBC over the EIO region. As the impact of continental outflow increases in the NIO-W, the coating on rBC increased once again (median RCT ~1.92 and ACT~86 nm). Interestingly, highly coated BC particles were found less frequently over the NIO-W (with west coast air masses) compared to its eastern counterpart, the NIO-E region, which experienced east coast/Bay of Bengal air masses originating from more industrialized upwind locations, e.g., Moorthy et al., 2005; Kompalli et al., 2013). Thus, a clear contrast in the mixing state parameters is evident, which is due to differences in respective coastal sources (Moorthy et al., 2008; Peng et al., 2016; Gong et al., 2016), and possible transit times over these two regions.

It is known that the BC mixing state depends on various factors, which include the BC size distribution, nature of sources, the concentration of condensable materials that BC encounters during its atmospheric lifetime, and processes such as

photochemical ageing (Liu et al., 2013; Ueda et al., 2016; Miyakawa et al., 2017; Wang et al., 2018). The values of BC coating parameters (bulk RCT and ACT) seen in the present study that examined outflow characteristics are comparable to the values reported in pollution in-plume air mass regions elsewhere (e.g., Cheng et al., 2019; Brooks et al., 2019). Recently, Kompalli et al. (2020a) have reported seasonal mean bulk relative coating thickness (RCT) in the range ~1.3-1.8 and ACT~ 50-70 nm over Bhubaneswar when the site received polluted outflow from the Indo-Gangetic Plain (IGP). Brookes et al. (2019) have noticed thickly coated BC particles (ACT~50-200 nm) across northern India, especially the IGP and north-east India, during their recent aircraft experiments. As Cheng et al. (2018) and Ko et al. (2020) have highlighted, coating parameters derived from the SP2 instruments having different system configurations (detection limits of scattering intensity and range of volume equivalent diameters covered) and different techniques used in the estimation of the optical diameters from scattering amplitudes (Metcalf et al., 2012; Gong et al., 2016; Raatikainen et al., 2017; Cheng et al., 2018; Liu et al., 2019; Ko et al., 2020) can vary considerably. This caveat needs to be borne in mind when making inter-study comparisons. Also, the earlier studies are mostly made in the 'near-field' situation, whereas the present study examined the coating characteristics in a 'far-field' scenario (far away from potential sources, especially NIO and EIO regions). The caveat here is that the present study is not a Lagrangian experiment, and it is possible that the "far-field" measurements are influenced by mixing with the surrounding environment. Nevertheless, such a high degree of coatings on BC considerably enhances its absorption cross-section, thereby, causes substantial absorption enhancement (by the factors in the range of 1.6-3.4) and affects the radiative forcing (Moffet and Prather 2009; Shiraiwa et al., 2010; Thamban et al.,2017; Liu et al., 2015; Wang et al., 2018). The implication of the observed thick coatings on BC to regional radiative forcing needs further detailed investigation in future studies.

### 3.2.2 Coated BC diameter, $F_{BC}$ and bulk mixing ratio ($M_{R,bulk}$)

The spatial variation of number concentration (in $cm^{-3}$) of non-BC (i.e., purely scattering) particles detected by the SP2 and the fraction of rBC-containing particles ($F_{BC}$; t the ratio of rBC number concentration to the total number concentration in size range 10-1000 nm from the SMPS and APS measurements) are shown in Fig. 5 (a & b). In the bottom panels of the same figure are shown the volume-weighted coated BC size ($D_{p,v}$) (in µm) (Fig. 5c) and bulk mixing ratio of coating mass to rBC mass ($M_{R,bulk}$) (Fig. 5d) calculated using the equations. 3 and 4.

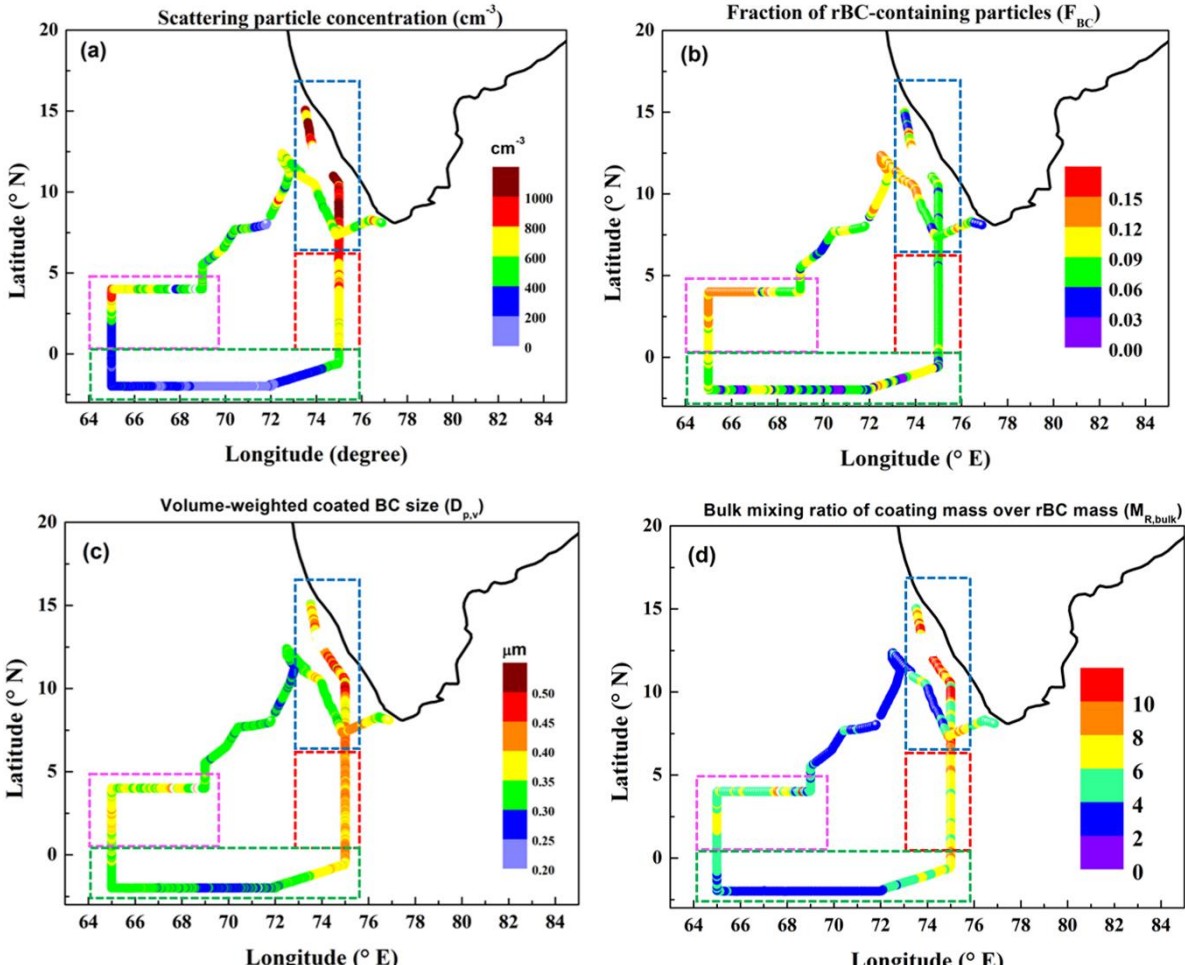

**Figure 5.** Spatial variation of the (a) scattering (non-BC) particle number concentration (in cm$^{-3}$), (b) fraction of rBC-containing particles ($F_{BC}$), (c) volume-weighted coated BC size ($D_{p,v}$) (in μm), and (d) bulk mixing ratio of coating mass to rBC mass ($M_{R,bulk}$). Rectangles with dashed borders highlight different sub-regions.

The overall spatial variation patterns of scattering particle concentrations and various mixing state parameters are similar to those of the rBC mass and number concentrations seen earlier, with the highest values over the SEAS, decreasing gradually towards the NIO (east and west) to reach the lowest values over the remote EIO. The figure reveals the following:

(i) The non-BC (scattering) particle concentrations were higher, >1000 cm$^{-3}$, in the coastal waters (the SEAS), decreasing towards farther oceanic regions and reached values as low as <200 cm$^{-3}$ in the remote EIO in-line with the expected reduction in the influence of the sources (Fig. 5a).

(ii) The rBC particles constituted about 8-12% of the total particle number concentration over different sub-regions, on average. (Fig. 5b). The $F_{BC}$ values showed the largest variability over the SEAS among all the regions (Fig. 5b).

(iii) Coated rBC particles were larger ($D_{p,v}$ ~0.35-0.50 μm) over coastal waters (SEAS), highlighting a substantial enhancement of the overall rBC particle sizes due to thick coatings on them in the polluted outflow air masses. The values diminished farther away, and the lowest values (<0.30 μm) are seen over the EIO region (Fig. 5c).

(iv) The bulk mixing ratio of coating mass over rBC mass ($M_{R,bulk}$) revealed high values (2.5-15) with large variability over the regions with extensive outflow (SEAS) due to the presence of thickly coated BC particles in these regions. Though $M_{R,bulk}$ values were very low over EIO, occasional higher values (> 4) are also seen (Fig. 5d).

Interestingly, concentrations of scattering (non-BC) particles (Fig. 5a) over the SEAS (mean ~ 973 ± 187cm$^{-3}$) and the NIO-E (mean ~747 ± 69cm$^{-3}$) are comparable to or higher than the values reported over the IGP outflow site, Bhubaneswar (winter mean ~950 ± 464 cm$^{-3}$; annual mean ~702 ± 458 cm$^{-3}$) as reported by Kompalli et al. (2020a). This highlighted the strength of

the outflow to the oceanic regions. Any increase in non-BC particle abundance impacts the fraction of rBC containing particles. The mean fraction of rBC containing particles ($F_{BC}$) that were in the range 0.08-0.12 over different sub-regions, occasionally decreased to very low values ~ 0.02 to 0.04, owing to a large influx of ultrafine particles (sizes < 100 nm) during the new particle formation events occurred due to substantial amounts of condensable vapors (Kompalli et al., 2020b). The highest fraction of rBC containing particles was seen over the NIO-W (0.12 ± 0.03) region, whereas the largest range of $F_{BC}$ values (0.03-0.21) among all the regions were observed over the SEAS. The present mean $F_{BC}$ values seen over the northern and equatorial Indian Ocean are lower than those reported over the Finnish Arctic (~0.24 across the 350 to 450 nm size range), a background site receiving aged air masses (Raatikainen et al., 2015). However, earlier studies over the continental landmass of India have shown much higher number fractions with mean $F_{BC}$ values ~ 0.51 ± 0.02 and 0.50 ± 0.03 over two stations, Gual Pahari (polluted site) and Mukteshwar (regional background site) in northern India (Raatikainen et al., 2017). This was attributed to the strong influence of regional anthropogenic activities on BC loading. In the present study, rBC particles constituted about 25% to 30% of the measured scattering particles over almost the entire oceanic region north of 5 °N, whereas they occasionally decreased to 15 to 20 % over the far oceanic regions. Kompalli et al. (2020a) had reported widely varying mean fractions (25-69% of the measured scattering particles in different seasons) over Bhubaneswar (eastern India) with the same instrument. They attributed it to the seasonal variation of the scattering (non-BC) particle population. Sharma et al. (2017) have reported 10-16% of particles containing rBC cores in the range of ~ 200–400 nm optical diameter over Alert, Canadian Arctic. The presence of lower $F_{BC}$ values over the marine regions in this study, which received a strong continental outflow, is not surprising, considering the observed large number concentration of total particles (Kompalli et al., 2020b) and the non-BC scattering particles in the detection range of the SP2 (200-400 nm) (along with rBC particles)

Strong continental outflows (from the polluted regions) are more likely to contain significant amounts of condensable material that can act as a potential coating on rBC (Liu et al., 2014, 2019; Raatikainen et al., 2017). This reflected in the observed high values of coated BC particle diameters (0.36-0.55 µm) in this study (Fig.5c). The present $D_{p,v}$ values over the northern Indian Ocean region are higher than those recently reported by Brooks et al. (2019) (~0.25-0.30 µm) over the IGP and eastern India, but comparable to the values reported by Raatikainen et al., (2017) for thickly coated BC particles in polluted outflow environments in northern India.

The higher bulk mixing ratio of coating mass over rBC mass values ($M_{R,bulk}$ ~2.5-15) (Fig. 5d) are seen over the adjacent marine regions, which is due to the presence of thickly coated BC particles. Though lower compared to other sub-regions, substantial $M_{R,bulk}$ (~ 4.24 ± 1.45) values were found even over the EIO region. Such high $M_{R,bulk}$ values were reported in the literature from extremely polluted environments and biomass burning source dominant regions (Liu et al., 2017; 2019). The presence of such non-absorbing coated mass on the rBC cores has significant radiative implications. Recently, Liu et al. (2017) have examined the measured and modeled optical properties of BC as a function of mass ratio ($M_{R,bulk}$) under different environments and found that significant absorption enhancement occurs when the coating mass over rBC mass is larger than 3. They suggested that in such a scenario (i.e., $M_{R,bulk}$ >3), the core-shell model reproduces the measured scattering cross-section.

A summary of rBC physical properties and mixing state parameters in different oceanic regions are presented in Table-3.

**Table-3:** A summary of regional mean values of rBC physical properties and mixing state parameters during the ICARB-2018. The values after ± are standard deviations.

| Parameter | SEAS | NIO-E | EIO | NIO-W |
|---|---|---|---|---|
| rBC mass concentration (ng m$^{-3}$) | 938±293 | 546 ± 80 | 206 ± 114 | 614 ± 211 |
| rBC number concentration (cm$^{-3}$) | 378 ± 137 | 191 ± 32 | 76 ± 38 | 227 ± 76 |
| Scattering particle concentration (cm$^{-3}$) | 973 ± 187 | 747 ± 69 | 262 ± 140 | 580 ± 156 |

| | | | | |
|---|---|---|---|---|
| Mass median diameter (µm) | $0.19 \pm 0.01$ | $0.20 \pm 0.01$ | $0.19 \pm 0.01$ | $0.19 \pm 0.004$ |
| Number median diameter (µm) | $0.10 \pm 0.002$ | $0.11 \pm 0.003$ | $0.11 \pm 0.003$ | $0.107 \pm 0.002$ |
| Relative coating thickness | $2.16 \pm 0.19$ | $2.05 \pm 0.07$ | $1.76 \pm 0.16$ | $1.93 \pm 0.10$ |
| Absolute coating thickness (nm) | $109 \pm 20$ | $104 \pm 7$ | $72 \pm 17$ | $85 \pm 21$ |
| Fraction of rBC-containing particles ($F_{BC}$) | $0.08 \pm 0.03$ | $0.08 \pm 0.01$ | $0.08 \pm 0.03$ | $0.12 \pm 0.03$ |
| volume-weighted coated BC size ($D_{p,v}$) (µm) | $0.41 \pm 0.04$ | $0.41 \pm 0.01$ | $0.33 \pm 0.04$ | $0.37 \pm 0.02$ |
| Bulk mixing ratio of coating mass over rBC mass ($M_{R,bulk}$) | $8.31 \pm 2.40$ | $6.91 \pm 0.71$ | $4.24 \pm 1.45$ | $5.76 \pm 1.17$ |

Table-3 highlights the spatial heterogeneity in rBC microphysical properties over the northern Indian ocean. It reveals the contrast in outflow strength with varying extents BC and non-BC species abundance in the west coast/peninsular India, east coast/Bay of Bengal air masses. Table-3 also highlights the diminishing strength of the outflow as seen from the lower concentrations, coatings and associated mixing state parameters over the EIO region.

## 3.3 BC segregation by size-resolved mixing state

The above discussions have established that:

(i)      The extent of the coating as measured by the coating thickness and the bulk mixing ratio of coating mass over rBC mass, and hence the mixing of BC with condensable species, is highest closer to the coast where the outflow is strong and decreases to farther oceanic regions.

(ii)      The rBC core diameters, as well as the fractional concentration of rBC to total concentration, remained more or less comparable throughout the oceanic regions surveyed, suggesting an impact of similar sources of mixed origin. However, the coated BC diameters varied according to the magnitude of coating over different regions.

To examine these common sources, the size-resolved BC mixing state is examined from the variation in scattering enhancement ($E_{sca}$, equation. 5) as a function of BC core diameter ($D_c$) in Fig.6. The corresponding bulk absolute coating thickness (ACT) values are mapped to the $E_{sca}$ (solid white lines) in the figure. The data collected from 16:41:24 hrs of 21-January-2018 to 18:02:46 of 22-January-2018 (Indian standard time) was used to construct the figure (which falls in the transition period between the SEAS and NIO-E regions). The same analysis repeated for a few other data sets over the other regions also yielded similar results. Following the methodology described in the previous publications (Liu et al., 2014, 2019; Brooks et al., 2019), the BC particles are segregated according to the discontinuous distribution in $E_{sca}$- $D_c$ (dashed black lines in the figure). Four classes of BC particles are described as: (a) small BC with a thin coating, i.e., with BC core diameters < 0.18 µm and ACT < 50 nm; (b) moderately coated BC, with ACT in the range 50–200 nm; (c) thickly coated BC with ACT> 200 nm; and (d) large uncoated BC, with BC core size >0.18 µm and coating thicknesses < 50 nm. In the present study, there is no noteworthy presence of a clear 'smaller sized BC with a thin coating', which generally attributed to fresh traffic emissions (e.g., Liu et al. (2014) over London; Liu et al. (2019) over Beijing). Brooks et al. (2019) also found smaller contributions from such particles during aircraft observations over northwest and northeast parts of India carried out in the dry season.

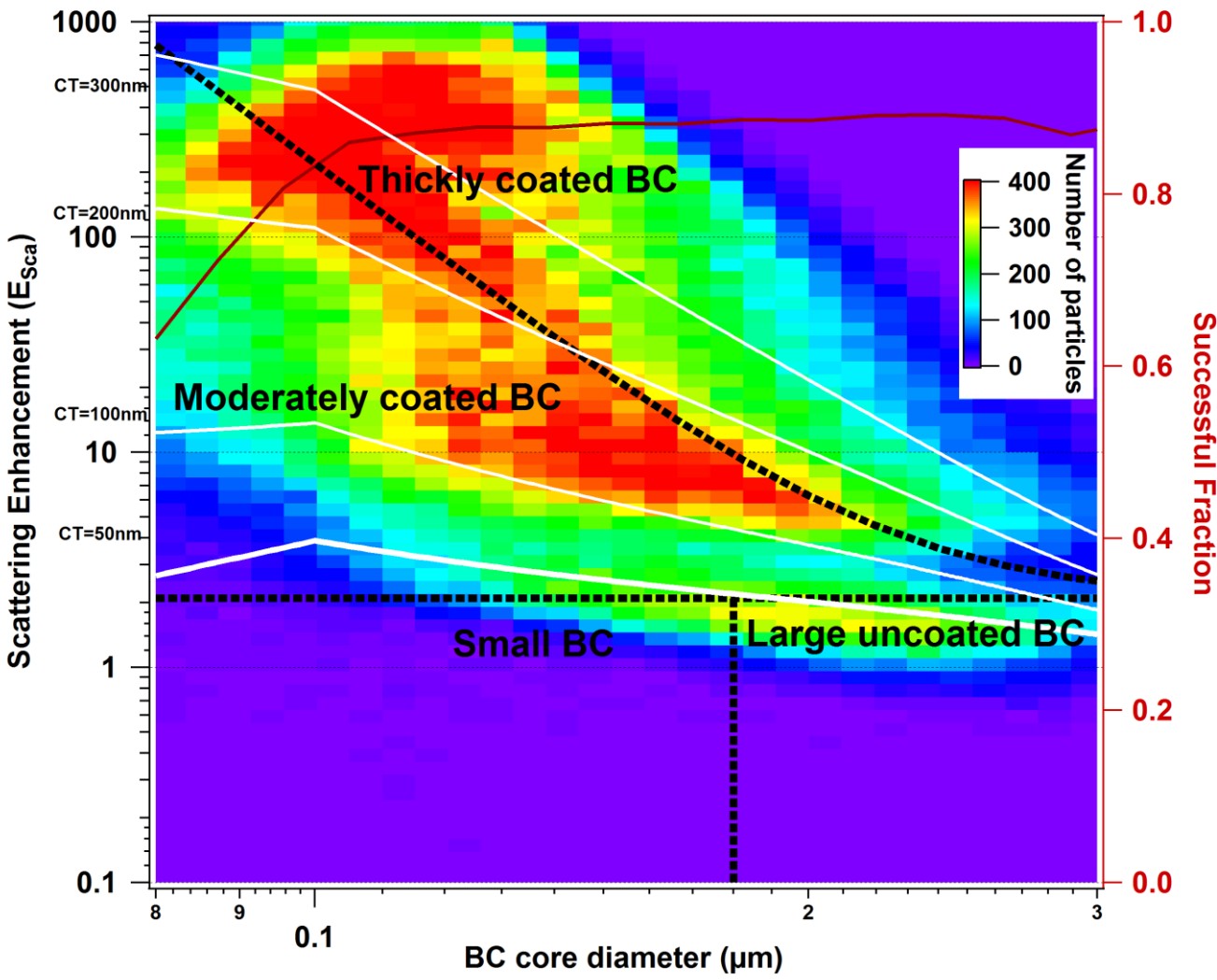

**Figure 6. Scattering enhancement ($E_{sca}$) as a function of BC core size ($D_c$) for the typical outflow airmasses during the ICARB-2018. The plot is coloured by particle number concentration. The solid brown line, with the corresponding scale on the right axis, shows the number fraction of BC particles that were successfully determined according to their scattering signal at each $D_c$ size. The image plot is a two-dimensional histogram for the detected particles. The particles are separated as four groups using the borders (from top to bottom) at y = 3.38+0.000436\*x^-5.7, y = 2.1, x = 0.18, as shown by dashed black lines on the figure (Liu et al., 2019). The solid white lines show the absolute coating thickness (ACT, nm) mapped on the $E_{sca}$- $D_c$ plot.**

The main features in the figure are:

(a) A reasonable amount of moderately coated BC particles having ACT ~50-100 nm with scattering enhancements in the range $E_{sca}$ ~2-10 were seen, and also a significant proportion of moderately coated BC particles with large scattering enhancement ($E_{sca}$ ~10-100), which increased with BC core sizes during the ICARB cruise. This highlighted the substantial contribution from a combination of mixed sources that co-emit BC and condensable material (Liu et al., 2014; 2019).

(b) Even the smaller-sized BC cores (which possibly originate from fossil fuel emissions and transform to larger cores during the atmospheric transit) were also significantly coated with resultant scattering enhancement > 100 during the ICARB-2018, which highlights the extent of chemical-ageing of BC particles in the continental outflow. Such faster chemical-ageing of smaller cores is possible in the polluted air masses (Gong et al., 2016). Similarly, faster chemical-ageing of large BC particles that generally originate from biomass burning sources

results in moderate to thicker coating (Schwarz et al., 2008; Gong et al., 2016) and amplified scattering enhancement (Liu et al., 2019) which is also seen from Fig.6.

(c) Remarkably, a greater proportion of thickly coated particles with varying BC core diameters and a wide range (5-800) of scattering enhancement values were also observed, further highlighting a strong mixed source influence of the continental outflow.

Besides the normal mode similar to the one (with core diameter <0.22 μm and coating thickness of 50–200 nm) reported from the aircraft measurements of BC mixing state over the Indian continent by Brooks et al. (2019), an additional mode of BC with BC core sizes ~110-130 nm and a significant coating thickness (> 200 nm) is seen during this study. Such a mode highlights the influence of long-range transport to the ocean from the continent on BC chemical-ageing. Though the BC mass loading decreases during the long-range transport, the remaining BC cores gain a greater coating over the ocean than over land (e.g., Moteki et al., 2007). Further, it indicates a strong secondary production of aerosol components during the transport over the ocean, contributing to the BC chemical-ageing. The much thicker coatings seen during the ICABR-2018 compared to the observations from the ground-based site (Kompalli et al., 2020a) and the aircraft measurements (Brooks et al., 2019) over the Indian region are also indicative of other sources (with poor combustion efficiencies such as biomass burning) being prevalent in this region. Gong et al. (2016) have reported thick ACT (~110–300 nm) values during a biomass burning pollution episode in urban Shanghai, comparable to the present BC mode. These values are higher than the values reported from the aircraft measurements over biomass burning plumes (~ 150 nm; Ditas et al., 2018), the southeast Atlantic Ocean ( ~ 90 nm in the boundary layer and ~120 nm in the free troposphere; Taylor et al., 2020) and aged smoke in Amazonia (55-90 nm) (Darbyshire et al., 2019). Therefore, the measurement over the ocean thus offers an opportunity to study a more chemically-aged BC from the continent outflow.

The high proportion of thick coatings on BC particles may result in significant increases in absorption by the BC. As reported by Brooks et al. (2019) from the measurements over the IGP, a significant fraction of moderately coated BC particles with increased scattering enhancement were found to have higher mass absorption coefficient values over the Indo Gangetic Plain. They attributed this to vigorous mixing between various sources due to the high amounts of secondary aerosol formation and photochemical ageing across northern India. The present scenario, with a steady pollutant outflow containing contributions from diverse sources, is akin to it. Such alterations to the mixing state of BC can contribute to significant enhancement in the absorbing characteristics of BC aerosols over the marine regions as they underdo chemical-ageing while transiting from upwind source regions in the west and east coasts of India, which may have noteworthy regional climate implications.

The large uncoated rBC particles (core diameters > 0.18 μm and thin coatings of ACT < 50 nm) with low scattering enhancements were also found during our measurements but in smaller quantities, consistent with the findings of Brooks et al. (2019) for the Indian region during the pre-monsoon and monsoon seasons. Liu et al. (2019) described that large uncoated rBC particles are indicative of coal-burning emissions and suggested that larger core BC particles with no or thin coatings display minimal $E_{sca}$ values. This can be due to significant bias introduced by the sphericity assumption used in the Mie calculations and possible difference in the refractive index of BC produced from coal-burning from the value assumed (~2.26 ± 1.26i) in the Mie calculations used to derive $E_{sca}$. However, this aspect cannot be addressed in the present study in the absence of measurements of BC morphology or source-dependent refractive index of rBC. As such, a more detailed in-situ analysis is necessary to address this.

### 3.4 Association between BC coating thickness and NR-PM1.0 chemical species

The information on the nature of the coating material along with the state of mixing of BC particles gives insight into the magnitude of the mixing-induced absorption enhancement for BC (Cappa et al., 2012, 2018; Peng et al., 2016; Liu et al., 2017). We have used the concurrent measurements of non-refractory PM1.0 (NR-PM1.0) aerosol mass and chemical composition data from the ACSM to infer the nature of coating material during the ICARB-2018. Though it may be possible for BC to

be mixed externally with coarse mode aerosols like dust or sea-salt aerosols in the real atmosphere, figure 6 directly dispels the importance of this notion.

General features of NR-PM1.0 chemical composition are shown in Fig.7, which shows the sub-regional mean mass concentrations and mass fractions (MF) of different species over the oceanic regions covered during the cruise.

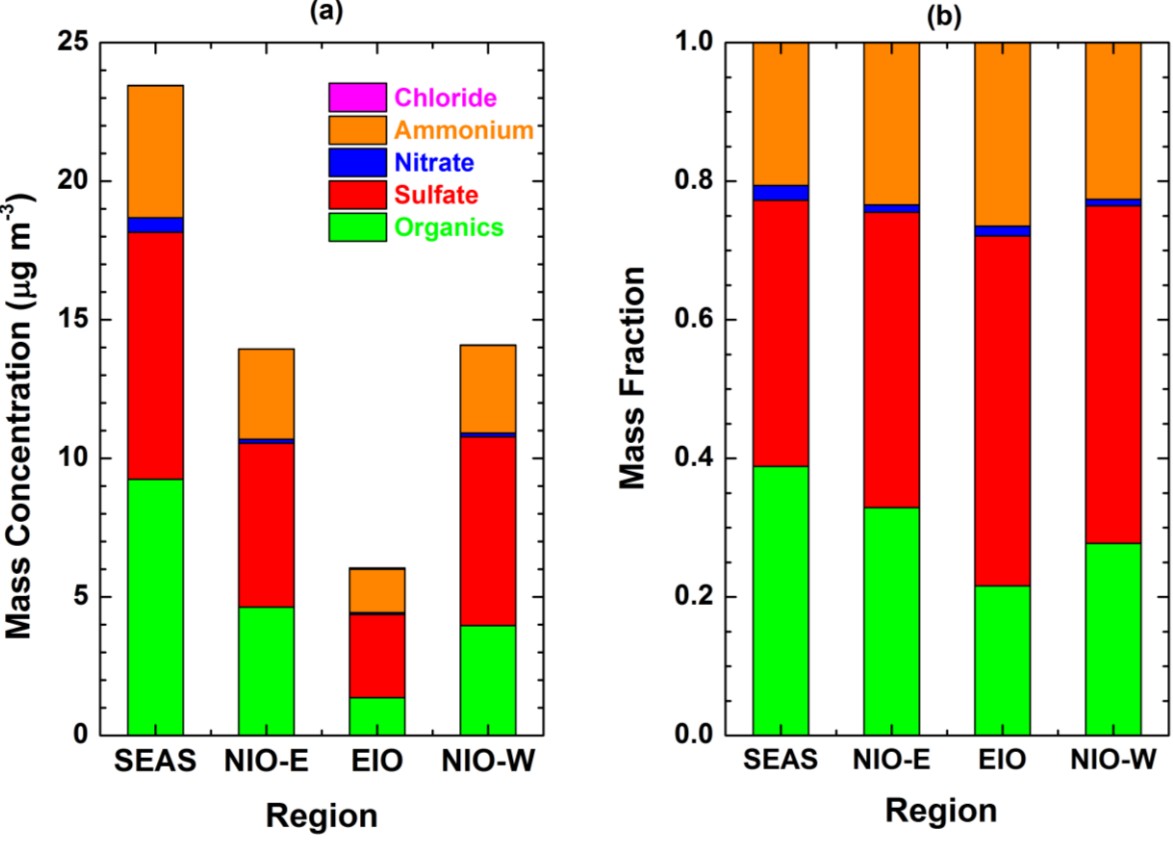

**Figure 7. (a) Mass concentration and (b) mass fraction of NR-PM1.0 chemical species, organics, sulfate, ammonium, nitrate, and chloride over different sub-regions during the ICARB-2018.**

In line with the pattern for rBC, the overall NR-PM1.0 aerosol mass concentration was also highest over the SEAS (mean ~23.44 ± 5.07 µg m$^{-3}$), decreasing steadily further away from the continent to reach the lowest concentrations (~5.48 ± 2.83 µg m$^{-3}$) over the EIO, where mostly oceanic air masses prevailed. Sulfate and organics were the two major species dominating the NR-PM1.0 composition. Of these two, sulfate dominated most of the oceanic sub-regions with a mass- fraction ≥ 0.4 except for the SEAS region, where organics (~ 0.39) were also equally important. The mass-fraction of ammonium, which is formed through gas-phase and aqueous-phase chemical reactions of NH$_3$ produced from animal wastes, fertilizers, ocean, and soil, was in the range ~0.20-0.23, while nitrate (0.01-0.02) and chloride are negligible in all the regions. To summarise, the South Asian outflow plumes consist of the more or less equal proportions of organics and sulfate aerosols in the vicinity of west coast/peninsular India, gradually changing to a sulfate-rich aerosol system in the remote oceanic regions. This marked change from the prominent presence of organics plus sulfate over coastal regions to a strong sulfate dominance over the remote oceanic regions emphasized the atmospheric processes (formation, transformation, and removal) that are at play in determining the lifecycle of these species during their transport. Possible oxidation of primary particulate organic matter due to heterogeneous reactions involving oxidants such as OH, O$_3$, and NO$_3$ during long-range transport can result in their volatilization, thus restricting their lifetime (Molina et al., 2004; Donahue et al., 2006; DeCarlo et al., 2010). However, enhanced sulfate production is possible through the gas to particle conversion in the SO$_2$ rich air masses (especially when ambient relative humidity is higher) (Unger et al., 2006; Meng et al., 2016) originated over the Indian region or through dimethyl sulfide (DMS) pathway from the marine emissions (Zorn et al., 2008; Shank et al., 2012). During long-range transport, in-situ oxidation of SO$_2$ by OH

radicals in the gas phase followed by condensation onto pre-existing particles, or reaction of S(IV) via H₂O₂ and O₃ in the aqueous phase can lead to enhanced sulfate concentrations. All such processes could have contributed to observed spatial heterogeneity in the observed organics to sulfate concentrations. Detailed investigation of these processes is not the scope of the present study.

5    Further, the association between ammonium and sulfate (Fig. S3) indicated an $NH_4^+$ deficit environment (Aswini et al., 2020). In supplementary figure S4, the ratio (expressed as %) of non-refractory coating mass on BC to the total NR-PM1.0 mass concentrations (from the ACSM) are shown. The mean ratios (varying between 23-35%) for different sub-regions are also mentioned in the figure.

   We have examined the association between bulk absolute coating thickness (inferences did not change even we use

10  RCT) and mass concentrations of NR-PM1.0 organic aerosols, sulfate, and ammonium during the campaign (Fig. 8) when ACT is low (ACT < 50 % of MMD) (Fig.8, a-c) and high (ACT > 50% of MMD) (Fig. 8 d-f). The colour bar indicates the values of the corresponding mass median diameters (MMD). The solid line is the linear least-squares fit between the variables, and the corresponding correlation coefficient (Pearson's r) is also shown in the figure.

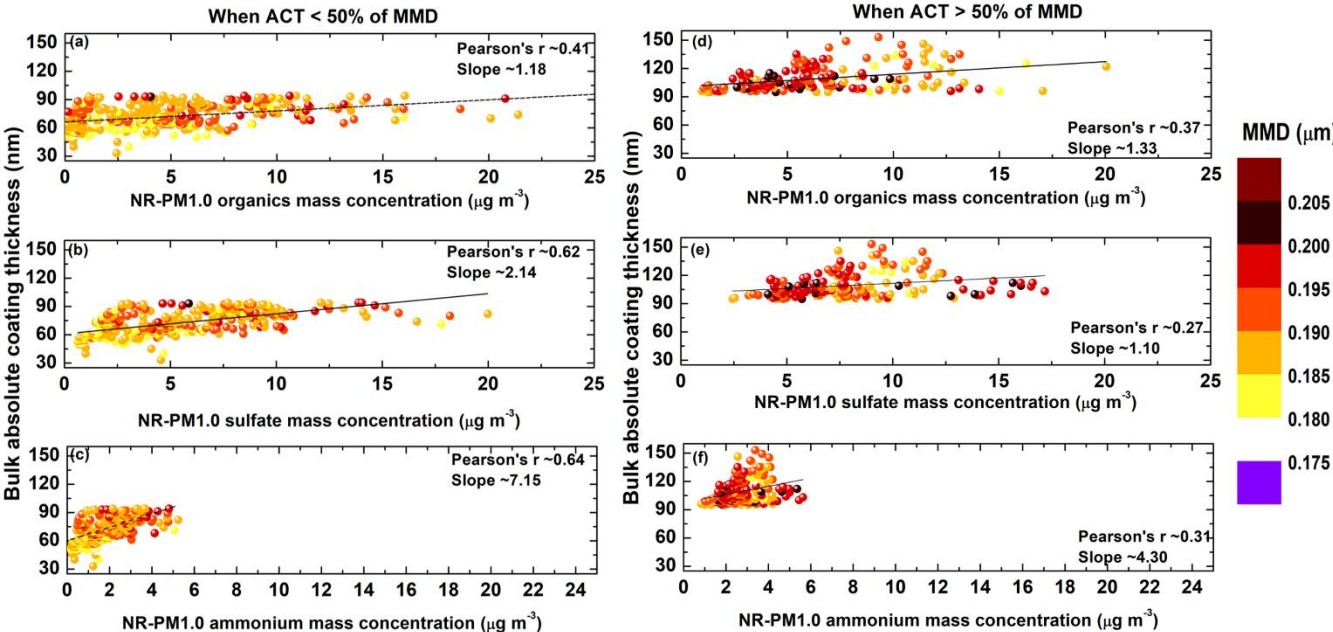

15 **Figure 8. Scatter plot between the mass concentrations of non-refractory PM1.0 organics, sulfate, and ammonium aerosols and bulk absolute coating thickness during the ICARB-2018 for the observations with low ACT (a-c) and high ACT (d-f). Colour represents the corresponding mass median diameter value. Solid lines represent the linear least-squares fit to the points. Regression slopes and correlation coefficients are written in each panel.**

Regression analysis suggested two distinct regression lines in the sulfate and ammonium plots. The bulk ACT showed a

20  reasonable association (r > 0.6; p <0.01) with NR-PM1.0 sulfate and ammonium and a weaker association with organics (r ~0.41; p <0.01) during low (ACT < 50% of MMD) observations. Corresponding slopes indicated that ACT on BC intensified more steeply with an enhanced mass concentration of sulfate (slope ~2.14) compared to organics (slope ~1.18), whereas the steepest enhancement in the ACT was noticed for ammonium (slope ~7.15). In contrast, there is no significant association between bulk ACT and mass concentrations of NR-PM1.0 species during high ACT (ACT > 50 % of MMD) observations. In

25  this case, higher ACT values prevailed for all values of sulfate (or organics) and ammonium. The low and high ACT groups of observations are distinctly separated by the corresponding MMDs values, where more frequent higher MMD values were seen for the BC population having a higher coating.

Thus, Fig. 8 suggested the possibility of complex coating on BC in the proximity of continental outflow where higher ACT (also higher MMD), significant amounts of organics, and ammonium were observed. In the far oceanic regions, sulfate

components can act as a relatively dominant coating material on BC (compared to organics). This is not unexpected considering the chemically-aged BC particles that were present in the remote marine boundary layer dominated by the sulfate aerosol system. Previously, Ueda et al. (2018) reported morphological features of soot-containing particles over the Southern Ocean and suggested that aged soot-containing particles were transformed by soluble materials derived from dimethyl sulfide (DMS) oxidation during the summer. Though this DMS oxidation pathway may not be strong for in-situ production during the winter, it may be important when the reactions during entrainment of free tropospheric air masses into remote marine atmospheric boundary layer are considered (e.g., Clarke and Kapustin, 2002). Zhang et al. (2008) suggested that soot particles acquire a large mass-fraction of sulfuric acid during atmospheric ageing. It is well known that semivolatile organic compounds are lost during atmospheric ageing due to evaporation following the dilution and oxidation of primary organics due to heterogeneous reactions (Donahue et al., 2006; Lambe et al., 2013). Such chemical-ageing (the degree of oxygenation) of organics has previously been identified by increasing $f_{44}$ and decreasing $f_{43}$, which are the ratios between the mass-to-charge ratios at 44 and 43 and the total organics signal in the component mass spectrum, respectively (Ng. et al., 2011). During this cruise, gradual increases in $f_{44}$ and decreases in $f_{43}$ were observed with distance from the coast. However, as described earlier, sulfate concentrations can increase due to the heterogeneous in-situ oxidation of $SO_2$ and condensation of sulfuric acid onto pre-existing particles after gas-phase oxidation of $SO_2$ during long-range transport (Kompalli et al., 2020a and references therein). These processes can contribute to the alteration of the mixing state of BC, which is a relatively longer-lived species, and determine its dominant coating material. Another interesting possibility is organo-sulfates acting as a coating material on BC, but examining it is beyond the scope of the present study.

The MMD values of rBC showed no particular pattern with changing NR-PM1.0 species mass concentrations. Further, the rBC mass concentrations showed a good association with organics (r ~0.78) and sulfate (r ~0.71) (not shown), corroborating the presence of multiple sources which co-emit particulate organics and sulfate along with BC aerosols. As described earlier, the mean fraction of r-BC containing particles remained similar (~8%) over different regions (SEAS, NIO-E and EIO). However, frequent new particle formation events (Kompalli et al., 2020b) have resulted in lower FBC (< 2%), which is reflected in the amounts of NR-PM1.0 mass that are bound to rBC particles. It was observed that new particle formation and subsequent growth to larger sizes preceded an enhancement in the mass concentrations of submicron aerosols (mostly organics and occasionally sulfate). This suggests another possibility of a reduction in the primary particulate pollution, and the available secondary condensate promotes nucleation in such cases. Also, the portion of the non-refractory submicron aerosol mass bound to the BC particles varied over a wide range of 10-40% over different regions (Fig. S4 in the supplement), and especially lower values were seen during NPF periods (not shown here). Figure S5 in the supplement shows the association between the sum of mass concentrations of organics, sulfate, and ammonium aerosols and bulk absolute coating thickness over different sub-regions covered during the ICARB-2018 is shown. This reveals a very good association (r~ 0.90) over the remote EIO and a weak association over the NIO-E regions, whereas no association was found over the regions SEAS and NIO-W. It also advocates that variation in the submicron aerosol composition can explain alterations to rBC mixing state over the remote regions, whereas the rBC mixing state would be complex in the vicinity of the source regions. All these suggested the presence of distinct sources and atmospheric processes involving a range of condensing gaseous species contributing to the BC coating, altering its mixing state by enhancing the extent of the coating. However, it is not possible to precisely demarcate the sources and coating substances at the individual particle level with the present methodology using the SP2 and ACSM in tandem. It can be achieved through an instrument like a soot particle aerosol mass spectrometer (SP-AMS) (e.g., Gong et al., 2016; Liu et al., 2018). Alternatively, in-situ morphological analysis techniques (like transmission electron microscope (TEM) equipped with an energy dispersive X-ray analyzer) (Adachi et al., 2010, 2014; Ueda et al., 2016, 2018) are needed for the identification of particle-level morphology.

The present study focused on the characterization of refractory BC microphysical properties, including its mixing state information for the first-time over the Southeastern Arabian Sea and the northern and equatorial Indian Ocean regions, which

were influenced by distinct outflow air masses (that originated over the east coast/Bay of Bengal, the west coast/ peninsular India and marine regions) and traversed over the oceanic regions. The sensitivity of the optical, hygroscopic properties of BC to these observed characteristics and the estimation of regional radiative and climatic implications due to such highly chemically aged, thickly coated large BC particles form the scope of future study.

## 4. Summary and Conclusions

In the present study, first-ever measurements of refractory BC microphysical properties (mass/number size distributions and mixing state parameters) over the oceanic regions adjacent to the Indian continent during periods of outflow have been carried out as part of the ICARB-2018. Major findings are the following:

(1) The rBC mass concentrations were highest over the coastal region of the Southeastern Arabian Sea (mean~938 ± 293 ng m$^{-3}$), which received continental outflow directly from the western coast/peninsular India and dropped to the lowest concentrations over the remote EIO (206 ± 114 ng m$^{-3}$), where the impact of outflow was very weak. Nevertheless, the significant concentrations (mean > 500 ng m$^{-3}$) observed over the NIO region, distant from the sources, highlighted the transport efficiency of the rBC and the widespread nature of the continental outflow.

(2) BC size distributions indicated a combination of sources and the BC core sizes corresponding to aged (temporally and/or chemically) continental outflow. Despite widely varying magnitudes of the rBC mass loading, the mass median diameter (MMD) values were in a narrow range of 0.18-0.21 µm in all the regions due to the persistent outflow.

(3) Importantly, the continental outflow from the Indian region to the adjacent oceans is characterized by thickly coated BC particles, which may have significant regional climatic implications. A great degree of coating on rBC particles and large variability of mixing state parameters was found over the SEAS region (median RCT~ 2.15, ACT ~109 nm) that is greatly impacted by short-range continental outflow. The coating parameters showed a clear east-west contrast over the northern Indian Ocean with higher coatings and a narrow distribution over the NIO-E region (RCT ~ 2.05, ACT~104 nm) where east coast/ the Bay of Bengal air masses prevailed, compared to the NIO-W (RCT ~ 1.92, ACT~86 nm) that received west coast/peninsular India air masses. Observed lower coatings on rBC over the remote EIO (RCT~1.73, ACT~ 69 nm) suggested the possible role of preferential scavenging processes removing both internally mixed, large, thickly coated BC particles and potential condensable soluble material.

(4) The average fraction of rBC containing particles ($F_{BC}$) was in the range of 0.08-0.12 over different regions, which highlighted the presence of significant non-BC particles, more so over the coastal regions. Despite similar rBC core diameters (and the fraction of rBC containing particles) due to like sources, coated BC diameters varied according to the magnitude of the coating. The highest volume-weighted coated BC size ($D_{p,v}$) values were seen over the SEAS (mean ~ 0.41 ± 0.04 µm) and the NIO-1 (mean ~ 0.41 ± 0.01 µm) reflecting the vast extent of coating on BC in the air masses impacted by pollution outflow, compared to the EIO (0.33 ± 0.04 µm) which largely received marine air masses. Further, high values (2.5-15) of the bulk mixing ratio of coating mass to rBC mass ($M_{R,bulk}$) were noticed in the outflow regions due to the presence of such thickly coated BC particles. Substantial $M_{R,bulk}$ (~ 4.24 ± 1.45) values were found even over the EIO region, which may be associated with both emissions and atmospheric processes like chemical-ageing contributing to non-refractory coatings on rBC particles.

(5) Examination of the size-resolved BC mixing state revealed, unlike the studies reported over urban regions, the absence of any notable small BC with a thin coating (MMD< 0.18 µm and ACT < 50 nm), typically associated with traffic emissions. A significant proportion of moderately coated BC particles (with bulk ACT ~50-100 nm) with two different ranges of scattering enhancement ($E_{sca}$ ~ 2-10 and ~10-100) were noticed due to the mixed nature of BC sources (a combination of solid-fuel and fossil fuel emissions) in the outflow. Importantly, a higher proportion of

thickly coated particles (ACT >200 nm) with $E_{sca}$ spanning over a wide range of 5-800 highlighted a distinct mixed source influence of the continental outflow from the Indian region to the surrounding oceanic regions.

(6) The non-refractory submicrometre aerosol chemical composition in the continental outflow is mostly dominated by sulfate, except for the coastal regions where organics were also found in significant quantities. The association between the non-refractory mass concentrations of particulate less than 1 µm in diameter (NR-PM1.0) and bulk absolute coating thickness of rBC was weak for high ACT observations (seen in the proximity of sources), suggesting complex coatings. In contrast, for the BC population with low ACT values observed over the remote ocean where sulfate was the dominant NR-PM1.0 species, a significant correlation (r ~0.62; p <0.01) was found between sulfate and rBC coating.

**Data availability**

Data are available upon request from the contact author, S. Suresh Babu (s_sureshbabu@vssc.gov.in).

**Competing interests**

The authors declare that they have no conflict of interest.

**Author contributions**

SSB and SKK conceptualized the experiment and finalized the methodology. VSN, JV, SKK, MMG, and SSB involved in the data collection onboard ship. SKK carried out the scientific analysis of the data supported by MF, DL. SKK drafted the manuscript. SSB, SKS, KKM, and HC carried out the review and editing of the manuscript.

**Acknowledgements**

The ICARB-2018 experiment was carried out under the ISRO Geosphere-Biosphere Programme. Authors acknowledge the National Centre for Polar and Ocean Research (NCPOR) of the Ministry of Earth Sciences, Government of India, for providing the shipboard facilities onboard ORV Sagar Kanya. We acknowledge the NOAA Air Resources Laboratory for the provision of the HYSPLIT transport and dispersion model and READY website (https://www.arl.noaa.gov/hysplit/ready/, last access: 20 September 2019) used in this study. We acknowledge the use of data and imagery from LANCE FIRMS operated by NASA's Earth Science Data and Information System (ESDIS) with funding provided by NASA Headquarters (http://earthdata.nasa.gov/firms, last access: 24 February 2020). S. S. Babu acknowledges the Swarna Jayanti Fellowship from Department of Science Technology, Government of India for supporting the ACSM used in the study. TROPOMI retrieved tropospheric $NO_2$ gridded data was obtained from http://www.temis.nl/airpollution/no2col/no2regio_tropomi.php., last access: 27 February 2020) ERA-Interim wind data from ECMWF (European Center for Medium range Weather Forecasting; https://apps.ecmwf.int/datasets/data/interim-full-daily/levtype=sfc/ , last access: 25 February 2020) are acknowledged. H Coe acknowledges support from NERC through grant number NE/L013886/1.

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

**Table A1.** Symbols and abbreviations

| Symbols/abbreviations | Meaning |
| --- | --- |
| ACSM | Aerosol chemical speciation monitor |
| ACT | Absolute coating thickness |
| $D_c$ | BC core diameter |
| $D_{p,v}$ | volume-weighted coated BC size |
| $E_{sca}$ | scattering enhancement |
| EIO | Equatorial Indian Ocean |
| $F_{BC}$ | Fraction of rBC containing particles |
| ICARB | Integrated Campaign for Aerosols, gases and Radiation Budget |
| IGP | Indo Gangetic Plain |
| MMD | Mass median diameter |
| $M_{R,bulk}$ | bulk mixing ratio of coating mass over rBC mass |
| NIO | Northern Indian Ocean |
| NIO-E | Northern Indian Ocean eastern leg |
| NIO-W | Northern Indian Ocean western leg |
| NMD | Number median diameter |
| NR-PM1.0 | Non-refractory PM1.0 |
| SAS | Southern Arabian Sea |
| SEAS | Southeastern Arabian Sea |
| SP2 | Single particle soot photometer |
| rBC | Refractory black carbon |
| RCT | Relative coating thickness |