# Peer review of "Mixing state of refractory black carbon aerosol in the South Asian outflow over the northern Indian Ocean during winter"

_Atmospheric Chemistry and Physics, 2020_

## Referee Comment (RC1) · Anonymous Referee #1 · 8 Dec 2020

The manuscript details the analysis of SP2 data collected aboard a research ship as part of the ICARB-18 campaign with a focus on the microphysical properties of refractory black carbon (rBC)-containing particles. As the authors point out, the SP2 detects individual rBC particles through laser-induced incandescence thereby providing mass loadings and size/mass distributions. By combining the incandescence channel with the co-located SP2 scattering channel, the mixing state of individual rBC particles can also be probed. Their analysis reveals that outflow regions are characterized by higher rBC loadings and more thickly coated rBC-containing particles with a decrease in coating thickness in the oceanic regions. Studies such as this are essential in filling in measurement and information gaps in the South Asian region and thus are important for improving our quantitative understanding of the radiative forcing impacts of BC aerosols in this region. Despite the over use of very long sentences, the manuscript is clearly written. The manuscript shares a lot similarity - with a couple of notable exceptions - to that published by Brooks et al., who conducted an aircraft-based study of northern India.

This Reviewer has a concern that the authors are, in some instances, over interpreting data collected by a single instrument and, in doing so, drawing potentially incomplete (or erroneous) conclusions - as exemplified by their use of rBC number fraction (which is discussed below). Therefore, it is recommended that the manuscript be revised as per comments below and resubmitted. To be clear, the findings contained in this manuscript are of good value and need to be published, but in its current state, the manuscript is not quite ready.

As alluded to above, 90% of this manuscript - based on rough estimate of pages dedicated to measurement type - involves only data collected by the single particle soot photometer (SP2). (The remaining 10% is on aerosol composition via an ACSM.) While the SP2 in an incredibly sensitive instrument towards rBC particles and its mixing state, it is not sensitive to changes in rBC aggregate morphology - that is rearrangement of a fractal aggregate to one that is more collapsed - and it suffers from limited detection sensitivity towards pure scatterers (i.e., non-BC particles).

First a comment on rBC morphology. For a given mass, the SP2 will report the same incandescence intensity independent of the rBC particle fractal dimension. Therefore a statement such as "The observed narrow range of mean NMD (0.10-0.11 $\mu$m) and MMD (0.19- 0.20 $\mu$m) over the entire region (Fig. 3c and 3d) reveals that the prevailing BC over the entire study region has a mixed-source origin and the particles are aged during the long advection leading to transformation processes (such as collapsing of the BC cores)" is very misleading [page 10, lines 11-12]. The authors again put forth a similar line of reasoning in the conclusion [page 23, lines 4-5] "BC size distributions indicated highly mixed BC sources, likely including a combination of fossil fuel and solid

fuel sources, along with the restructuring of the cores of the aged BC particles." The authors present no supporting data that there is any restructuring of the cores and thus cannot conclude that such processes are taking place. Indeed, there is lab studies which suggest that aggregate restructuring occurs through capillary condensation at the interstitial locations between primary particles and thus such restructuring would occur very early in the coating process. The authors are encouraged to review the work by the Khalizov group (e.g., Invanova et al., AST 2020 and references therein).

As highlighted above, this reviewer has a major issue with the authors operational definition of the "fraction of particles containing rBC" (which the authors define as the sum of the number concentration of BC and non-BC scattering particles detected by the SP2). Using the SP2 to measure the number concentration of non-BC will severely bias the derived number fraction of rBC particles because the scattering channel has a severely limited detection range (i.e., $\sim$200 nm - 400 nm). In contrast, the incandescence channel has a nominal detection sensitivity range, in terms of a mass equivalent diameter, from about 80 nm to $\sim$ 500 nm. In short, the SP2 severely undercounts pure scattering particles and therefore deriving a rBC number fraction using the SP2 scattering data will bias the rBC number fractions high and could very well mask the impact(s) that <200 nm non-BC particles might exert in differing regions. One would think (hope) that the instrument suite deployed aboard the research ship during the campaign had a CPC measurement, which would provide an excellent measurement of aerosol number concentration for particles ranging in diameter from $\sim$ 10 nm to 1000 nm, thereby greatly improving the utility of examining the rBC number fraction. Given the presence of higher anthropogenic emissions nearest the coast, it is not unreasonable to expect sub-200 nm pure scattering particles to be abundant. If no CPC, or equivalent measurement, is available then the discussion of variation in fraction of particles containing rBC is meaningless and thus should removed.

The authors report the observation of exceptionally thickly coated particles. Brooks et al., from which the current manuscript is nominally modeled after, reported on the rBC

mixing state in northern India during the pre-monsoon and monsoon seasons. Using their Figure 6 "pre-monsoon (IGP)" plot it is estimated that a 130 nm BC core had a nominal coating thickness is ∼110 nm. A similar back-of-the-envelope estimate from the current manuscript's Figure 6 suggests a coating thickness of ∼250 nm (the authors are strongly encouraged to adjust the scaling of the color mapping in Figure 6 so that more variation in the coating distribution might be seen). This difference in coating thicknesses for a 130 nm core represents about a 4.5x difference in coating volume. This is a big difference and as such, this Reviewer finds it a bit of stretch for the authors to simply state that "...the present study.....are comparable to the values reported in pollution in-plume air mass regions elsewhere (e.g., Cheng et al., 2019; Brooks et al., 2019)." [page 14, lines 19-20] Given this large difference, surly the authors could offer some discussion on why such a difference exists; that is, why are their rBC coatings so much thicker? Indeed, their reported coating thicknesses exceed what is observed for wildfires, which are known to rapidly create thickly coated rBC particles. This is perplexing and interesting finding that begs the question of why is there such a big difference, yet, no further discussion is provided by the authors beyond a cursory comparison with Brooks. The authors are encouraged to think about this finding and perhaps provide a more detailed discussion to help explain why the coating thicknesses found in this study are some much greater than that found by Brooks and in wildfires.

The authors report the organic contribution to the coating as being responsible for ∼ 40% near the coast and in the remote ocean, accounting for a little over 20%, as inferred from the ACSM. What is the influence of having a mixture of sulfate and organics on the refractive index assumed in the coating calculation? Additionally, while a density of 1.7 g/cc is reasonable for sulfates, it is high for organics (1.4 g/cc). Perhaps authors could conduct some sensitivity calculations to see if these factors exert influence on the derived coating thicknesses.

The authors are sometimes using the term "aging" in a very confusing and, potentially, misleading way. For example, the authors write [page 14, lines 4-7] "While the larger

BC particles are scavenged rather quickly, the smaller, less-aged, and relatively less-coated BC particles (occasionally, even bare soot particles) can persist in the outflow and be transported to the remote marine regions." This is misleading - more thinly coated BC particles are not necessarily "less-aged". Particles can undergo photolysis or heterogenous oxidation that can bring about fragmentation leading to material loss as particles age leading to thinner coatings or larger diameter particles can be preferentially scavenged leaving behind smaller and more thinly coated particle. So to label a smaller, less-coated BC particle - even bare soot particles - as less-aged is wrong. Correct this.

Do the authors have access to optical property measurements, specifically aerosol light absorption? If they do, they are strongly encouraged to roll that into this study so as to make this a more complete analysis.

Finally, it seems to this reviewer that the authors have additional data mining that they could do. Given their bulk coating mass-to-core ratio and the mass loadings of non-refractory aerosols via the ACSM, it might prove very interesting to examine the ratio of non-refractory coating to overall non-refractory aerosol mass. Using the bulk mixing ratio of coating mass over rBC mass (Table 3) along with the rBC mass for the SEAS leg (Table 3) and the reported NR-PM mass in Figure 7, a back-of-the-envelope calculation suggests that about ∼40% of the NR-PM mass is bound to BC particles, whereas for the EIO leg this ratio drops to ∼ 15%. This suggests the preferential loss of coating and also suggests that the rBC number fraction of particles does change significantly (as per this Reviewer's comments above on this subject).

Specific issues.

Page 3, line 5. "Produced by the incomplete (low-temperature) combustion of hydrocarbon fuels..." "Low temperature" is subjective, please quantify or remove.

Page 3, line 20. "The sources of BC are highly heterogeneous." What do the authors mean by "heterogenous"? There are three sources of BC: fossil fuel, biomass burning (wildfires and agricultural burns), and biofuel combustion. Please clarify what you mean by heterogeneous.

Page 3, line 20. "It has a long atmospheric lifetime." Compared to what? Certainly not $CO_2$, which has a nominal lifetime of 100 years. The authors are encouraged to read the paper by Lund et al., (npj Climate and Atmospheric Science (2018)1:31) who report nominal BC lifetimes < 6 days. Please clarify what you mean by long atmospheric lifetime.

Page 8, lines 20-21. "….Esca is helpful in identifying the nature of sources." This is only true under the assumption of no material loss via oxidative and/or photochemistry that could either alter overall particle size and/or the refractive index of the coating.

Page 9, lines 13 and 15. "folds" should be singular. Please change.

Page 10, lines 12-18. The authors write "….the NIO-E region depicted slightly larger mean MMD ($\sim$0.20 $\mu$m) due to frequent larger values (35 % of the measurements showed MMD> 0.20 $\mu$m) compared to all the other regions (Fig. 3d). This is a result of either of two possibilities: (i) Self-coagulation of rBC cores due to enhanced atmospheric aging, which increases the rBC core diameters (at the same time, sedimentation of larger particles resulting in a large reduction in number concentration and mass concentration); (ii) The second and less likely possibility of a sizeable contribution (though not dominant) from solid fuel sources (biomass/crop residue/coal burning) in the upwind regions to the observed BC concentrations which were transported by the air masses traversing through the eastern coast of India and the Bay of Bengal" In figure 2c, the nominal rBC number concentration is $\sim$200 /cc. Since coagulation goes a N^2, such a low concentration indicates that the rate of coagulation will be very slow. The authors should do a calculation to ascertain whether such low concentrations can result in enough self-coagulated rBC particles within the time frame of there measurements to account for $\sim$ 35% of the measurements in this region showing MMD> 0.2 um. Also, this Reviewer is quite surprised that there is very little discussion about

the possible impacts of cloud processing of rBC particles that this might have on this observation (not to be confused with scavenging).

Page 15, lines 14-17. Do the authors have access to any independent measurements of size distributions beyond solely relying on the SP2? As pointed out above, independent measurements of the microphysical properties would help in buttressing interpretations and, in some cases (e.g., rBC number fraction) enable a robust analysis to be performed.

---

## Referee Comment (RC2) · Anonymous Referee #2 · 14 Dec 2020

The information on various characteristics of rBC components of particles over the Indian Ocean is important. Overall, the presentation is clear, but the paper is longer than necessary for the information presented. The introduction is nicely written, but the discussion of results is verbose in many spots and should be written more succinctly. My major technical concerns relate to the use of the scattering signal from the SP2 (see specific comments on page 15 and beyond) and the non-refractory material contributing to the BC coatings. I hope the authors find these comments helpful.

Page 3:

1. Line 20 – "It has a long lifetime." This sentence is a bit abrupt and 'long lifetime'

needs to be defined.

2. Line 21 - What is it about sources of BC in a clean environment that reduce its relative aging? Are BC emissions unaccompanied by fewer other emitted components?

3. Lines 30-32 – "When air masses from such complex source regions are transported to remote regions devoid of any sources of BC, the aging becomes important, and the abundance of distinct species with varying lifetimes in the atmosphere differs significantly." This statement needs some clarification: 1) why and where does the aging process suddenly become important; 2) what does the abundance of distinct species and their lifetimes have to do with BC?

4. Lines 35-40 – This sentence is much longer than needed, especially since it does not tell us anything about the subject.

Page 4:

5. Line 5 – Everything but the AMS is referenced. Why not the AMS?

6. Line 8 – "such" is redundant.

7. Line 15 – There is no verb. Perhaps, "what are the sources of BC and how does its mixing state evolve during transport to and over the ocean".

8. Line 28 - Do you know that particles smaller than 10 um were efficiently sampled or are you just assuming they were?

Page 6:

9. Line 7 – In recent years, the terminology for this is commonly "equivalent black carbon" or EBC (Petzold et al., Recommendations for reporting "black carbon" measurements, Atmos. Chem. Phys., 13, 8356- 8379, doi:10.5194/acp-13-8365-2013, 2013.). Your oBC may be reasonable, but there is no reason to introduce your new definition, unless there has been some more recent change in terminology that I am unaware of.

[Figure]

Page 7:

10. Lines 19-20 – It is at best questionable as to whether the ACSM truly measures PM1.0. It needs a reference.

11. Line 24 – Why are so many details of the SP2 given, yet you are unwilling to give the "prescribed methodology"? The prescribed methodology is more germane to your analysis than the details of the ACSM that have been known for over a decade.

12. Line 37 – Page 8, line 3 – Why fit a log-normal distribution to the data when you can calculate MMD and NMD directly from the measurements? Are you sure the data always follow one log-normal mode?

Page 9:

13. Line 7 – What does the error indicate: experimental uncertainty in the mean, 25th percentile or standard deviation, etc.?

14. Lines 9-11 – If you are going to suddenly toss in oBC (or EBC) measurements, then you need to discuss why they are about a factor of two higher than the rBC.

15. Lines 14-15 – Concentrations of rBC reaching 200 ng/m3 would not be defined as extremely low in other parts of the world. In the Arctic, for example, such concentrations, which can be present in the Arctic Haze, are considered high. Please replace with something like "The lower concentrations. . ."

16. Lines 18-21 – Since the trajectories suggest that NIO-W and NIO-E are roughly equidistant from the sources, does this mean that the source strengths on the west and east coasts are similar, and that the reduction from the SEAS region is mostly due to dispersion during transport?

Page 10:

17. Lines 2-5 – References needed.

18. Line 5 – Do you mean "have", rather than "comprised BC particles"?

19. Lines 8-12 – Are you saying here that you can identify the sources contributing to the rBC based on the MMD of the rBC component of the particle?

20. Lines 11-12 – Your MMD fall in line with the "aged continental outflow" you provide 11 references for. You refer to the continental outflow studies by saying "On the other hand", which suggests that they are different from a combination of urban emissions and bio+coal emissions. Are you making your determination on lines 11-12 based on the 11 studies you reference, or are you assuming that your results are a combination of urban emissions and the bio+coal emissions?

21. Lines 13-18 – You suggest that the slight increase in MMD of the rBC over the NIO-E region is due to a complex set of processes (self-coagulation and sedimentation) occurring in those particle populations rather than differences in source types, even though you have already identified these regions as a mix of source types. Also, you give no reason why the same processes (self-coagulation and sedimentation) do not similarly affect the rBC in the other regions, in particular the NIO-W regions where the trajectories appear to have similar transport times. Why are source differences "less likely", given the large number of sources spread out over 1000-2000 km on both coastlines (east and west)?

22. Lines 19-20 – What do the lower MMD associated with EIO imply?

23. For the above reasons (comments 16-21), this section, from lines 1-21, needs to be written with more clarity and justification.

Page 11-12:

24. Table 2 and Lines 1-11, page 12 – There are measurements from the Arctic that could be added to this table. For example, Sharma et al. (ACP, 2017) found rBC MMD greater than 300 nm at a high Arctic location.

Page 13:

25. Line 8 – When you say dispersion, do you mean mixing with other air masses with smaller coating thicknesses? Reactions, unless leading to fragmentation and volatilization, are more likely to increase coating thickness than decrease it. Although there was relatively little precipitation during your study, transport across the SAS region might have impacted coatings. How might precipitation have affected the rBC size distribution and coating thickness over NIO-W? (I see you discuss this later with respect to the EIO region.)

26. Line 12 – "Nearly 95% of the SEAS measurements. . ."?

27. Line 15 – What are you contrasting with? Just start with "Over the NIO-E region. . ."

28. Line 19 – Does that mean that sources on the east coast are stronger than on the west coast. This relates back to my comment 15 above.

Page 14:

29. Line 9 – Smaller rather than inferior.

30. Lines 13-15 – It would be clearer to say "due to differences in respective coastal sources (references) and possible transit times." What are the relative differences in air mass transit times from the west coast to NIO-W and from the east coast to NIO-E, and what are the differences in east coast and west coast source strengths? See comment 20 above.

31. Lines 16-18 – Repetitious.

32. Lines 24-26 – If comparisons cannot be made, then what is the point of any of the measurements?

33. Lines 27-29 – With the caveat that your study is not a Lagrangian experiment, and your "far-field" measurements are influenced by mixing with the surrounding environment.

34. Line 32 - What is "It" that needs further investigation: coatings, enhancement,

radiative forcing?

Page 15:

35. Line 8 – What does "very high" reference to?

36. Lines 11-13 – I question your statement that rBC particles constituted about 25% to 30% of the total number concentration. Does the scattering particle concentration measured using the SP2 represent the total number concentration, and does the use of the scattering particles from the SP2 inflate the fractional value? For this fractional estimate, do you only include rBC components that have a detectable scattering signal? If so, then you could say that rBC particles constituted about 25% to 30% of the measured scattering particles. If not, then your fractional estimates are more difficult to interpret. Please correct here and in Table 3.

37. Line 19 – "highER"

38. Lines 20–21 – Is the lower limit of detection of scattering particles the same in this study as it was in the Kompalli et al (2020a) study?

Page 16:

39. Lines 3-14 – Above (see my comment 32), you say that comparisons of coating thickness cannot be made, yet here you compare fractional estimates from a few studies with no consideration given to potential differences in the scattering estimates from the studies. Please correct.

40. Line 20 – Terminate this sentence at India. You are only measuring a tiny bit of air relative to a large continent. Your results suggest there are similarities between the measurements. There may be something more to that, but you can't say the strength of emissions flowing from the north and south are similar based on this little bit of information. If you had extensive measurements from an airborne platform as well as your ship platform, you might be able to draw some inferences.

[Figure]

41. Lines 22-33 – The first and last sentences of this paragraph are repetitive. The mixing ratio is just another way of representing coating thickness, and those processes have been discussed. The reference to Liu is useful, but that is all that is needed here.

Page 17:

42. Lines 10-25 and Figure 6 – An interesting figure, but please indicate what data (where and when) are used in the construction of Figure 6. The last sentence (lines 23-24) is another case of repetition and too many words.

43. Line 25 – I suggest writing this sentence as "The high proportion of thick coatings on BC particles may result in significant increases in absorption by the BC."

Page 19:

44. Lines 8-11 – Re-write as "The large uncoated rBC particles (core diameters > 0.18 $\mu$m and thin coatings of ACT < 50 nm) with low scattering enhancements were also found during our measurements but in smaller quantities, consistent with the findings of Brooks et al. (2019) for the Indian region during the pre-monsoon and monsoon seasons."

45. Line 9 – smallER

46. Lines 18-19 – This should already be discussed in the introduction.

47. Lines 21-25 – Rather than spend four lines discussing circumstantial evidence, Figure 6 directly dispels the importance of this notion?

Page 20:

48. Lines 10-12 – You can't call the SEAS region "organic-rich". Organics and sulphate are equal over SEAS in Figure 7.

49. Lines 15-20 – You treat the change in organic composition from SEAS to EIO as a chemical loss of organics (this should be referenced). Is it not more likely that

the change in organics is due to mixing of marine air and polluted continental air, as suggested by the EIO trajectories for EIO?

50. What do the details given in the paragraph starting on line 23 and ending on page 21, line 12, have to do with BC? Most of this appears to be a rehash of Aswini et al. (2020), and could be stated much more succinctly.

Page 21:

51. Line 13 to Page 22, line 21 and Figure 8 - The figure and the associated regressions do not give any significant information other than for the case of smaller coatings: sulphate appears to an important contributor to the coating thickness. However, the role of sulphate in that case has already been established from Figure 7. What might be useful here is a plot of the coating thickness against the sum of organics, sulphate and ammonium. If the sums of those species do not represent a significant fraction of the variation in the coating thicknesses, then either there is a problem with some of the measurements (ACSM or coating thicknesses) or there is some chemical specie(s) not measured by the ACSM that is involved in the coating.

52. I recommend removing the discussion from line 23 of page 20 to line 21 of page 22 as well as Figure 8. Alternatively, you need to significantly improve the value of this discussion.

Page 23:

53. Lines 17-26 – As discussed above, the fractional analysis needs to be improved, and the associated conclusions on these lines altered accordingly.

54. As above, the NR chemical composition discussion needs improvement.

---

## Author Comment (AC1) · 22 Feb 2021

**Response to reviewers**

**We thank both reviewers for their constructive and comprehensive comments, which, as outlined below, have helped improve the manuscript. This document outlines the review comments in** *plain italics***, followed by the author's response in bold, and the tracked changes in the main texts are in blue.**

**Reviewer-1:** **ACP-2020-836-RC1**

*The manuscript details the analysis of SP2 data collected aboard a research ship as part of the ICARB-18 campaign with a focus on the microphysical properties of refractory black carbon (rBC)-containing particles. As the authors point out, the SP2 detects individual rBC particles through laser-induced incandescence thereby providing mass loadings and size/mass distributions. By combining the incandescence channel with the co-located SP2 scattering channel, the mixing state of individual rBC particles can also be probed. Their analysis reveals that outflow regions are characterized by higher rBC loadings and more thickly coated rBC-containing particles with a decrease in coating thickness in the oceanic regions. Studies such as this are essential in filling in measurement and information gaps in the South Asian region and thus are important for improving our quantitative understanding of the radiative forcing impacts of BC aerosols in this region. Despite the over use of very long sentences, the manuscript is clearly written. The manuscript shares a lot similarity - with a couple of notable exceptions - to that published by Brooks et al., who conducted an aircraft-based study of northern India. This Reviewer has a concern that the authors are, in some instances, over interpreting data collected by a single instrument and, in doing so, drawing potentially incomplete (or erroneous) conclusions - as exemplified by their use of rBC number fraction (which is discussed below). Therefore, it is recommended that the manuscript be revised as per comments below and resubmitted. To be clear, the findings contained in this manuscript are of good value and need to be published, but in its current state, the manuscript is not quite ready.*

**We thank the reviewer for the summary, the positive recommendation, and the comments for improvement. We have revised the manuscript as detailed below.**

*As alluded to above, 90% of this manuscript - based on rough estimate of pages dedicated to measurement type - involves only data collected by the single particle soot photometer (SP2). (The remaining 10% is on aerosol composition via an ACSM.) While the SP2 in an incredibly sensitive instrument towards rBC particles and its mixing state, it is not sensitive to changes in rBC aggregate morphology - that is rearrangement of a fractal aggregate to one that is more collapsed - and it suffers from limited detection sensitivity towards pure scatterers (i.e., non-BC particles).*

**We agree. We have included the following discussion in the revised manuscript.**

**Page 7, Line 22:**

"It is recognized that the SP2 cannot provide the details of rBC aggregate morphology or the relative position of the BC within the particle, which can be determined better through microscopy-based studies (e.g., Adachi et al., 2010; Ueda et al., 2018). However, the intensity of the incandescence signal detected by the SP2 is proportional to the refractory black carbon mass present in the particle and is independent of particle morphology and mixing state (Slowik et al., 2007a; Moteki and Kondo, 2007; Schwarz et al., 2008). Again, though the SP2 has limited detection sensitivity towards pure scatterers because of the limited size range it covers, the light scattering information at 1064 has been widely used to accurately derive the size of the coated particle (Gao et al., 2007; Moteki et al., 2010; Shiraiwa et al., 2008; 2010; Laborde et al. 2013; Taylor et al., 2015; Liu et al., 2017)."

References:

Adachi, K., S. H. Chung, and P. R. Buseck (2010), Shapes of soot aerosol particles and implications for their effects on climate, J. Geophys. Res., 115, D15206, doi:10.1029/2009JD012868.

Laborde, M., Crippa, M., Tritscher, T., Jurányi, Z., Decarlo, P. F., Temime-Roussel, B., Marchand, N., Eckhardt, S., Stohl, A., Baltensperger, U., Prévôt, A. S. H., Weingartner, E., and Gysel, M.: Black carbon physical properties and mixing state in the European megacity Paris, Atmos. Chem. Phys., 13, 5831–5856, doi:10.5194/acp-13-5831-2013, 2013.

Moteki, N. and Kondo, Y.: Effects of mixing state on black carbon measurements by laser-induced incandescence, Aerosol Sci. Technol., 41, 398–417, doi:10.1080/02786820701199728, 2007.

Moteki, N., Kondo, Y., and Nakamura, S.: Method to measure refractive indices of small nonspherical particles: Application to black carbon particles, J. Aerosol Sci., 41, 513–521, doi:10.1016/j.jaerosci.2010.02.013, 2010.

Schwarz, J. P., Spackman, J. R., Fahey, D.W., Gao, R. S., Lohmann, U., Stier, P., Watts, L. A., Thomson, D. S., Lack, D. A., Pfister, L., Mahoney, M. J., Baumgardner, D., Wilson, J. C., and Reeves, J. M.: Coatings and their enhancement of black carbon light absorption in the tropical atmosphere, J. Geophys. Res., 113, D03203, doi:10.1029/2007jd009042, 2008.

Shiraiwa, M., Kondo, Y., Moteki, N., Takegawa, N., Sahu, L. K., Takami, A., Hatakeyama, S., Yonemura, S., and Blake, D. R.: Radiative impact of mixing state of black carbon aerosol in Asian outflow, J. Geophys. Res., 113, D24210, doi:10.1029/2008jd010546, 2008.

Shiraiwa, M., Kondo, Y., Iwamoto, T., and Kita, K.: Amplification of Light Absorption of Black Carbon by Organic Coating, Aerosol Sci. Technol., 44, 46–54, doi:10.1080/02786820903357686, 2010.

Slowik, J. G., Cross, E. S., Han, J.-H., Davidovits, P., Onasch, T. B., Jayne, J. T., Williams, L. R., Canagaratna, M. R., Worsnop, D. R., Chakrabarty, R. K., Moosmueller, H., Arnott, W. P.,

Schwarz, J. P., Gao, R.-S., Fahey, D. W., Kok, G. L., and Petzold, A.: An inter-comparison of instruments measuring black carbon content of soot particles, Aerosol Sci. Technol., 41, 295– 314, doi:10.1080/02786820701197078, 2007a.

*First a comment on rBC morphology. For a given mass, the SP2 will report the same incandescence intensity independent of the rBC particle fractal dimension. Therefore a statement such as "The observed narrow range of mean NMD (0.10-0.11 µm) and MMD (0.19- 0.20 µm) over the entire region (Fig. 3c and 3d) reveals that the prevailing BC over the entire study region has a mixed-source origin and the particles are aged during the long advection leading to transformation processes (such as collapsing of the BC cores)" is very misleading [page 10, lines 11-12]. The authors again put forth a similar line of reasoning in the conclusion [page 23, lines 4-5] "BC size distributions indicated highly mixed BC sources, likely including a combination of fossil fuel and solid fuel sources, along with the restructuring of the cores of the aged BC particles." The authors present no supporting data that there is any restructuring of the cores and thus cannot conclude that such processes are taking place. Indeed, there is lab studies which suggest that aggregate restructuring occurs through capillary condensation at the interstitial locations between primary particles and thus such restructuring would occur very early in the coating process. The authors are encouraged to review the work by the Khalizov group (e.g., Invanova et al., AST 2020 and references therein).*

**Yes, we agree. Complying with the comment, the discussions and conclusion are modified in the revised manuscript as given below.**

**Page 11, Line 12:**

[revised manuscript text omitted]

Zhang, R., Khalizov, A. F., Pagels, J., Zhang, D., Xue, H., and McMurry, P. H.: Variability in

**morphology, hygroscopicity, and optical properties of soot aerosols during atmospheric processing., Proc. Natl. Acad. Sci. USA, 105, 10291–10296, doi:10.1073/pnas.0804860105, 2008.**

*As highlighted above, this reviewer has a major issue with the authors operational definition of the "fraction of particles containing rBC" (which the authors define as the sum of the number concentration of BC and non-BC scattering particles detected by the SP2). Using the SP2 to measure the number concentration of non-BC will severely bias the derived number fraction of rBC particles because the scattering channel has a severely limited detection range (i.e., ~200 nm - 400 nm). In contrast, the incandescence channel has a nominal detection sensitivity range, in terms of a mass equivalent diameter, from about 80 nm to ~ 500 nm. In short, the SP2 severely undercounts pure scattering particles and therefore deriving a rBC number fraction using the SP2 scattering data will bias the rBC number fractions high and could very well mask the impact(s) that <200 nm non-BC particles might exert in differing regions. One would think (hope) that the instrument suite deployed aboard the research ship during the campaign had a CPC measurement, which would provide an excellent measurement of aerosol number concentration for particles ranging in diameter from ~ 10 nm to 1000 nm, thereby greatly improving the utility of examining the rBC number fraction. Given the presence of higher anthropogenic emissions nearest the coast, it is not unreasonable to expect sub-200 nm pure scattering particles to be abundant. If no CPC, or equivalent measurement, is available then the discussion of variation in fraction of particles containing rBC is meaningless and thus should removed.*

**We agree with the reviewer's point that t*he SP2 severely undercounts pure scattering particles, and therefore deriving an rBC number fraction using the SP2 scattering data will bias the rBC number fractions high and could very well mask the impact(s) that <200 nm non-BC particles might exert in differing regions*. We have performed additional analysis and revised the manuscript. During the present study, the total particle concentrations are available from the condensation particle counter (with a 1-minute sampling interval). The results on particle number size distributions and new particle formation events during the ICARB-2018 are available in Kompalli et al. (2020b). As suggested by the reviewer, the fraction of BC-containing particles was calculated using the total number concentrations from the SMPS, CPC, and rBC number concentration from the SP2. Accordingly, panel (b) of figure 5 and Table-3 have been modified in the revised manuscript. Also, discussions in the manuscript are revised as below to reflect the above points.**
**Page 8, Line 10:**
**"Continuous measurements of the particle number size distributions in size range 10 to 414 nm**

have also been carried out aboard, at the 5-minute interval, using a scanning mobility particle sizer spectrometer (SMPS; TSI Inc., USA) during the campaign (Kompalli et al., 2020b). The SMPS consists of an electrostatic classifier (TSI 3080), a long differential mobility analyzer to size segregate the particles based on their electrical motilities (Wiedensohler 1988), which are subsequently counted by using a water-based condensation particle counter (TSI 3786). Concurrent measurements of the particle number size distributions in the aerodynamic diameters range 542 to 19800 nm (which can be converted to stokes diameters using an effective particle density) have also been made using the aerodynamic particle sizer (Make: TSI, Model: 3321) that works based on 'time-of-flight' technique (Leith and Peters 2003). Though the contribution from the particles in the sizes measured by the APS to the overall aerosol number concentrations is found to be < 2 %, combining both these measurements gives the total particle number concentrations covering a wide size range (10 -10000 nm). We have used the particle number size distributions (and total particle number concentrations) from 10-1000 nm from the SMPS+APS measurements, along with the number concentration of rBC from the SP2 to estimate the fraction of rBC-containing particles."

**Page 16, Lines 21-22:**

"The spatial variation of number concentration (in $cm^{-3}$) of non-BC (i.e., purely scattering) particles detected by the SP2 and the fraction of rBC-containing particles ($F_{BC}$; the ratio of rBC number concentration to the total number concentration in size range 10-1000 nm from the SMPS and APS measurements) are shown in Fig. 5 (a & b)"."

**Page 17, Line 11:**

"The rBC particles constituted about 8-12% of the total number concentration over different sub-regions, on an average. (Fig. 5b)"."

**Page 18, Line 2:**

[revised manuscript text omitted]

*The authors report the observation of exceptionally thickly coated particles. Brooks et al., from which the current manuscript is nominally modeled after, reported on the rBC mixing state in northern India during the pre-monsoon and monsoon seasons. Using their Figure 6 "pre-monsoon (IGP)" plot it is estimated that a 130 nm BC core had a nominal coating thickness is ~110 nm. A similar back-of-the-envelope estimate from the current manuscript's Figure 6 suggests a coating thickness of ~250 nm (the*

*authors are strongly encouraged to adjust the scaling of the color mapping in Figure 6 so that more variation in the coating distribution might be seen). This difference in coating thicknesses for a 130 nm core represents about a 4.5x difference in coating volume. This is a big difference and as such, this Reviewer finds it a bit of stretch for the authors to simply state that "…the present study…..are comparable to the values reported in pollution in-plume air mass regions elsewhere (e.g., Cheng et al., 2019; Brooks et al., 2019)." [page 14, lines 19-20] Given this large difference, surly the authors could offer some discussion on why such a difference exists; that is, why are their rBC coatings so much thicker? Indeed, their reported coating thicknesses exceed what is observed for wildfires, which are known to rapidly create thickly coated rBC particles. This is perplexing and interesting finding that begs the question of why is there such a big difference, yet, no further discussion is provided by the authors beyond a cursory comparison with Brooks. The authors are encouraged to think about this finding and perhaps provide a more detailed discussion to help explain why the coating thicknesses found in this study are some much greater than that found by Brooks and in wildfires.*

**The color scale of Figure 6 is adjusted in the revised manuscript as below.**

[Figure]

**The point raised by the reviewer is an important and pertinent one and needs a detailed discussion. In page 14, lines 19-20, we have compared the present rBC _'bulk'_ coating parameters**

(bulk RCT ~1.73 to 2.15 and ACT~ 69-109 nm) (eq. 1) to the corresponding values reported by Brooks et al. (2019) (in their Fig. 2-4) over the highly polluted IGP region. We have used the bulk coating thickness because the coating thickness for individual particles is dependent on rBC core size. This approach, thus, reduces the uncertainties arising from smaller particles because of their less important contribution to the integrated volume. Further, the $E_{sca}$-$D_c$ analysis is aimed at identifying the different sources. As suggested by the reviewer, we have adjusted the color scale of the figure to reflect variations in the coating distribution among rBC particles. This analysis aims not to predict the $E_{sca}$ quantitatively or identify size segregated coating but rather to use the $E_{sca}$ vs. $D_c$ distributions to identify differences in rBC properties from which we can categorize different sources on a particle by particle basis. Because of this, in Page 18, lines 15 to 24, we have noted the fact that "a greater proportion of thickly coated particles with varying BC core diameters and a wide range (5-800) of scattering enhancement values were observed during the present study". The nominal ACT values shown in figure 6 of the present manuscript are to be taken in this context. We have added the following discussion in the revised according to the reviewer's comments:

Page 21, Line 6:

"Besides the normal mode similar to the one (with core diameter <0.22 μm and coating thickness of 50–200 nm) reported from the aircraft measurements of BC mixing state over the Indian continent by Brooks et al. (2019), an additional mode of BC with BC core sizes ~110-130 nm and a significant coating thickness (> 200 nm) is seen during this study. Such a mode highlights the influence of long-range transport to the ocean from the continent on BC ageing. Though the BC mass loading decreases during the long-range transport, the remaining BC cores gain a greater coating over the ocean than over land (e.g., Moteki et al., 2007). Further, it indicates a strong secondary production of aerosol components during the transport over the ocean, contributing to the BC ageing. The much thicker coatings seen during the ICABR-2018 compared to the observations from the ground-based site (Kompalli et al., 2020a) and the aircraft measurements (Brooks et al., 2019) over the Indian region are also indicative of other sources (with poor combustion efficiencies such as biomass burning) being prevalent in this region. Gong et al. (2016) have reported thick ACT (~110–300 nm) values during a biomass burning pollution episode in urban Shanghai, comparable to the present BC mode. These values are higher than the values reported from the aircraft measurements over biomass burning plumes (~ 150 nm; Ditas et al., 2018), the southeast Atlantic Ocean ( ~ 90 nm in the boundary layer and ~120 nm in the free troposphere; Taylor et al., 2020) and aged smoke in Amazonia (55-90 nm) (Darbyshire et al., 2019). Therefore, the measurement over the ocean thus offers an opportunity to study a

**more aged BC from the continent outflow."**

**Additional references:**

Ditas, J., Ma, N., Zhang, Y., Assmann, D., Neumaier, M., Riede, H., Karu, E., Williams, J., Scharffe, D., Wang, Q., Saturno, J., Schwarz, J. P., Katich, J. M., McMeeking, G. R., Zahn, A., Hermann, M., Brenninkmeijer, C. A., Andreae, M. O., Pöschl, U., Su, H., and Cheng, Y.: Strong impact of wildfires on the abundance and aging of black carbon in the lowermost stratosphere, P. Natl. Acad. Sci. USA, 115, E11595–E11603, https://doi.org/10.1073/pnas.1806868115, 2018.

Darbyshire, E., Morgan, W. T., Allan, J. D., Liu, D., Flynn, M. J., Dorsey, J. R., O'Shea, S. J., Lowe, D., Szpek, K., Marenco,F., Johnson, B. T., Bauguitte, S., Haywood, J. M., Brito, J. F., Artaxo, P., Longo, K. M., and Coe, H.: The vertical distribution of biomass burning pollution over tropical South America from aircraft in situ measurements during SAMBBA, Atmos. Chem. Phys., 19, 5771–5790, https://doi.org/10.5194/acp- 19-5771-2019, 2019.

Taylor, J. W., Wu, H., Szpek, K., Bower, K., Crawford, I., Flynn, M. J., Williams, P. I., Dorsey, J., Langridge, J. M., Cotterell, M. I., Fox, C., Davies, N. W., Haywood, J. M., and Coe, H.: Absorption closure in highly aged biomass burning smoke, Atmos. Chem. Phys., 20, 11201–11221, https://doi.org/10.5194/acp-20-11201-2020, 2020.

*The authors report the organic contribution to the coating as being responsible for ~40% near the coast and in the remote ocean, accounting for a little over 20%, as inferred from the ACSM. What is the influence of having a mixture of sulfate and organics on the refractive index assumed in the coating calculation?*

This is an important question, and sorry for the confusion created. We have not quantified the percentage contribution of organics or sulfate to the coating on rBC particles using the ACSM measurements. We have examined the association between the non-refractory submicron aerosol mass concentrations (NR-PM1.0) and bulk absolute coating thickness of rBC for low and high ACT observations. There is no significant association between bulk ACT and mass concentrations of NR-PM1.0 species during high ACT (ACT > 50 % of MMD) observations, suggesting complex coatings. However, for the BC population with low ACT values, a significant correlation (r ~0.62; p <0.01) was found between sulfate and rBC coating.

The value of 1.5+0i used for the coating refractive index in this study corresponds to values for ambient scattering aerosols reported in earlier studies (Metcalf et al., 2012; Laborde et al., 2013; Taylor et al., 2015). Laborde et al. (2013) have also suggested this as optimum value, which

is in the range of refractive indices of inorganic salts ($(NH_4)_2SO_4$ = 1.51; NaCl = 1.53) and secondary organic aerosol (~1.44-1.5) at λ = 1064 nm. Taylor et al. (2015) have presented a detailed assessment of the sensitivity of core/shell parameters derived using the single-particle soot photometer and reported that the refractive index of coatings has only a minor influence. They have also detailed the sources of uncertainties associated with it. Therefore, a mixture of sulfate and organics will not have a notable influence on the refractive index assumed in the coating calculation or coating parameters. We have added the following details in the revised manuscript.

**Page 7, Line 10:**

"Further, the amplitude of the scattering signal provides information about the scattering cross-section of the particles, which is used to determine the optical sizing of the particles. In the case of BC-containing particles, the scattering signal gets distorted as it passes through the laser beam because of the intense thermal heating of the particle and evaporation of the coating. Thus, the scattering signal of the BC particle is reconstructed using a leading-edge only (LEO) fitting technique, as described in Gao et al. (2007); Liu et al. (2010, 2014, 2017). This scattering cross-section is matched with the modeled values in a Mie-lookup table to derive the optical diameter of a BC particle or the coated BC size ($D_p$). Here, the total particle is treated as an ideal two-component sphere with a concentric core-shell morphology, with a core (rBC) refractive index value of 2.26 − 1.26i (Moteki et al., 2010; Liu et al., 2014; Taylor et al., 2015) and a coating refractive index of 1.5+0i (which is an optimum value and in the range of refractive indices of inorganic salts ($(NH_4)_2SO_4$ = 1.51; NaCl = 1.53) and secondary organic aerosol (~1.44-1.5) at λ = 1064 nm (Schnaiter et al., 2005; Metcalf et al., 2012; Lambe et al., 2013; Laborde et al., 2013; Taylor et al., 2015). These two diameters ($D_p$ and Dc) are used to infer the coating thickness."

The following references are added in the revised manuscript:

Metcalf, A. R., Craven, J. S., Ensberg, J. J., Brioude, J., Angevine, W., Sorooshian, A., Duong, H. T., Jonsson, H. H., Flagan, R. C., and Seinfeld, J. H.: Black carbon aerosol over the Los Angeles Basin during CalNex, J. Geophys. Res., 117, D00V13, https://doi.org/10.1029/2011jd017255, 2012.

Schnaiter, M., Linke, C., Mohler, O., Naumann, K. H., Saathoff, H., Wagner, R., Schurath, U., and Wehner, B.: Absorption amplification of black carbon internally mixed with secondary organic aerosol, J. Geophys. Res., 110, D19204, doi:10.1029/2005jd006046, 2005.

Lambe, A. T., Cappa, C. D., Massoli, P., Onasch, T. B., Forestieri, S. D., Martin, A. T., Cummings,

M. J., Croasdale, D. R., Brune, W. H.,Worsnop, D. R., and Davidovits, P.: Relationship between oxidation level and optical properties of secondary organic aerosol, Environ. Sci. Technol., 47, 6349–6357, 2013. https://doi.org/10.1021/es401043j.

*Additionally, while a density of 1.7 g/cc is reasonable for sulfates, it is high for organics (1.4 g/cc). Perhaps authors could conduct some sensitivity calculations to see if these factors exert influence on the derived coating thicknesses.*

**The coated material density was used only to derive the bulk mixing ratio of coating mass over rBC mass ($M_{R,bulk}$). The other mixing state parameters, such as coating thickness, are independent of coating density values. As described previously, for the estimation of core and coated BC particle diameters, only the core density value (1.8 g cm$^{-3}$) along with the refractive indices of core (2.26 – 1.26i) and coating (1.5+0i) were used. In this regard, we have revised the portion on $M_{Rbulk}$ in the manuscript as suggested by the reviewer.**

**In the revised manuscript, we have used the effective dry density of ambient NR-PM1.0 based on the measured near-real-time chemical composition by assuming densities of organics and inorganics species as 1.4 (Hallquist et al., 2009) and 1.77 g cm$^{-3}$ (Park et al., 2004), respectively. These revisions are also reflected in Figure 5 and Table-3.**

[revised manuscript text omitted]

*The authors are sometimes using the term "aging" in a very confusing and, potentially, misleading way. For example, the authors write [page 14, lines 4-7] "While the larger BC particles are scavenged rather quickly, the smaller, less-aged, and relatively less- coated BC particles (occasionally, even bare soot particles) can persist in the outflow and be transported to the remote marine regions." This is misleading - more thinly coated BC particles are not necessarily "less-aged". Particles can undergo photolysis or heterogeneous oxidation that can bring about fragmentation leading to material loss as particles age leading to thinner coatings or larger diameter particles can be preferentially scavenged leaving behind smaller and more thinly coated particle. So to label a smaller, less-coated BC particle*

*- even bare soot particles - as less-aged is wrong. Correct this.*

**Sorry for the confusion created apparently by the adjectives being considered sequentially. We did not mean that the thinly coated particles are less aged. We have removed 'less-aged' to avoid confusion and revised the discussion using the statements suggested by the referee as below.**

**Page 15, Line 28:**

**"While the larger BC particles are scavenged rather quickly, the smaller and relatively less-coated BC particles (occasionally, even bare soot particles) can persist in the outflow and be transported to the remote marine regions (Ueda et al., 2018). As the particles spend more time in the atmosphere, they tend to gain coating material on them. Simultaneously, the loss of coating material on the particles cannot be ruled out due to photolysis or heterogeneous oxidation that can bring about fragmentation, leading to thinner coatings. Thus, preferential scavenging of larger particles leaving behind smaller and more thinly coated particles and atmospheric processes leading to loss of condensable material, explains the broad range of MMD (Fig. 3d) and lower RCT values observed over the EIO. Furthermore, in cleaner maritime regions like the equatorial Indian Ocean, the aging of BC occurs slowly due to reduced availability of coating material."**

*Do the authors have access to optical property measurements, specifically aerosol light absorption? If they do, they are strongly encouraged to roll that into this study so as to make this a more complete analysis.*

**We had an aethalometer aboard, measuring BC using optical absorption. However, the team concerned with those data are still in the process of analysing and as such, we are unable to include this in the present manuscript.**

*Finally, it seems to this reviewer that the authors have additional data mining that they could do. Given their bulk coating mass-to-core ratio and the mass loadings of non-refractory aerosols via the ACSM, it might prove very interesting to examine the ratio of non-refractory coating to overall non-refractory aerosol mass. Using the bulk mixing ratio of coating mass over rBC mass (Table 3) along with the rBC mass for the SEAS leg (Table 3) and the reported NR-PM mass in Figure 7, a back-of-the-envelope calculation suggests that about ~40% of the NR-PM mass is bound to BC particles, whereas for the EIO leg this ratio drops to ~ 15%. This suggests the preferential loss of coating and also suggests that the rBC number fraction of particles does change significantly (as per this Reviewer's comments above on this subject).*

**We thank the reviewer for this suggestion. Additional analyses have been carried out accordingly, and the following figure and discussion are added in the revised manuscript.**

"In supplementary figure S4, the ratio (expressed as %) of non-refractory coating mass on BC to the total NR-PM1.0 mass concentrations (from the ACSM) are shown. The mean ratios (varying between 23-35%) for different sub-regions are also mentioned in the figure."

[Figure]

**Figure S4: Spatial variation of the ratio (in %) of non-refractory coating mass on BC to the total mass concentration of non-refractory aerosols during the ICARB-2018. The mean values over different sub-regions are also written in the figure.**

"It revealed that a substantial portion of the non-refractory submicron aerosol mass is bound to the BC particles over the regions with stronger outflow. While higher mean values of the ratio were observed over the SEAS (~35 ± 12 %) and NIO-E (31 ± 4) regions in the proximity of the sources, as one move away to the farther oceanic EIO, the NR-PM1.0 that is bound to BC particles dropped considerably (mean ~23 ± 8%). The values occasionally dropped to as low as 5-6%, indicating that (a) preferential loss of coating on rBC particle or (ii) a substantial contribution from non-rBC particles to the NR-PM1.0 mass loading due to new particle formation and subsequent growth events (Kompalli et al., 2020b). Though the mean fraction of r-BC containing particles remained similar (~8%) over different regions (SEAS, NIO-E and EIO), frequent new particle formation events (Kompalli et al., 2020b) resulted in lower $F_{BC}$ (< 2%) over the EIO region which reflected in the amounts of NR-PM1.0 mass that are bound to rBC particles."

**Specific issues**:

*Page 3*, *line 5.* *"Produced by the incomplete (low-temperature) combustion of hydro-carbon fuels. . ."* *"Low temperature" is subjective, please quantify or remove.*

**Complied with. The word 'low-temperature' is removed.**

*Page 3*, *line 20.* *"The sources of BC are highly heterogeneous." What do the authors mean by "heterogenous"? There are three sources of BC: fossil fuel, biomass burning (wildfires and agricultural burns), and biofuel combustion. Please clarify what you mean by heterogeneous.*

**Complied with. The sentence is now revised as "The sources of BC are highly varying, both seasonally and spatially, over the Indian region (e.g., Kompalli et al., 2014; Prasad et al., 2018 and references therein)"**

**References:**

[revised manuscript text omitted]

*Also, this Reviewer is quite surprised that there is very little discussion about the possible impacts of cloud processing of rBC particles that this might have on this observation (not to be confused with scavenging).*

**The reviewer has made a very important observation. These aspects are now included in the revised manuscript.**

**Page 11, Line 33:**

**"Notably, the NIO-E region depicted slightly larger mean MMD (~0.20 µm) due to frequent larger values (35 % of the measurements showed MMD> 0.20 µm) compared to all the other regions (Fig. 3d). This is a result of the following possibilities: (i) Self-coagulation of rBC cores due to enhanced atmospheric aging during their transport from the source regions in the east-coast to the adjacent marine regions (at the same time, sedimentation of larger particles resulting in a large reduction in number concentration and mass concentration). It may be noted that coagulation, though increases the rBC core diameters and reduces number concentrations, is a slow process. The coagulation rate depends on the square of the particle number concentrations and is the least between particles of the same size. Thus, the coagulation rates would be higher near source regions of the nascent aerosols and dropping off gradually at farther distances; (ii) the second and most important possibility is associated with the cloud processing of rBC particles. The insoluble BC particles remain within a non-precipitating cloud as interstitial particles. A cloud undergoes multiple evaporation-condensation cycles before it transforms into a precipitating system. During such cycles, interstitial BC in cloud droplets can grow larger (especially following the evaporation of cloud droplets containing multiple rBC particles) due to agglomeration with other interstitial rBC aerosols; (iii) The third possibility is the varying nature of dominant sources. A sizeable increase in the contribution from solid fuel sources (biomass/crop residue/coal burning) in the upwind regions (the eastern coast of India) through the transported air masses can lead to larger BC cores (Brooks et al., 2019; Kompalli et al., 2020a)."**

**Another point reviewer made is whether the nominal low rBC number concentration such as ~200 cm$^{-3}$ can result in enough self-coagulated rBC particles within the time frame of these measurements to account for ~ 35% of the measurements in this region showing MMD > 0.2 µm. Since rBC particles originate from the regions of great source strengths (possibly, significant aerosol number concentrations compared to the values seen over the NIO-E), the rate of coagulation will be greatest near the source regions where nascent BC particles dominate. But, as the BC aerosols traverse longer distances to reach the marine regions away from the source regions, as pointed out by the reviewer, number concentration (and the rate of coagulation) is expected to decrease with**

time. As such, subsequently, the contribution of the coagulation process to particle size transformation will be negligible. Apart from BC-BC coagulation, BC may coagulate with non-BC particles, which leads to increased BC coating thickness. The observed thicker coatings over the NIO-E are consistent with this.

*Page 15, lines 14-17. Do the authors have access to any independent measurements of size distributions beyond solely relying on the SP2? As pointed out above, independent measurements of the microphysical properties would help in buttressing interpretations and, in some cases (e.g., rBC number fraction) enable a robust analysis to be performed.*

Yes, we had independent measurements of aerosol size distributions using SMPS and used those in the revised manuscript (Figure 5b and Table-3 have been modified, and relevant discussion is revised), gratefully acknowledging the reviewer's valuable suggestions.

---

## Author Comment (AC2) · 22 Feb 2021

**Response to reviewers**

We thank both reviewers for their constructive and comprehensive comments, which, as outlined below, have helped improve the manuscript. This document outlines the review comments in *plain italics*, followed by the author's response in **bold**, and the tracked changes in the main texts are in **blue**.

**Reviewer-2:** **ACP-2020-836-RC2**

*The information on various characteristics of rBC components of particles over the Indian Ocean is important. Overall, the presentation is clear, but the paper is longer than necessary for the information presented. The introduction is nicely written, but the discussion of results is verbose in many spots and should be written more succinctly. My major technical concerns relate to the use of the scattering signal from the SP2 (see specific comments on page 15 and beyond) and the non-refractory material contributing to the BC coatings. I hope the authors find these comments helpful.*

**We thank the reviewer for appreciating the importance of our work and the detailed comments. We have revised the manuscript considering these and another anonymous reviewer. Our responses to the specific comments of the reviewer are given below, following the same as for reviewer #1:**

*Page 3:*

*(1) Line 20 – "It has a long lifetime." This sentence is a bit abrupt and 'long lifetime' needs to be defined.*

**Complied with. The sentence is now revised as "Aerosol BC has an average atmospheric lifetime of about a week (Lund et al., 2018; Bond et al., 2013). It is prone to regional as well as long-range transport during its short atmospheric lifetime and found even over remote regions, such as the Polar Regions, albeit in lower concentrations (Raatikainen et al., 2015; Liu et al., 2015; Sharma et al., 2017; Zanatta et al., 2018)."**

**The following references are added:**

Lund, M. T., Samset, B. H., Skeie, R. B., Watson-Parris, D., Katich, J. M., Schwarz, J. P., & Weinzierl, B.: Short black carbon lifetime inferred from a global set of aircraft observations. npj Climate and Atmospheric Science, 1, 1-8. https://doi.org/10.1038/s41612-018-0040-x, 2018.

Sharma, S., Leaitch, W. R., Huang, L., Veber, D., Kolonjari, F., Zhang, W., Hanna, S. J., Bertram, A. K., and Ogren, J. A.: An evaluation of three methods for measuring black carbon in Alert, Canada, Atmos. Chem. Phys., 17, 15225–15243, https://doi.org/10.5194/acp-17-15225-2017, 2017.

Zanatta, M., Laj, P., Gysel, M., Baltensperger, U., Vratolis, S., Eleftheriadis, K., Kondo, Y.,

Dubuisson, P., Winiarek, V., Kazadzis, S., Tunved, P., and Jacobi, H.-W.: Effects of mixing state on optical and radiative properties of black carbon in the European Arctic, Atmos. Chem. Phys., 18, 14037–14057, https://doi.org/10.5194/acp-18-14037-2018, 2018.

*Line 21 - What is it about sources of BC in a clean environment that reduce its relative aging? Are BC emissions unaccompanied by fewer other emitted components?*

**Yes. Higher concentrations of condensable vapors (some of which are co-emitted species) contribute to faster aging of BC (Wang et al., 2014; Peng et al., 2016) in polluted environments, as compared to cleaner environments.**

*Lines 30-32 – "When air masses from such complex source regions are transported to remote regions devoid of any sources of BC, the aging becomes important, and the abundance of distinct species with varying lifetimes in the atmosphere differs significantly." This statement needs some clarification: 1) why and where does the aging process suddenly become important; 2) what does the abundance of distinct species and their lifetimes have to do with BC?*

**This sentence is modified in the revised manuscript as below:**

**Page 3, Line 39:**

 **"When air masses from such complex source regions are transported to remote regions devoid of any BC sources, the mixing state of BC becomes complicated. This is due to (a) restructuring of the BC aggregates during the transport due to different processes (Kutz and Schmidt-Ott, 1992; Weingartner et al., 1995; Slowik et al., 2007b; Pagels et al., 2009), and (ii) varying nature and amounts of coating material arising due to the different atmospheric lifetimes and microphysical processes involving different species (McFiggans et al., 2015)."**

**The following references are added in the revised manuscript:**

Kütz, S., and Schmidt-Ott, A.: Characterization of agglomerates by condensation-induced restructuring. J. Aerosol Sci., 23, 357–360, https://doi.org/10.1016/0021-8502(92)90423-S, 1992.

Pagels, J., Khalizov, A. F., McMurry, P. H., and Zhang, R. Y.: Processing of Soot by Controlled Sulphuric Acid and Water Condensation Mass and Mobility Relationship, Aerosol Sci. Technol., 43, 629–640, https://doi.org/10.1080/02786820902810685, 2009.

Slowik, J. G., Cross, E. S., Han, J.-H., Kolucki, J., Davidovits, P., Williams, L. R., et al.: Measurements of morphology changes of fractal soot particles using coating and denuding experiments: Implications for optical absorption and atmospheric lifetime. Aerosol Sci.

Technol., 41(8), 734–750, 2007b. https://doi.org/10.1080/02786820701432632.

Weingartner, E., Burtscher H., and Baltensperger U.: Growth and structural change of combustion aerosols at high relative humidity, Environ. Sci. Technol., 29(12), 2982-2986, 1995. https://doi.org/10.1021/es00012a014.

McFiggans, G., Alfarra, M. R., Allan, J. D., Coe, H., Hamilton, J. F., Harrison, R. M., Jenkin, M. E., Lewis, A. C., Moller, S.J., and Williams, P. I. (2015). A review of the state-of-the-science relating to secondary particulate matter of relevance to the composition of the UK atmosphere. Full technical report to Defra, project AQ0732.

*(2) Lines 35-40 – This sentence is much longer than needed, especially since it does not tell us anything about the subject.*

**Complied with. This sentence is modified in the revised manuscript as below:**

"**The past field campaigns, such as the Indian Ocean Experiment (INDOEX) during 1998-1999 (Ramanathan et al., 2001), the Integrated Campaign for Aerosols, gases, and Radiation Budget (ICARB) during March-May 2006 (phase-1); December-January 2008-2009 (phase-2) (Moorthy et al., 2008; Babu et al., 2012; Kompalli et al., 2013) have characterized regional aerosols over the northern Indian Ocean during different seasons.**"

*Page 4:*

*(3) Line 5 – Everything but the AMS is referenced. Why not the AMS?*

**Complied with. The following references are added for the AMS.**

"**…such as the aerosol mass spectrometer (AMS) (Jayne et al., 2000; Jimenez et al., 2003; Allan et al., 2003),…**".

**References:**

Allan, J., Jimenez, J., Williams, P., Alfarra, M., Bower, K., Jayne, J., Coe, H., and Worsnop, D.: Quantitative sampling using an Aerodyne aerosol mass spectrometer: 1. Techniques of data interpretation and error analysis, J. Geophys. Res.-Atmos., 108, 4090, doi:10.1029/2003JD001607, 2003.

Jayne, J. T., Leard, D. C., Zhang, X. F., Davidovits, P., Smith, K. A., Kolb, C. E., and Worsnop, D. R.: Development of an aerosol mass spectrometer for size and composition analysis of submicron particles, Aerosol Sci. Technol., 33, 49–70, 2000.

Jimenez, J. L., Jayne, J. T., Shi, Q., Kolb, C. E., Worsnop, D. R., Yourshaw, I., Seinfeld, J. H., Flagan, R. C., Zhang, X. F., Smith, K. A., Morris, J. W., and Davidovits, P.: Ambient

aerosol sampling using the Aerodyne Aerosol Mass Spectrometer, J. Geophys. Res.-Atmos., 108(D7), 8425, doi:10.1029/2001JD001213, 2003.

*(4) Line 8 – "such" is redundant.*

**Deleted.**

*(5) Line 15 – There is no verb. Perhaps, "what are the sources of BC and how does its mixing state evolve during transport to and over the ocean".*

**This is corrected in the revised manuscript as below.**

**" … ii) examination of the extent of BC transport from distinct source regions and changes to its mixing state during the transport to the ocean, and (iii) quantification of the degree of coating on BC and identification of the nature of potential coating species by using concurrent chemical composition measurements during the South Asian outflow."**

*(6) Line 28 - Do you know that particles smaller than 10 um were efficiently sampled or are you just assuming they were?*

**Prior to the experiment, we have characterized the sampling inlet system and examined the sampling losses, both theoretically and experimentally. For this purpose, the particle number size distribution measurements were used. We found that the difference between the number concentrations with and without inlet system was < 5% for sizes up to 1000 nm, 5-20% between 1000- 6000 nm, and > 30 % for sizes > 6000 nm.**

*Page 6:*

*(7) Line 7 – In recent years, the terminology for this is commonly "equivalent black carbon" or EBC (Petzold et al., Recommendations for reporting "black carbon" measurements, Atmos. Chem. Phys., 13, 8356- 8379, doi:10.5194/acp-13-8365-2013, 2013.). Your oBC may be reasonable, but there is no reason to introduce your new definition, unless there has been some more recent change in terminology that I am unaware of.*

**We have now made the abbreviations consistent with Petzold et al. (2013). oBC is now changed to equivalent black carbon (EBC) in the manuscript.**

*Page 7:*

*(8) Lines 19-20 – It is at best questionable as to whether the ACSM truly measures PM1.0. It needs a reference.*

**We have provided the details as below:**

**"Supplementing the above, we have used the information on the mass concentration of non-refractory PM1.0 aerosols (organics, sulfate, ammonium, nitrate, and chloride) from a collocated aerosol chemical speciation monitor (ACSM; Model: 140; Aerodyne, USA; Ng et al., 2011). The objective here is to identify the possible coating material on rBC particles. The ACSM consists of a particle sampling inlet, three vacuum chambers (differentially pumped by turbopumps, backed by the main diaphragm pump), a residual gas analyzer (RGA) mass spectrometer (Pfeiffer Vacuum GmbH). The particles are drawn to an aerodynamic lens assembly having $D_{50}$ limits (50% transmission range) of 75-650 nm and 30 to 40 % transmission efficiency at 1 μm (Liu et al., 2007) through a 100 μm critical orifice. These particles are focused into a narrow beam and transmitted to a vacuum environment where they are flash-vaporized by the thermal capture vaporizer (Xu et al., 2017; Hu et al., 2017a, 2017b) operating at 525 °C. Subsequently, these vapors are ionized via 70 eV electron impact ionization and detected with a quadrupole mass spectrometer.**

**Additional References:**

Hu, W. W., Campuzano-Jost, P., Day, D. A., Croteau, P., Canagaratna, M. R., Jayne, J. T., Worsnop, D. R., and Jimenez, J. L.: Evaluation of the new capture vaporizer for aerosol mass spectrometers (AMS) through laboratory studies of inorganic species, Atmos. Meas. Tech., 10, 2897–2921, https://doi.org/10.5194/amt-10-2897-2017, 2017a.

Hu, W. W., Campuzano-Jost, P., Day, D. A., Croteau, P., Canagaratna, M. R., Jayne, J. T., Worsnop, D. R., and Jimenez, J. L.: Evaluation of the new capture vaporizer for aerosol mass spectrometers (AMS) through field studies of inorganic species, Aerosol Sci. Technol., 51, 735–754, https://doi.org/10.1080/02786826.2017.1296104, 2017b.

Liu, P. S. K., Deng, R., Smith, K. A., Williams, L. R., Jayne, J. T., Canagaratna, M. R., Moore, K., Onasch, T. B., Worsnop, D. R., and Deshler, T.: Transmission efficiency of an aerodynamic focusing lens system: comparison of model calculations and laboratory measurements for the Aerodyne Aerosol Mass Spectrometer, Aerosol. Sci. Tech., 41, 721–733, 2007.

Xu, W., Croteau, P., Williams, L., Canagaratna, M., Onasch, T., Cross, E., Zhang, X., Robinson, W., Worsnop, D., and Jayne, J.: Laboratory characterization of an aerosol chemical speciation monitor with PM2:5 measurement capability, Aerosol Sci. Technol., 51, 69–83, https://doi.org/10.1080/02786826.2016.1241859, 2017.

*(9) Line 24 – Why are so many details of the SP2 given, yet you are unwilling to give the "prescribed methodology"? The prescribed methodology is more germane to your analysis than the details of the ACSM that have been known for over a decade.*

**We have updated the details on the SP2 methodology, and as suggested by the reviewer, added**

**more details on the ACSM data methodology in the 'Analysis' section in the revised manuscript.**

**Page 7, Line 40:**

"**The data is processed as per the prescribed methodology (Ng et al., 2011; Kompalli et al., 2020a). We have used software provided by the manufacturer (Aerodyne Research, ACSM Local, version 1.6.0.3, within IGOR Pro version 7.0.4.1) for processing and analysis of data. Using the default fragmentation table (Allan et al., 2004), the measured fractions of unit mass resolution spectra signals were apportioned to individual aerosol species. The required corrections for the instrument performance for the varying inlet pressures and $N_2$ signal were performed (Ng et al., 2011; Sun et al., 2012). Mass-dependent ion transmission efficiency correction of the residual gas analyzer was carried out using the signals from the internal diffuse naphthalene source (m/z 128). The calibrations of ionization efficiency (IE) and relative IE (RIE) calibrations were performed prior to the experiment by using monodisperse (300 nm) particles of $NH_4NO_3$ and $(NH_4)_2SO_4$ (Jayne et al., 2000; Allan et al., 2003; Jimenez et al., 2003; Canagaratna et al., 2007). The present ACSM consists of a capture vaporizer with an inner cavity to reduce the particle bounce (Xu et al., 2017), resulting in a higher collection efficiency (about unity) (Hu et al., 2017a; 2017b). Therefore, the composition-dependent collection efficiency correction prescribed by Middlebrook et al. (2012), applicable to standard vaporizer instruments, was not applied to our data. More than 1200 quality checked individual observations with a time resolution of ~30 minutes formed the database for this study.**"

**The following references are added:**

**Allan, J., Jimenez, J., Williams, P., Alfarra, M., Bower, K., Jayne, J., Coe, H., and Worsnop, D.: Quantitative sampling using an Aerodyne aerosol mass spectrometer 1. techniques of data interpretation and error analysis, J. Geophys. Res., 108, 4090, doi:10.1029/2002JD002358, 2003.**

**Allan, J., Delia, A. E., Coe, H., Bower, K. N., Alfrarra, R., Jimenez, J. L., Middlebrook, A. M., Drewnick, F., Onasch, T. B., Canagaratna, M. R., Jayne, J., and Worsnop, D. R.: Technical note: a generalised method for the extraction of chemically resolved mass spectra from Aerodyne aerosol mass spectrometer data, J. Aerosol Sci., 35, 909–922, doi:10.1016/j.jaerosci.2004.02.007, 2004.**

**Canagaratna, M., Jayne, J., Jimenez, J., Allan, J., Alfarra, M., Zhang, Q., Onasch, T., Drewnick, F., Coe, H., Middlebrook, A., Delia, A., Williams, L., Trimborn, A., Northway, M., DeCarlo, P., Kolb, C., Davidovits, P., and Worsnop, D.: Chemical and microphysical characterisation of ambient aerosols with the Aerodyne aerosol mass spectrometer, Mass Spectrom. Rev., 26, 185–222, doi:10.1002/Mas.20115, 2007.**

**Hu, W. W., Campuzano-Jost, P., Day, D. A., Croteau, P., Canagaratna, M. R., Jayne, J. T., Worsnop, D. R., and Jimenez, J. L.: Evaluation of the new capture vapourizer for aerosol mass**

spectrometers (AMS) through laboratory studies of inorganic species, Atmos. Meas. Tech., 10, 2897–2921, https://doi.org/10.5194/amt-10-2897-2017, 2017a.

Hu, W. W., Campuzano-Jost, P., Day, D. A., Croteau, P., Canagaratna, M. R., Jayne, J. T., Worsnop, D. R., and Jimenez, J. L.: Evaluation of the new capture vaporizer for aerosol mass spectrometers (AMS) through field studies of inorganic species, Aerosol Sci. Technol., 51, 735–754, https://doi.org/10.1080/02786826.2017.1296104, 2017b.

Jayne, J., Leard, D., Zhang, X., Davidovits, P., Smith, K., Kolb, C., and Worsnop, D.: Development of an aerosol mass spectrometer for size and composition analysis of submicron particles, Aerosol Sci. Tech., 33, 49–70, 2000.

Jimenez, J. L., Jayne, J. T., Shi, Q., Kolb, C. E., Worsnop, D. R., Yourshaw, I., Seinfeld, J. H., Flagan, R. C., Zhang, X., Smith, K. A., Morris, J., and Davidovits, P.: Ambient aerosol sampling with an aerosol mass spectrometer. J. Geophys. Res.-Atmos., 108, 8425, doi:10.1029/2001JD001213, 2003.

Sun, Y., Wang, Z., Dong, H., Yang, T., Li, J., Pan, X., Chen, P. and Jayne, T.P.: Characterization of Summer Organic and Inorganic Aerosols in Beijing, China with an Aerosol Chemical Speciation Monitor. Atmos. Environ. 51, 250–259, doi:10.1016/j.atmosenv.2012.01.013, 2012.

Xu, W., Croteau, P., Williams, L., Canagaratna, M., Onasch, T., Cross, E., Zhang, X., Robinson, W., Worsnop, D., and Jayne, J.: Laboratory characterization of an aerosol chemical speciation monitor with PM2:5 measurement capability, Aerosol Sci. Technol., 51, 69–83, https://doi.org/10.1080/02786826.2016.1241859, 2017.

*(10)Line 37 – Page 8, line 3 – Why fit a log-normal distribution to the data when you can calculate MMD and NMD directly from the measurements? Are you sure the data always follow one log-normal mode?*

**Since the fairly narrow measurement range of the SP2 instrument is unlikely to represent the total ambient number and mass concentrations of BC, extrapolation of a log-normal distribution is used to predict these missing masses (below and above the detection limits), as has been done in earlier studies (McMeeking et al., 2010; Metcalf et al., 2012; Reddington et al., 2013; Liu et al., 2010, 2014). As rightly pointed by the reviewer, unimodal distribution always may not be the case for BC core size distributions. At times, an additional mode at sizes below the detection limit of the SP2 is also possible near source regions. However, such a mode (~ 50 nm) is unlikely far away from source regions, which is the case in the present study.**

*Page 9:*

*(11) Line 7 – What does the error indicate: experimental uncertainty in the mean, 25th percentile or standard deviation, etc.?*

**The values after ± indicate standard deviation. This is now explicitly mentioned in the revised manuscript.**

"**The highest values and variabilities (standard deviation) in the rBC mass (mean ~938 ± 293 ng m$^{-3}$) and number (~378 ± 137 cm$^{-3}$) concentrations…**"

*(12) Lines 9-11 – If you are going to suddenly toss in oBC (or EBC) measurements, then you need to discuss why they are about a factor of two higher than the rBC.*

**We have removed this sentence in the revised manuscript.**

(13) *Lines 14-15 – Concentrations of rBC reaching 200 ng/m3 would not be defined as extremely low in other parts of the world. In the Arctic, for example, such concentrations, which can be present in the Arctic Haze, are considered high. Please replace with something like "The lower concentrations. .."*

**Complied with. The sentence is now revised:**

" **The lower concentrations (<200 ng m$^{-3}$) highlighted…**"

*(14) Lines 18-21 – Since the trajectories suggest that NIO-W and NIO-E are roughly equidistant from the sources, does this mean that the source strengths on the west and east coasts are similar, and that the reduction from the SEAS region is mostly due to dispersion during transport?*

**Both the SEAS and NIO-W regions received continental outflow air masses from the west coast/peninsular India. Dispersion during the transport is an important factor governing the decrease in BC concentrations away from the source regions which is reflected in the spatial heterogeneity between SEAS and NIO-W. However, the strength of advection or sources between NIO-E and NIO-W cannot be inferred from this. We have referred to the earlier publications that reported the contrast between the east and west coast regions of India for this purpose. (Fig.S1 c&d; Moorthy et al., 2005; Kompalli et al., 2013)**

*Page 10:*

*(15) Lines 2-5 – References needed.*

**These sentences are rewritten in the revised manuscript, and references are provided.**

**Page 10, Line 11:**

"**The MMD values of the rBC size distributions are strongly influenced by the source of BC emissions (e.g., Ko et al., 2020; Cheng et al., 2018). Recently, Ko et al. (2020) have compared the MMD and NMD values of rBC size distributions from different dominant sources. Previous studies report that the MMD (and NMD) values over the regions dominated by fresh fossil fuel emissions are smaller (MMD ~100-178 nm and NMD ~ 30 to 80 nm) compared to the areas with dominant solid-fuel sources (biomass, biofuel, coal-burning) (MMD ~130-210 nm and NMD ~100 to 140 nm), whereas, well-aged and background BC particles in outflow regions have MMD values in between (MMD ~ 180-225 nm and NMD ~ 90-120 nm)** (McMeeking et al., 2010; Liu et al., 2010, 2014; Kondo et al., 2011; Cappa et al., 2012; Sahu et al., 2012; **Metcalf et al., 2012;** Laborde et al., 2013; Reddington et al., 2013; Gong et al., 2016; Raatikainen et al., 2017; **Krasowsky et al., 2018;** Brooks et al., 2019; Kompalli et al., 2020a; **Ko et al., 2020)**".

**The following additional references are added:**

**Krasowsky, T. S., Mcmeeking, G. R., Sioutas, C., and Ban-Weiss, G.: Characterizing the evolution of physical properties and mixing state of black carbon particles: From near a major highway to the broader urban plume in Los Angeles, Atmos. Chem. Phys., 18, 11991–12010, https://doi.org/10.5194/acp-18-11991- 2018, 2018.**

**Ko, J., Krasowsky, T., and Ban-Weiss, G.: Measurements to determine the mixing state of black carbon emitted from the 2017–2018 California wildfires and urban Los Angeles, Atmos. Chem. Phys., 20, 15635–15664, https://doi.org/10.5194/acp-20-15635-2020, 2020.**

*(16) Line 5 – Do you mean "have", rather than "comprised BC particles"?*

**This sentence is revised.**

*(17) Lines 8-12 – Are you saying here that you can identify the sources contributing to the rBC based on the MMD of the rBC component of the particle?*

**No. Though MMD is an indicator, unequivocal identification of sources is not possible based on the MMD values of the rBC particles alone. This is explicitly stated in the revised manuscript.**

**Page 12, Line 3:**

"**It may be noted that the exact sources cannot be identified from the MMD value of rBC size distributions alone. More details on source apportionment are provided in section 3.3**".

*(18)Lines 11-12 – Your MMD fall in line with the "aged continental outflow" you provide 11 references for. You refer to the continental outflow studies by saying "On the other hand", which suggests that they are different from a combination of urban emissions and bio+coal emissions. Are you making your determination on lines 11-12 based on the 11 studies you reference, or are you assuming that your results are a combination of urban emissions and the bio+coal emissions?*

**Based on the supporting evidence from the previous studies, we argue that the results suggest a combination of mixed emissions and aging. Also, complying with both the reviewers' comments, the discussion has been revised as below:**

**Page 11, Line 13:**

[revised manuscript text omitted]

Zhang, R., Khalizov, A. F., Pagels, J., Zhang, D., Xue, H., and McMurry, P. H.: Variability in morphology, hygroscopicity, and optical properties of soot aerosols during atmospheric processing., Proc. Natl. Acad. Sci. USA, 105, 10291–10296, doi:10.1073/pnas.0804860105, 2008.

*(19)Lines 13-18 – You suggest that the slight increase in MMD of the rBC over the NIO- E region is due to a complex set of processes (self-coagulation and sedimentation) occurring in those particle populations rather than differences in source types, even though you have already identified these regions as a mix of source types. Also, you give no reason why the same processes (self-coagulation and sedimentation) do not similarly affect the rBC in the other regions, in particular the NIO-W regions where the trajectories appear to have similar transport times. Why are source differences "less likely", given the large number of sources spread out over 1000-2000 km on both coastlines (east and west)?*

**This is an important point, as also commented by Reviewer-1. We have revised the discussion as under:**

**Page 11, Line 33:**

[revised manuscript text omitted]

*(21) For the above reasons (comments 16-21), this section, from lines 1-21, needs to be written with more clarity and justification.*

**Complied with. This portion is thoroughly revised in the manuscript.**

*Page 11-12:*

*(22) Table 2 and Lines 1-11, page 12 – There are measurements from the Arctic that could be added to this table. For example, Sharma et al. (ACP, 2017) found rBC MMD greater than 300 nm at a high*

*Arctic location.*

**Thanks for pointing this out. We have complied with and revised Table-2.**

[revised manuscript text omitted]

*Page 13:*

*(23) Line 8 – When you say dispersion, do you mean mixing with other air masses with smaller coating thicknesses? Reactions, unless leading to fragmentation and volatilization, are more likely to increase coating thickness than decrease it. Although there was relatively little precipitation during your study, transport across the SAS region might have impacted coatings. How might precipitation have affected the rBC size distribution and coating thickness over NIO-W? (I see you discuss this later with respect to the EIO region.)*

**Here dispersion and reactions refer to reduced availability of the coating material as we move away from the source regions. As the particles spend more time in the atmosphere, they tend to**

gain coating material on them. But as Referee-1 also has suggested, simultaneously, the loss of coating material on the particles cannot be ruled out due to photolysis or heterogeneous oxidation that can bring about fragmentation, leading to thinner coatings. Thus, preferential scavenging of larger particles leaving behind smaller and more thinly coated particles and atmospheric processes leading to loss of condensable material, explains lower RCT values observed over the EIO. This aspect is now described in the manuscript.

The passage of a large-scale meteorological system (due to western disturbances) which resulted in widespread rainfall over the SAS region, has occurred at a later period. During this period, extensive precipitation also occurred over the peninsular and western parts of the Indian mainland. But the passage of this system has happened after the cruise covered the NIO-W region. Thus, the large-scale meteorological system does not appear to have impacted the measurements of the NIO-W region and the outflow air masses originating from the western coast of India remained unperturbed.

*(24)Line 12 – "Nearly 95% of the SEAS measurements…"?*

**This sentence is revised in the manuscript as below:**

"**Nearly 95% of the observational points over the SEAS region indicated that rBC particles have an additional coating over its cores to the extent of > 90% of its size.**"

*(25)Line 15 – What are you contrasting with? Just start with "Over the NIO-E region…"*

**Complied with.**

*(26)Line 19 – Does that mean that sources on the east coast are stronger than on the west coast. This relates back to my comment 15 above.*

**Yes. We have revised this aspect in the manuscript.**

*Page 14:*

*(27)Line 9 – Smaller rather than inferior.*

**Complied with.**

**"It possibly resulted in the observed smaller coatings…"**

*(28)Lines 13-15 – It would be clearer to say "due to differences in respective coastal sources*

*(references) and possible transit times."* What are the relative differences in air mass transit times from the west coast to NIO-W and from the east coast to NIO-E, and what are the differences in east coast and west coast source strengths? See comment 20 above.

**Complied with. We have revised the sentence as suggested by the reviewer.**

**"Thus, a clear contrast in the mixing state parameters is evident which is due to differences in respective coastal sources (Moorthy et al., 2008; Peng et al., 2016; Gong et al., 2016), and possible transit times over these two regions.**

*(29) Lines 16-18 – Repetitious.*

**This sentence is revised and moved to the introduction portion in the revised manuscript.**

**Page 3, Line 26:**

**"The alteration to BC mixing state depends on various factors, which include the BC size distribution, nature of sources, the concentration of condensable materials that BC encounters during its atmospheric lifetime, and processes such as photochemical aging (Liu et al., 2013; Ueda et al., 2016; Miyakawa et al., 2017; Wang et al., 2018)."**

*(30) Lines 24-26 – If comparisons cannot be made, then what is the point of any of the measurements?*

**This sentence is revised as below:**

**Page 16, Line 7:**

**"As Cheng et al. (2018) and Ko et al. (2020) have highlighted, coating parameters derived from the SP2 instruments having different system configurations (detection limits of scattering intensity and range of volume equivalent diameters covered) and different techniques used in the estimation the optical diameters from scattering amplitudes (Metcalf et al., 2012; Gong et al., 2016; Raatikainen et al., 2017; Cheng et al., 2018; Liu et al., 2019; Ko et al., 2020) can vary considerably. This caveat needs to be borne in mind when making inter-study comparisons."**

*(31) Lines 27-29 – With the caveat that your study is not a Lagrangian experiment, and your "far-field" measurements are influenced by mixing with the surrounding environment.*

**Thanks. The point is made explicit in the revised manuscript.**

**Page 16, Line 13:**

**"Also, the earlier studies are mostly made in the 'near-field' situation, whereas the present study examined the coating characteristics in a 'far-field' scenario (far away from potential sources, especially NIO and EIO regions). The caveat here is that the present study is not a**

**Lagrangian experiment, and it is possible that the "far-field" measurements are influenced by mixing with the surrounding environment."**

*(32) Line 32 - What is "It" that needs further investigation: coatings, enhancement, radiative forcing?*

**Complied with. This sentence is revised as below:**

**"The implication of the observed thick coatings on BC to regional radiative forcing needs further detailed investigation in future studies".**

**Page 15:**

*(33) Line 8 – What does "very high" reference to?*

**Clarified as below:**

**"The non-BC (scattering) particle concentrations were higher (>1000 cm$^{-3}$) in the coastal waters (the SEAS),…"**

*(34) Lines 11-13 – I question your statement that rBC particles constituted about 25% to 30% of the total number concentration. Does the scattering particle concentration measured using the SP2 represent the total number concentration, and does the use of the scattering particles from the SP2 inflate the fractional value? For this fractional estimate, do you only include rBC components that have a detectable scattering signal? If so, then you could say that rBC particles constituted about 25% to 30% of the measured scattering particles. If not, then your fractional estimates are more difficult to interpret. Please correct here and in Table 3.*

**This is an important point as has also been pointed out by Reviewer -1. We have performed additional analysis and revised the manuscript. During the present study, the total particle concentrations are available from the condensation particle counter (with a 1-minute sampling interval). The results on particle number size distributions and new particle formation events during the ICARB-2018 are available in Kompalli et al. (2020b). As suggested by the reviewer, the fraction of BC-containing particles was calculated using the total number concentrations from the SMPS, CPC, and rBC number concentration from the SP2. Accordingly, panel (b) of figure 5 and Table-3 have been modified in the revised manuscript. Also, discussions in the manuscript are revised as below to reflect the above points.**

**Page 8, Line 10:**

**"Continuous measurements of the particle number size distributions in size range 10 to 414 nm have also been carried out aboard, at the 5-minute interval, using a scanning mobility particle**

sizer spectrometer (SMPS; TSI Inc., USA) during the campaign (Kompalli et al., 2020b). The SMPS consists of an electrostatic classifier (TSI 3080), a long differential mobility analyzer to size segregate the particles based on their electrical motilities (Wiedensohler 1988), which are subsequently counted by using a water-based condensation particle counter (TSI 3786). Concurrent measurements of the particle number size distributions in the aerodynamic diameters range 542 to 19800 nm (which can be converted to stokes diameters using an effective particle density) have also been made using the aerodynamic particle sizer (Make: TSI, Model: 3321) that works based on 'time-of-flight' technique (Leith and Peters 2003). Though the contribution from the particles in the sizes measured by the APS to the overall aerosol number concentrations is found to be < 2 %, combining both these measurements gives the total particle number concentrations covering a wide size range (10 -10000 nm). We have used the particle number size distributions (and total particle number concentrations) from 10-1000 nm from the SMPS+APS measurements, along with the number concentration of rBC from the SP2 to estimate the fraction of rBC-containing particles."

**Page 16, Lines 21-22:**

"The spatial variation of number concentration (in $cm^{-3}$) of non-BC (i.e., purely scattering) particles detected by the SP2 and the fraction of rBC-containing particles ($F_{BC}$; the ratio of rBC number concentration to the total number concentration in size range 10-1000 nm from the SMPS and APS measurements) are shown in Fig. 5 (a & b)".

**Page 17: Line 11:**

"The rBC particles constituted about 8-12% of the total number concentration over different sub-regions, on an average. (Fig. 5b)".

**Page 18: Line 2:**

[revised manuscript text omitted]

*(36) Lines 20–21 – Is the lower limit of detection of scattering particles the same in this study as it was in the Kompalli et al (2020a) study?*

**Yes. The same instrument has been used in both studies. This is now mentioned in the manuscript.**

*Page 16:*

*(37) Lines 3-14 – Above (see my comment 32), you say that comparisons of coating thickness cannot be made, yet here you compare fractional estimates from a few studies with no consideration given to potential differences in the scattering estimates from the studies. Please correct.*

**Complied with. This portion is rewritten in the revised manuscript.**

*(38) Line 20 – Terminate this sentence at India. You are only measuring a tiny bit of air relative to a large continent. Your results suggest there are similarities between the measurements. There may be something more to that, but you can't say the strength of emissions flowing from the north and south are similar based on this little bit of information. If you had extensive measurements from an airborne platform as well as your ship platform, you might be able to draw some inferences.*

**Complied with.**

**"…Raatikainen et al., (2017) for thickly coated BC particles in polluted outflow environments in northern India ".**

*(39) Lines 22-33 – The first and last sentences of this paragraph are repetitive. The mixing ratio is just another way of representing coating thickness, and those processes have been discussed. The reference to Liu is useful, but that is all that is needed here.*

**Complied with. This portion is modified in the revised manuscript.**

**Page 18, Line 26:**

**"The higher bulk mixing ratio of coating mass over rBC mass values ($M_{R,bulk}$ ~2.5-15) (Fig. 5d) are seen over the adjacent marine regions, which is due to the presence of thickly coated BC particles. Though lower compared to other sub-regions, substantial $M_{R,bulk}$ (~ 4.24 ± 1.45) values were found even over the EIO region. Such high $M_{R,bulk}$ values were reported in the literature from extremely polluted environments and biomass burning source dominant regions (Liu et al., 2017; 2019). The presence of such non-absorbing coated mass on the rBC cores has significant radiative implications. Recently, Liu et al. (2017) have examined the measured and modeled optical properties of BC as a function of mass ratio ($M_{R,bulk}$) under different environments and found that significant absorption**

**enhancement occurs when the coating mass over rBC mass is larger than 3. They suggested that in such a scenario (i.e., $M_{R,bulk} > 3$), the core-shell model reproduces the measured scattering cross-section."**

*Page 17:*

*(40) Lines 10-25 and Figure 6 – An interesting figure, but please indicate what data (where and when) are used in the construction of Figure 6. The last sentence (lines 23-24) is another case of repetition and too many words.*

**Complied with, gratefully. The portion is now revised as below.**

**Page 19, Line 15:**

**"The data collected from 16:41:24 hrs of 21-January-2018 to 18:02:46 of 22-January-2018 (Indian standard time) was used for the construction of the figure (which falls in the transition period between SEAS and NIO-E regions). The same analysis repeated for few other data sets over the other regions also yielded similar results."**

**The sentence in lines 23-24 is now deleted.**

**""**

*Page 18 Line 25 – I suggest writing this sentence as "The high proportion of thick coatings on BC particles may result in significant increases in absorption by the BC."*

**Complied with. The sentence is rewritten in the revised manuscript.**

**"The high proportion of thick coatings on BC particles may result in significant increases in absorption by the BC. As reported by Brooks et al. (2019)…"**

*Page 19:*

*(41) Lines 8-11 – Re-write as "The large uncoated rBC particles (core diameters > 0.18 µm and thin coatings of ACT < 50 nm) with low scattering enhancements were also found during our measurements but in smaller quantities, consistent with the findings of Brooks et al. (2019) for the Indian region during the pre-monsoon and monsoon seasons."*

**Complied with. The sentence is rewritten as suggested.**

**"The large uncoated rBC particles (core diameters > 0.18 µm and thin coatings of ACT < 50 nm) with low scattering enhancements were also found during our measurements but in smaller quantities, consistent with the findings of Brooks et al. (2019) for the Indian region during the**

**pre-monsoon and monsoon seasons."**

*(42) Line 9 – smaller*

**Complied with.**

*(43) Lines 18-19 – This should already be discussed in the introduction.*

**Complied with. This line is moved to the introduction portion.**

**Page 3, Line 15:**

"**The information on the nature of the coating material along with the state of mixing of BC particles gives insight into the magnitude of the mixing-induced absorption enhancement for BC (Cappa et al., 2012, 2018; Peng et al., 2016; Liu et al., 2017).**"

*(44) Lines 21-25 – Rather than spend four lines discussing circumstantial evidence, Figure 6 directly dispels the importance of this notion?*

**Complied with.**

**Page 21, Line 41:**

"**Though it may be possible for BC to be mixed externally with coarse mode aerosols like dust or sea-salt aerosols in the real atmosphere, figure 6 directly dispels the importance of this notion**"

*Page 20:*

*(45) Lines 10-12 – You can't call the SEAS region "organic-rich". Organics and sulphate are equal over SEAS in Figure 7.*

**Complied with. This is revised now as below:**

"**To summarise, the South Asian outflow plumes consist of** the more or less equal proportions of organics and sulfate aerosols **in the vicinity of west coast/peninsular India, gradually changing to a completely sulfate-rich aerosol system in the remote oceanic regions.**"

*(46) Lines 15-20 – You treat the change in organic composition from SEAS to EIO as a chemical loss of organics (this should be referenced). Is it not more likely that the change in organics is due to mixing of marine air and polluted continental air, as suggested by the EIO trajectories for EIO?*

**Complied with. The following references are added in the revised manuscript.**

"**Possible oxidation of primary particulate organic matter due to heterogeneous reactions involving oxidants such as OH, O₃, and NO₃ during long-range transport from anthropogenic source regions can result in their volatilization, restricting their lifetime (Molina et al., 2004; Donahue et al., 2006; DeCarlo et al., 2010). However, enhanced sulfate production is possible through gas to particle conversion in the SO₂ rich air masses (especially when ambient relative humidity is higher) (Unger et al., 2006; Meng et al., 2016) originated over the Indian region or through the dimethyl sulfide (DMS) pathway from the marine emissions (Zorn et al., 2008; Shank et al., 2012)**".

**The following references are added:**

DeCarlo, P.F., Ulbrich, I.M., Crounse, J., de Foy, B., Dunlea, E.J., Aiken, A.C., Knapp, D., Weinheimer, A.J., Campos, T., Wennberg, P.O. and Jimenez, J.L., 2010. Investigation of the sources and processing of organic aerosol over the Central Mexican Plateau from aircraft measurements during MILAGRO. Atmos. Chem. Phys. 10, 5257–5280. doi:10.5194/acp-10-5257-2010.

Donahue, N.M., Robinson, A.L., Stanier, C.O. and Pandis, S.N., (2006), Coupled partitioning, dilution, and chemical aging of semivolatile organics. Environ. Sci. Technol. 40, 2635–2643.

Meng, C.C., Wang, L.T., Zhang, F.F., Wei, Z., Ma, S.M., Ma, X., Yang, J., (2016), Characteristics of concentrations and water-soluble inorganic ions in PM2.5 in Handan City, Hebei province, China. Atmos. Res. 171, 133-146.

Molina, M.J., Ivanov, A.V., Trakhtenberg, S., and Molina, L.T., (2004), Atmospheric evolution of organic aerosol, Geophys. Res. Lett. 31, L22104. 10.1029/2004gl020910.

Shank, L. M., Howell, S., Clarke, A. D., Freitag, S., Brekhovskikh, V., Kapustin, V., McNaughton, C., Campos, T., and Wood, R., (2012), Organic matter and non-refractory aerosol over the remote Southeast Pacific: oceanic and combustion sources, Atmos. Chem. Phys., 12, 557-576, https://doi.org/10.5194/acp-12-557-2012.

Unger, N., Shindell, D. T., Koch, D.M., and Streets D.G., (2006), Cross Influences of Ozone and Sulfate Precursor Emissions Changes on Air Quality and Climate, PNAS,103, 4377–4380.

Zorn, S. R., Drewnick, F., Schott, M., Hoffmann, T., and Borrmann, S., (2008), Characterization of the South Atlantic marine boundary layer aerosol using an aerodyne aerosol mass spectrometer, Atmos. Chem. Phys., 8, 4711–4728, doi:10.5194/acp-8-4711-2008.

*(47) What do the details given in the paragraph starting on line 23 and ending on page 21, line 12,*

*have to do with BC? Most of this appears to be a rehash of Aswini et al. (2020), and could be stated much more succinctly.*

**Complied with. Most of this discussion is moved to supplementary information in the revised manuscript.**

**Page 23, Line 5:**

"**Further, the association between ammonium and sulfate (Fig. S3) indicated an NH$_4^+$ deficit environment (Aswini et al., 2020)**".

**The below discussion is moved to the supplementary.**

"**The mean molar ratios were ~1.44, 1.47, 1.43, and 1.25 mol/mol over the SEAS, the NIO-E, the EIO, and the NIO-W regions, respectively, which indicated an NH$_4^+$ deficit (Aswini et al., 2020). The spatial variation of molar ratios of ammonium to sulfate is shown in the Supplement (Fig. S3, panel a). The association between submicrometre sulfate and ammonium mass concentrations depicted an excellent correlation (r>0.95) during the ICARB-2018 (Fig. S3b in the Supplement) with varying slopes indicating the extent of neutralization (higher molar ratios and an increased tendency towards neutralization of sulfate near the coastal regions compared far away regions) of sulfate by ammonium. The present observation is consistent with those reported by Aswini et al. (2020), who suggested that *just-enough* ammonium was present to neutralize sulfate during the ICARB-2018 based on concurrent bulk aerosol chemistry using offline (filter paper sampling) technique. It is possible that when insufficient ammonium is present in the atmosphere, it can lead to the sulfate aerosols existing in forms other than (NH$_4$)$_2$SO$_4$. A probable reason for ammonium deficiency could be the difference in the lifetimes of gaseous ammonia and SO$_2$, which are precursors for particulate sulfate and ammonium. Though dominant sources were located over the land for both the precursors, the sulfate aerosol can be formed from the oxidation of transported SO$_2$ over the ocean, whereas ammonia is lost rapidly away from the source regions. This is evident from the varying slopes seen in Fig. S3b.**"

*Page 21:*

*(48)Line 13 to Page 22, line 21 and Figure 8 - The figure and the associated regressions do not give any significant information other than for the case of smaller coatings: sulphate appears to an important contributor to the coating thickness. However, the role of sulphate in that case has already been established from Figure 7. What might be useful here is a plot of the coating thickness against the sum of organics, sulphate and ammonium. If the sums of those species do not represent a significant fraction of the variation in the coating thicknesses, then either there is a problem with some*

*of the measurements (ACSM or coating thicknesses) or there is some chemical specie(s) not measured by the ACSM that is involved in the coating.*

**We agree that it is not possible to establish the direct association between the coating material and submicron aerosol chemistry with the present approach. Therefore, we have not attempted to quantify the exact contribution of organics/sulfate to the coating on rBC particles. We have examined the association between the non-refractory submicron aerosol mass concentrations (NR-PM1.0) from the collocated ACSM and bulk absolute coating thickness of rBC for low and high ACT observations to infer the most probable coating material. For the same mass concentration of NR-PM1.0 species, varying ACT values were seen quite frequently.**

**As suggested by the reviewer, we have examined the association between absolute coating thickness with the sum of mass concentrations of organics, sulfate, and ammonium, which is shown below. Here also, we have separated the data into two categories consistent with the earlier discussion: ACT is low (ACT < 50 % of MMD) and high (ACT > 50% of MMD). This is done so because regression analysis suggested two distinct regression lines. The bulk ACT showed a reasonable association (r > 0.48; p <0.01) with the sum of masses of organics, sulfate, and ammonium for low coating thickness scenario, whereas and the association is weaker (r ~0.31; p <0.01) during high observations. Corresponding slopes indicated that ACT on BC intensified more steeply with an enhanced mass concentrations of NRPM1.0 species (slope ~0.73) during the low ACT scenario compared to the high ACT scenario (slope ~0.52). The dominance of one species (sulfate) is also one probable reason for the increased scatter in the association shown below (This discussion is not part of the manuscript and is given here only for the reviewer).**

[Figure]

**Figure. Scatter plot between the sum of mass concentrations of organics, sulfate, and ammonium aerosols and bulk absolute coating thickness during the ICARB-2018 for the observations with (a) high ACT and (b) low ACT. Color represents the corresponding mass median diameter value. Solid lines represent the linear least-squares fit to the points. Regression slopes and correlation coefficients are written in each panel.**

As described in reply to Reviewer-1, we have found that a substantial portion of the non-refractory submicron aerosol mass is bound to the BC particles over the regions with stronger outflow (Supplementary Figure S4 and corresponding discussion included in the revised manuscript). While higher mean values of the ratio were observed over the SEAS (~35 ± 12 %) and NIO-E (31 ± 4) regions in the proximity of the sources, as we move away to the farther oceanic EIO, the NR-PM1.0 that is bound to BC particles dropped considerably (mean ~23 ± 8%). The values occasionally dropped to as low as 5-6%, indicating that (a) preferential loss of coating on rBC particle, or (ii) a substantial contribution from non-rBC particles to the NR-PM1.0 mass loading (Kompalli et al., 2020b). Though the mean fraction of r-BC containing particles remained similar (~8%) over different regions (SEAS, NIO-E and EIO), frequent new particle formation events (Kompalli et al., 2020b) resulted in lower $F_{BC}$ (< 2%) over the EIO region which reflected in the amounts of NR-PM1.0 mass that are bound to rBC particles. We have observed that new particle formation and subsequent growth to larger sizes preceded an enhancement in the mass concentrations of submicron aerosols (mostly organics and occasionally sulfate). This indicated that an increase in the mass concentration of submicron aerosols need not translate into the coating on rBC particles.

*(49) I recommend removing the discussion from line 23 of page 20 to line 21 of page 22 as well as Figure 8. Alternatively, you need to significantly improve the value of this discussion.*

**Regarding this suggestion, we feel that this analysis highlights the complexity in the quantification of the rBC mixing state in the outflow regions such as the present study region. We feel it is appropriate to present the association between the submicron aerosols and coating on rBC to infer the probable coating material in different scenarios. Therefore, we have improved the discussion in the revised manuscript as below and prefer to retain it. The following portion is added in the revised manuscript:**

**Page 24, Line 21:**

**"As described earlier, the mean fraction of r-BC containing particles remained similar (~8%) over different regions (SEAS, NIO-E and EIO). But, frequent new particle formation events**

(Kompalli et al., 2020b) have resulted in lower $F_{BC}$ (< 2%), which is reflected in the amounts of NR-PM1.0 mass that are bound to rBC particles. It was observed that new particle formation and subsequent growth to larger sizes preceded an enhancement in the mass concentrations of submicron aerosols (mostly organics and occasionally sulfate). This suggests another possibility of a reduction in the primary particulate pollution, and the available secondary condensate promotes nucleation in such cases. Also, the portion of the non-refractory submicron aerosol mass bound to the BC particles varied over a wide range of 10-40% over different regions, and especially lower values were seen during NPF periods (not shown here). In Figure S5 in the supplementary, the association between the sum of mass concentrations of organics, sulfate, and ammonium aerosols and bulk absolute coating thickness over different sub-regions covered during the ICARB-2018 is shown. This reveals a very good association (r~ 0.90) over the remote EIO and a weak association over the NIO-E regions, whereas no association was found over the regions SEAS and NIO-W. It also advocates that variation in the submicron aerosol composition can explain alterations to rBC mixing state over the remote regions, whereas the rBC mixing state would be complex in the vicinity of the source regions".

[Figure]

**Figure S5: association between the sum of mass concentrations of organics, sulfate, and ammonium aerosols and bulk absolute coating thickness over different sub-regions covered during the ICARB-2018."**

*Page 23:*

*(50)Lines 17-26 – As discussed above, the fractional analysis needs to be improved, and the associated conclusions on these lines altered accordingly.*

**Complied with.**

*(51)As above, the NR chemical composition discussion needs improvement.*

**Complied with.**

---

## Referee Report (RR1)

Manuscript: Mixing state of refractory black carbon aerosol in the South Asian outflow over the northern Indian Ocean during winder (Kompalli et al.,)

Review of the Revised Manuscript

The authors have done a very good job addressed the concerns raised by this reviewer - as well as Reviewer #2 - as outlined in their responses. This reviewer was happy to see that the BC number fraction did drop (nominally by 3x) highlighting the limitations of relying solely on the SP2 to detect non-refractory aerosol particles. Also, the authors reevaluated the role of self-coagulation. The reviewer appreciates the author's efforts of clearly delineating bulk coating thicknesses versus individual BC particle coating thicknesses. Finally, this reviewer was pleased to see that the suggestion of examining the ratio of non-refractory coating to overall non-refractory aerosol mass.

The manuscript is stronger for the extra effort of the authors and underscores the true value of the peer-review process: more accurate and better articulated science. The manuscript is now recommended for publication.

Review manuscript

The manuscript details the analysis of SP2 data collected aboard a research ship as part of the ICARB-18 campaign with a focus on the microphysical properties of refractory black carbon (rBC)-containing particles. As the authors point out, the SP2 detects individual rBC particles through laser-induced incandescence thereby providing mass loadings and size/mass distributions. By combining the incandescence channel with the co-located SP2 scattering channel, the mixing state of individual rBC particles can also be probed. Their analysis reveals that outflow regions are characterized by higher rBC loadings and more thickly coated rBC-containing particles with a decrease in coating thickness in the oceanic regions. Studies such as this are essential in filling in measurement and information gaps in the South Asian region and thus are important for improving our quantitative understanding of the radiative forcing impacts of BC aerosols in this region. Despite the over use of very long sentences, the manuscript is clearly written. The manuscript shares a lot similarity - with a couple of notable exceptions - to that published by Brooks et al., who conducted an aircraft-based study of northern India.

This Reviewer has a concern that the authors are, in some instances, over interpreting data collected by a single instrument and, in doing so, drawing potentially incomplete (or erroneous) conclusions - as exemplified by their use of rBC number fraction (which is discussed below). Therefore, it is recommended that the manuscript be revised as per comments below and resubmitted. To be clear, the findings contained in this manuscript are of good value and need to be published, but in its current state, the manuscript is not quite ready.

As alluded to above, 90% of this manuscript - based on rough estimate of pages dedicated to measurement type - involves only data collected by the single particle soot photometer (SP2). (The remaining 10% is on aerosol composition via an ACSM.) While the SP2 in an incredibly sensitive instrument towards rBC particles and its mixing state, it is not sensitive to changes in rBC aggregate morphology - that is rearrangement of a fractal aggregate to one that is more collapsed - and it suffers from limited detection sensitivity towards pure scatterers (i.e., non-BC particles).

First a comment on rBC morphology. For a given mass, the SP2 will report the same incandescence intensity independent of the rBC particle fractal dimension. Therefore a statement such as "The

observed narrow range of mean NMD (0.10-0.11 μm) and MMD (0.19- 0.20 μm) over the entire region (Fig. 3c and 3d) reveals that the prevailing BC over the entire study region has a mixed-source origin and the particles are aged during the long advection leading to transformation processes (such as collapsing of the BC cores)" is very misleading [page 10, lines 11-12].  The authors again put forth a similar line of reasoning in the conclusion [page 23, lines 4-5] "BC size distributions indicated highly mixed BC sources, likely including a combination of fossil fuel and solid fuel sources, along with the restructuring of the cores of the aged BC particles." The authors present no supporting data that there is any restructuring of the cores and thus cannot conclude that such processes are taking place.  Indeed, there is lab studies which suggest that aggregate restructuring occurs through capillary condensation at the interstitial locations between primary particles and thus such restructuring would occur very early in the coating process.   The authors are encouraged to review the work by the Khalizov group (e.g., Invanova et al., AST 2020 and references therein).

As highlighted above, this reviewer has a major issue with the authors operational definition of the "fraction of particles containing rBC" (which the authors define as the sum of the number concentration of BC and non-BC scattering particles detected by the SP2).  Using the SP2 to measure the number concentration of non-BC will severely bias the derived number fraction of rBC particles because the scattering channel has a severely limited detection range (i.e.,  ~200 nm - 400 nm).  In contrast,  the incandescence channel has a nominal detection sensitivity range, in terms of a mass equivalent diameter, from about 80 nm to ~ 500 nm.  In short, the SP2 severely undercounts pure scattering particles and therefore deriving a rBC number fraction using the SP2 scattering data will bias the rBC number fractions high and could very well mask the impact(s) that <200 nm non-BC particles might exert in differing regions.  One would think (hope) that the instrument suite deployed aboard the research ship during the campaign had a CPC measurement, which would provide an excellent measurement of aerosol number concentration for particles ranging in diameter from ~ 10 nm to 1000 nm, thereby greatly improving the utility of examining the rBC number fraction.   Given the presence of higher anthropogenic emissions nearest the coast, it is not unreasonable to expect sub-200 nm pure scattering particles to be abundant.  If no CPC, or equivalent measurement, is available then the discussion of variation in fraction of particles containing rBC is meaningless and thus should removed.

The authors report the observation of exceptionally thickly coated particles.  Brooks et al., from which the current manuscript is nominally modeled after, reported on the rBC mixing state in northern India during the pre-monsoon and monsoon seasons.  Using their Figure 6 "pre-monsoon (IGP)" plot it is estimated that a 130 nm BC core had a nominal coating thickness is ~110 nm.  A similar back-of-the-envelope estimate from the current manuscript's Figure 6 suggests a coating thickness of ~250 nm (the authors are strongly encouraged to adjust the scaling of the color mapping in Figure 6 so that more variation in the coating distribution might be seen).  This difference in coating thicknesses for a 130 nm core represents about a 4.5x difference in coating volume.  This is a big difference and as such, this Reviewer finds it a bit of stretch for the authors to simply state that "…the present study…..are comparable to the values reported in pollution in-plume air mass regions elsewhere (e.g., Cheng et al., 2019; Brooks et al., 2019)." [page 14, lines 19-20]  Given this large difference, surly the authors could offer some discussion on why such a difference exists; that is, why are their rBC coatings so much thicker?  Indeed, their reported coating thicknesses exceed what is observed for wildfires, which are known to rapidly create thickly coated rBC particles.  This is perplexing and interesting finding that begs the question of why is there such a big difference, yet, no further discussion is provided by the authors beyond a cursory comparison with Brooks.   The authors are encouraged to think about this finding and perhaps provide a more detailed discussion to help explain why the coating thicknesses found in this study are some much greater than that found by Brooks and in wildfires.

The authors report the organic contribution to the coating as being responsible for ~ 40% near the coast and in the remote ocean, accounting for a little over 20%, as inferred from the ACSM. What is the influence of having a mixture of sulfate and organics on the refractive index assumed in the coating calculation? Additionally, while a density of 1.7 g/cc is reasonable for sulfates, it is high for organics (1.4 g/cc). Perhaps authors could conduct some sensitivity calculations to see if these factors exert influence on the derived coating thicknesses.

The authors are sometimes using the term "aging" in a very confusing and, potentially, misleading way. For example, the authors write [page 14, lines 4-7] "While the larger BC particles are scavenged rather quickly, the smaller, less-aged, and relatively less-coated BC particles (occasionally, even bare soot particles) can persist in the outflow and be transported to the remote marine regions." This is misleading - more thinly coated BC particles are not necessarily "less-aged". Particles can undergo photolysis or heterogenous oxidation that can bring about fragmentation leading to material loss as particles age leading to thinner coatings or larger diameter particles can be preferentially scavenged leaving behind smaller and more thinly coated particle. So to label a smaller, less-coated BC particle - even bare soot particles - as less-aged is wrong. Correct this.

Do the authors have access to optical property measurements, specifically aerosol light absorption? If they do, they are strongly encouraged to roll that into this study so as to make this a more complete analysis.

Finally, it seems to this reviewer that the authors have additional data mining that they could do. Given their bulk coating mass-to-core ratio and the mass loadings of non-refractory aerosols via the ACSM, it might prove very interesting to examine the ratio of non-refractory coating to overall non-refractory aerosol mass. Using the bulk mixing ratio of coating mass over rBC mass (Table 3) along with the rBC mass for the SEAS leg (Table 3) and the reported NR-PM mass in Figure 7, a back-of-the-envelope calculation suggests that about ~40% of the NR-PM mass is bound to BC particles, whereas for the EIO leg this ratio drops to ~ 15%. This suggests the preferential loss of coating and also suggests that the rBC number fraction of particles does change significantly (as per this Reviewer's comments above on this subject).

Specific issues.

Page 3, line 5. "Produced by the incomplete (low-temperature) combustion of hydrocarbon fuels…" "Low temperature" is subjective, please quantify or remove.

Page 3, line 20. "The sources of BC are highly heterogeneous." What do the authors mean by "heterogenous"? There are three sources of BC: fossil fuel, biomass burning (wildfires and agricultural burns), and biofuel combustion. Please clarify what you mean by heterogeneous.

Page 3, line 20. "It has a long atmospheric lifetime." Compared to what? Certainly not CO2, which has a nominal lifetime of 100 years. The authors are encouraged to read the paper by Lund et al., (npj Climate and Atmospheric Science (2018)1:31) who report nominal BC lifetimes < 6 days. Please clarify what you mean by long atmospheric lifetime.

Page 8, lines 20-21. "….$E_{sca}$ is helpful in identifying the nature of sources." This is only true under the assumption of no material loss via oxidative and/or photochemistry that could either alter overall particle size and/or the refractive index of the coating.

Page 9, lines 13 and 15. "folds" should be singular. Please change.

Page 10, lines 12-18. The authors write "….the NIO-E region depicted slightly larger mean MMD (~0.20 µm) due to frequent larger values (35 % of the measurements showed MMD> 0.20 µm) compared to all the other regions (Fig. 3d). This is a result of either of two possibilities: (i) Self-coagulation of rBC cores due to enhanced atmospheric aging, which increases the rBC core diameters (at the same time, sedimentation of larger particles resulting in a large reduction in number concentration and mass concentration); (ii) The second and less likely possibility of a sizeable contribution (though not dominant) from solid fuel sources (biomass/crop residue/coal burning) in the upwind regions to the observed BC concentrations which were transported by the air masses traversing through the eastern coast of India and the Bay of Bengal" In figure 2c, the nominal rBC number concentration is ~200 /cc. Since coagulation goes a $N^2$, such a low concentration indicates that the rate of coagulation will be very slow. The authors should do a calculation to ascertain whether such low concentrations can result in enough self-coagulated rBC particles within the time frame of there measurements to account for ~ 35% of the measurements in this region showing MMD> 0.2 um. Also, this Reviewer is quite surprised that there is very little discussion about the possible impacts of cloud processing of rBC particles that this might have on this observation (not to be confused with scavenging).

Page 15, lines 14-17. Do the authors have access to any independent measurements of size distributions beyond solely relying on the SP2? As pointed out above, independent measurements of the microphysical properties would help in buttressing interpretations and, in some cases (e.g., rBC number fraction) enable a robust analysis to be performed.

---

## Author Response (AR2)

**Response to reviewer**

We thank the reviewer for constructive suggestions, which have helped improve the manuscript. This document outlines the review comments in *plain italics*, followed by the author's response in bold, and the tracked changes in the main texts are in blue.

*Reviewer's comments:*

*The authors have substantially improved the paper. I have a few very minor comments related to the responses for the authors to consider.*

*1. Original Comment: Page 3, Line 21 - What is it about sources of BC in a clean environment that reduce its relative aging? Are BC emissions unaccompanied by fewer other emitted components?*

*Authors Response: Yes. Higher concentrations of condensable vapors (some of which are co-emitted species) contribute to faster aging of BC (Wang et al., 2014; Peng et al., 2016) in polluted environments, as compared to cleaner environments.*

*New Comment: When you refer to sources of BC in a clean environment, are you saying that there are no co-pollutants emitted with BC? What might be some sources of BC for which co-pollutant emissions are of less or little relative consequence?*

Sorry for the confusion created here. 'Relatively cleaner environment' refers to the regions where fewer amounts of condensable vapors prevail, thereby contributing to less-faster aging of BC than the polluted regions. As the reviewer rightly pointed out, BC  sources alone may not play much role in determining its relative chemical-ageing. This portion was revised in the first revison of the manuscript and is given below. " The alteration to BC mixing state depends on various factors, which include the BC size distribution, nature of sources, the concentration of condensable materials that BC encounters during its atmospheric lifetime, and processes such as photochemical ageing (Liu et al., 2013; Ueda et al., 2016; Miyakawa et al., 2017; Wang et al., 2018). Consequently, the nature and extent of coating on BC vary in space and time, and as such, BC in a polluted environment chemically-ages faster than in a relatively clean environment (e.g., Peng et al., 2016; Liu et al., 2010, 2019; Cappa et al.,2019)."

*2. In response to a comment regarding lines 30-32 of the original manuscript, the authors say "When air masses from such complex source regions are transported to remote regions devoid of any BC sources, the mixing state of BC becomes complicated. This is due to (a) restructuring of the BC aggregates during the transport due to different processes (Kutz and Schmidt-Ott, 1992; Weingartner et al., 1995; Slowik et al., 2007b; Pagels et al., 2009), and (ii) varying nature and amounts of coating material arising due to the*

*different atmospheric lifetimes and microphysical processes involving different species (McFiggans et al., 2015)."*

*New Comment: Rather than say the mixing state become more complicated, which I think is arguable, I suggest saying the mixing state may change. Also, make the bullets consistent.*

Complied with. We have incorporated these suggestions in the revised manuscript.

*3. Original Comment: Page 4, Line 28 - Do you know that particles smaller than 10 um were efficiently sampled or are you just assuming they were?*

*Authors Response – "Prior to the experiment, we have characterized the sampling inlet system and examined the sampling losses, both theoretically and experimentally. For this purpose, the particle number size distribution measurements were used. We found that the difference between the number concentrations with and without inlet system was < 5% for sizes up to 1000 nm, 5-20% between 1000-6000 nm, and > 30 % for sizes > 6000 nm."*

*New Comment: Good, but do you mention these results anywhere in the manuscript or supplement?*

We have not mentioned these details in the manuscript. The inlet design is based on Global Atmospheric Watch (GAW) guidelines/recommendations for aerosol sampling (WMO/GAW, 2016).

*4. New Comment: You use "aged" in many places, but it is an ambiguous term. 'Aged' is a temporal term, yet here its use in aerosol science most often pertains to the particle chemical composition rather than time. The Arctic aerosol can be one example of an aged aerosol both temporally and chemically. On the other hand, a biomass burning or biogenic aerosol may be well aged in a chemical sense (although not necessarily completely) on a much shorter time scale. It seems unreasonable to call them both 'aged' without any qualification. I suggest using "chemically-aged".*

Complied with. We have explicitly stated ' Chemically-aged" in the appropriate text throughout the revised manuscript.

*5. New Comment concerning comment on Page 11, Line 33 of original manuscript: Rather than "The insoluble BC particles…", I suggest "Less-soluble BC-containing particles may be interstitial within a non-precipitating cloud."*

Complied with.